# SegDAC: Improving Visual Reinforcement Learning by Extracting Dynamic Objectc-Centric Representations from Pretrained Vision Models

## Abstract

Visual reinforcement learning (RL) is challenging due to the need to extract useful representations from high-dimensional inputs while learning effective control from sparse and noisy rewards. Although large perception models exist, integrating them effectively into RL for visual generalization and improved sample efficiency remains difficult. We propose **SegDAC**, a **Seg**mentation-**D**riven **A**ctor-**C**ritic method. SegDAC uses Segment Anything (SAM) for object-centric decomposition and YOLO-World to ground the image segmentation process via text inputs. It includes a novel transformer-based architecture that supports a dynamic number of segments at each time step and effectively learns which segments to focus on using online RL, without using human labels. By evaluating SegDAC over a challenging visual generalization benchmark using Maniskill3, which covers diverse manipulation tasks under strong visual perturbations, we demonstrate that SegDAC achieves significantly better visual generalization, doubling prior performance on the hardest setting and matching or surpassing prior methods in sample efficiency across all evaluated tasks.

## 1 Introduction

Reinforcement learning (RL) from images is challenging because of the high dimensionality of visual inputs, their variability across environments and the limited robustness of policies under visual perturbations (Mnih et al., 2013; Pathak et al., 2017; Yarats et al., 2020; Lepert et al., 2024; Yuan et al., 2023; Lyu et al., 2024; Lepert et al., 2024). A natural idea is to leverage pre-trained vision models, but integrating them into online RL has proven to be challenging; one must choose the right layer to feed into the policy, manage inference costs during large-scale experience collection, and potentially adapt features that were not designed for RL in the first place. In parallel, many visual RL methods rely on data augmentation to improve robustness, while this can help, it adds complexity and does not address the core issue of learning meaningful general structure from images.

Images, like sentences, have an inherent structure. Words in a sentence carry meaning beyond individual characters, and similarly, objects in a scene provide more useful building blocks than raw pixels or patches (Chen et al., 2024; 2025; Locatello et al., 2020; Wang et al., 2023). Conventional pixel-based methods extract features from uniform grids, overlooking the object-level structure of the scene. By contrast, object-centric representations can capture entities and their relationships, offering a more abstract and semantically meaningful description of the environment. Prior segmentation-based RL approaches have explored this direction by decomposing an image into segments, meaning distinct parts of the image, but they often relied on fixed slots, precomputed masks, or strong supervision, which limited their flexibility and general applicability.

To address these challenges, we introduce **SegDAC** (**Seg**mentation-**D**riven **A**ctor-**C**ritic), a practical method that uses pretrained vision models to produce segments and a variable number of object embeddings at each step that are grounded by text inputs. We extract these embeddings with a simple and efficient procedure that preserves both local and contextual information from SAM patch features. Our transformer-based actor-critic then processes the segment tokens together with proprioception and action inputs, allowing the policy to focus on the objects that matter.

Table 1: Comparison of segmentation-based visual RL methods.

| Criterion | FTD | SAM-G | SegDAC (ours) |
|---|---|---|---|
| Needs Human Labels (RL Training) | ✓ No | ✗ Yes (Ground Truth Object Masks & Frames) | ✓ No |
| Leverages Pretrained Vision Models | ✓ Yes | ✓ Yes | ✓ Yes |
| Leverages Text | ✗ No | ✗ No | ✓ Yes |
| Representations per Frame | ✗ Fixed | ✗ Fixed | ✓ Dynamic |
| Learns From Sequence of Object Representations | ✗ No (Masked Pixels) | ✗ No (Masked Pixels) | ✓ Yes (Objects Embeddings) |
| Needs Auxiliary Losses | ✗ Yes | ✓ No | ✓ No |
| Needs Data Aug. (RL Training) | ✓ No | ✗ Yes | ✓ No |
| Grounded Segmentation | ✗ No (prompt-free) | ✓ Point Prompts | ✓ Text-Grounded |

Unlike most visual RL methods that learn an encoder from raw pixels, SegDAC operates fully in latent space. It does not rely on reference frames, ground truth segmentation masks, labels, reconstruction losses, or data augmentation, which makes SegDAC easy to use. To our knowledge, **SegDAC is the first online RL method that learns from dynamically computed, variable-length object embeddings using RGB images**.

Our main contributions are:

**Dynamic object-centric RL.** We propose a transformer-based actor-critic that operates directly on a variable-length set of segment embeddings, with no image reconstruction stage. Similar to how ViT clarified how to use image patches with transformers, our design specifies how to extract segment embeddings while keeping context, how to build queries and keys, and how to fuse proprioception and actions in a stable way for online RL. Our method is robust to variations in segment count and granularity (see section 5.2), which is important for real-world settings, and it learns which objects matter on its own, without any ground truth masks or reference frames.

**Text-Grounded Segmentation for Online RL.** To our knowledge, SegDAC is the first method to use text-grounded segmentation for online RL and to learn from a variable number of segment embeddings.

**Strong visual generalization.** We introduce a new ManiSkill3 visual generalization benchmark with strong out-of-distribution perturbations, SegDAC achieves up to 2x improvement in the hardest settings while matching or exceeding the sample efficiency of state-of-the-art baselines.

**Faster SAM-based training and inference.** Our unified design produces light segment embeddings, fast text-guided segmentation, simple mask post-processing, and fully latent-space training. This leads to a 2 to 5x speed-up compared to previous SAM-based approaches. Since our model processes segments instead of dense patches, it handles fewer tokens, which further improves inference speed. See Appendix K.

**A new direction for visual RL.** SegDAC is built from scratch, uses only the SAC loss, and does not rely on data augmentation, auxiliary objectives, or external datasets. This suggests that strong visual generalization can be achieved with a lighter and more direct pipeline, in contrast to many recent methods that build on DrQ-v2 and depend on heavy augmentation and additional loss terms during RL training.

## 2 RELATED WORK

We review prior work in model-free visual RL along three main themes: (1) learning directly from pixels, (2) leveraging pretrained models, and (3) segmentation/object-centric approaches. Tables 1 and 47 summarize how SegDAC compares.

**RL from Images:** Several model-free RL methods aimed to improve sample efficiency or robustness directly from pixels. SAC-AE (Yarats et al., 2020) used auxiliary reconstruction losses. RAD (Laskin et al., 2020), DrQ (Kostrikov et al., 2021), SODA (Hansen & Wang, 2021), SVEA (Hansen et al., 2021), DrQ-v2 (Yarats et al., 2021), and later methods such as MaDi (Grooten et al., 2023) and SADA (Almuzairee et al., 2024) improved generalization through data augmentation and training stabilizers. While effective, these approaches remained pixel-based. SegDAC avoids auxiliary losses and data augmentations entirely, showing that strong generalization can be achieved without techniques previously considered necessary. **Pretrained Vision Models in RL:** PIE-G (Yuan et al.,

2022), SAM-G (Wang et al., 2023), and IWM (Garrido et al., 2024) all adapted ImageNet pretrained encoders for RL. PIE-G remained pixel-based. SAM-G, built on PIE-G, used offline SAM features with human labels, fixing object representations before training. IWM applied pretrained encoders in a world model framework. SegDAC differs by integrating pretrained segmentation and detection models directly in online RL, producing dynamic, text-grounded object embeddings without labels or offline datasets. **Segmentation for RL:** Segmentation has been used in RL either to train a segmentation model with RL signals (Melnik et al., 2021) or to guide policies using SAM (Kirillov et al., 2023). Saliency-based methods predicted pixels of interest but lacked semantic abstraction (Zhang et al., 2024a; Wu et al., 2021; Wang et al., 2021; Bertoin et al., 2023). SAM-G (Wang et al., 2023) relied on offline ground truth segmentation masks to extract precomputed object features (see Appendix 43 for a visual of these ground truth masks), while FTD (Chen et al., 2024) applied prompt-free SAM segmentation and trained a CNN policy on stitched images, limiting flexibility and enforcing a fixed segment count. In contrast, SegDAC computes dynamic object representations online without ground truth segmentation masks, grounds EfficientViT-SAM (Zhang et al., 2024b) predictions with text inputs, and learns directly from a variable-length sequence of segment embeddings in latent space (table 1).

## 3 BACKGROUND

Visual reinforcement learning (RL) is commonly formulated as a Partially Observable Markov Decision Process (POMDP) (Littman, 2009), where the agent receives high-dimensional image observations without access to the full environment state. A finite-horizon POMDP is defined by $(S, O, A, P, R, \gamma)$, where $S$ is the state space, $O$ the observation space (e.g., RGB images), $A$ the action space, $P$ the unknown transition function, $R$ the reward function, $\gamma \in [0, 1]$ the discount factor, and $H$ the finite horizon of each episode. The objective is to learn a policy $\pi : O \mapsto A$ that maximizes the expected discounted return $\mathbb{E}_\pi \left[ \sum_{t=0}^{H-1} \gamma^t r_t \right]$.

We build on the Soft Actor-Critic (SAC) algorithm (Haarnoja et al., 2018; 2019), a widely used actor-critic method for continuous control (Yarats et al., 2020; Laskin et al., 2020; Kostrikov et al., 2021; Hansen & Wang, 2021; Hansen et al., 2021; Bertoin et al., 2023; Yuan et al., 2022; Grooten et al., 2023; Chen et al., 2024; Almuzairee et al., 2024). SAC maximizes both return and policy entropy: the critic minimizes the Bellman residual, while the actor maximizes the critic's Q-value under the current policy. For the exact mathematical formulation and loss functions, we refer readers to the original SAC papers, since SegDAC uses the vanilla SAC losses during training without modification. Unlike standard visual RL pipelines that stack multiple frames, we extract segment embeddings from a single frame at each time step and process them as a variable-length sequence.

## 4 METHOD

An overview of SegDAC is shown in Figure 1. Given an RGB image and a set of grounding text inputs, the grounded segmentation module (section 4.1) uses YOLO-World (Cheng et al., 2024) to generate bounding boxes based on the text inputs and EfficientViT-SAM (Zhang et al., 2024b) to produce segment masks and patch embeddings. The segment embedding extraction module (section 4.2) extracts features within each mask to form a variable-length sequence of segment embeddings, which is then passed to a transformer-based actor critic (section 4.3) to predict actions/Q-values.

Operating directly on segment embeddings enables reasoning over semantically meaningful regions while avoiding pixel-level noise. Further details on the grounded segmentation module and segment embeddings extraction module are provided in the next section. Refer to the appendix for text inputs, definitions, and implementation details.

### 4.1 GROUNDED SEGMENTATION MODULE

Our grounded segmentation module, shown in Figure 3, is inspired by Grounded SAM (Ren et al., 2024), which combines Grounding DINO (Liu et al., 2024) with SAM (Kirillov et al., 2023) for text-guided segmentation. However, we found it too slow for online RL. Instead, we adopt a faster pipeline: YOLO-World (Cheng et al., 2024) generates bounding boxes from an open-vocabulary

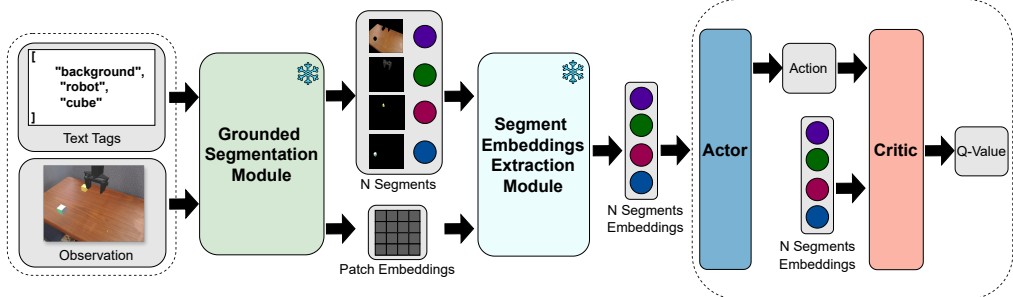

Figure 1: SegDAC overview and components.

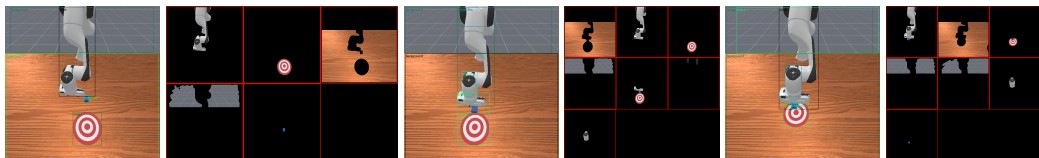

Figure 2: Examples of grounded segments for the PushCube task. Left: image with YOLO-World bounding boxes. Right: segments extracted by EfficientViT-SAM using boxes.

of simple text inputs and EfficientViT-SAM (Zhang et al., 2024b) extracts segments within those boxes. For simplicity, we refer to EfficientViT-SAM as SAM throughout the rest of the paper. A key characteristic of our module is that the number of output segments, $N$, is variable across different time steps, unlike methods that operate on a fixed number of object representations (Chen et al., 2024; Locatello et al., 2020; Shi et al., 2024; Daniel & Tamar, 2022).

Prompt-free segmentation relies on a dense grid of points and requires heavy post-processing. To avoid this inefficiency, we adopt a prompt-based design. We use YOLO-World to detect objects using a list of text tags (eg: "cube", "apple", "bowl", ...). This detector predicts $K$ bounding boxes in zero-shot. We pass these boxes to SAM as prompts to produce $N$ segments per frame. The number $N$ varies naturally across time. This approach directs SAM toward task-relevant objects and provides semantically grounded segmentations.

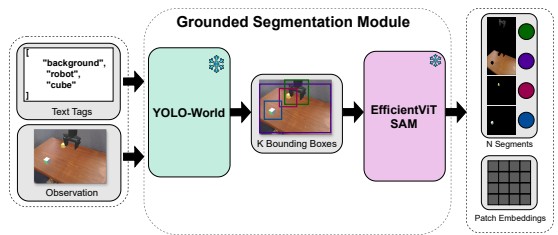

Figure 3: Our grounded segmentation module leverages the strengths of both YOLO-World and EfficientViT-SAM models to efficiently detect segments from the image.

The text inputs consist of simple object names or small groups of words/concepts instead of long descriptions. This reduces ambiguity and makes text grounding practical for online RL. These inputs are flexible: they can be defined per task, shared across tasks, or generated by external models like VLMs (Vasu et al., 2025) or image taggers (Zhang et al., 2023b; 2025). For simplicity, we define task-specific lists in our experiments. However, we could also concatenate them into a shared list of concepts. Since YOLO-World is incentivized to output boxes only for visible objects, our architecture naturally supports a shared vocabulary across tasks.

We demonstrate this flexibility in Appendix B.1.2 and Appendix B.1.3. We show that SegDAC remains stable when replacing text inputs with synonyms or when using text inputs that are shared across all tasks. For efficiency, we reuse the pretrained patch embeddings from the SAM encoder (Zhang et al., 2024b; Kirillov et al., 2023) to compute segment embeddings (Section 4.2). Both YOLO-World and SAM remain frozen; we discuss this choice in Appendix L.

Figures 2 shows examples of the grounded segments extracted for the PushCube-v1 task. While YOLO-World's predictions are not always perfect, they effectively decompose the scene into bound-

ing boxes based on the text tags. Notably, incorrect label predictions do not affect segmentation quality, as SAM relies only on bounding box coordinates, helping avoid compounding errors.

Motivated by our ablation experiments in Appendix B.1, we also include a generic "background" text tag to encourage detection of additional segments. This helps the agent learn to ignore irrelevant regions and improves generalization compared to using only task-specific tags. The set of text inputs is kept broad and simple.

## 4.2 SEGMENT EMBEDDINGS EXTRACTION MODULE

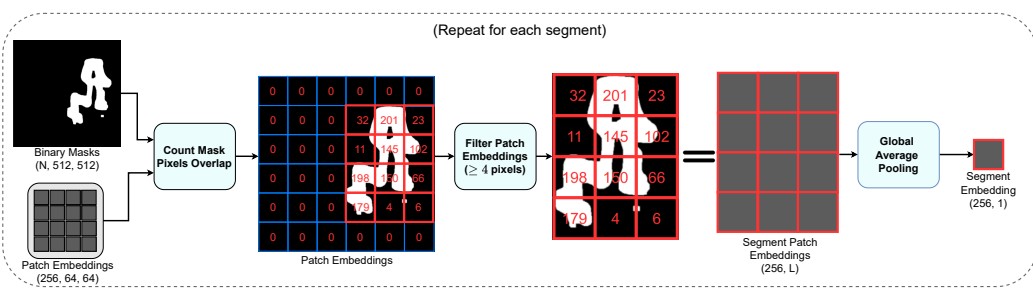

Figure 4: Segment embedding extraction: For each SAM-predicted mask, we select patch embeddings with sufficient overlap ($\geq 4$ pixels), then apply global average pooling to obtain one embedding per segment.

Our segment embeddings extraction module takes as input $N$ binary segmentation masks and the patch embeddings from SAM's encoder, and outputs $N$ segment embeddings, where $N$ varies from frame to frame. Our approach is not tied to a specific image encoder, although we use SAM's patch embeddings for efficiency, the architecture is compatible with other encoders such as DinoV3 (Siméoni et al., 2025). As shown in Figure 4, this module has no trainable parameters.

To compute a segment embedding, we first identify the SAM patch embeddings that overlap spatially with the segment's binary mask. For each patch, we count how many active pixels (value = 1) from the mask fall inside the patch boundary. Patches with fewer than a small threshold (eg: 4 pixels) are discarded. We then apply global average pooling over the remaining relevant patch embeddings to produce a single embedding vector of dimension $S$ (eg: $S = 256$). This procedure is applied independently for each of the $N$ masks.

This extraction process is very fast in practice. Since SAM's patch embeddings already include contextual information from the full image, the resulting segment embeddings also carry shared context. We believe this shared context helps SegDAC remain robust when parts of objects are missed by the segmentation module. We show this effect in more detail in section 5.2. This contrasts with prior work that encodes each segment independently without shared context (Shi et al., 2024), which can lead to weaker representations when segmentation is imperfect.

An analysis of the segment embedding extraction process including patch overlap heatmap, mask-patch alignment is provided in the Appendix F.

## 4.3 ACTOR CRITIC NETWORKS

Our actor and critic each use their own transformer decoder, with separate weights, projection heads, and encoding layers. In both cases, the inputs are segment embeddings, proprioception and a learned query token. Following the same motivation as using patches in ViT, we treat each segment as a token and design the architecture so it can process these tokens in a stable way for online RL. Each token receives a learned token-type encoding so the model can distinguish between segments, proprioception and the learned query. Segment tokens also receive a positional encoding based on bounding box coordinates, which provides a simple spatial reference that matches the object-centric structure of the inputs and is crucial to allow the network to distinguish between two identical objects but that are in a different positions in the scene.

For the critic, the query is formed by concatenating the action vector with a learned token and projecting the result through a small MLP. We found this design to work best in practice compared

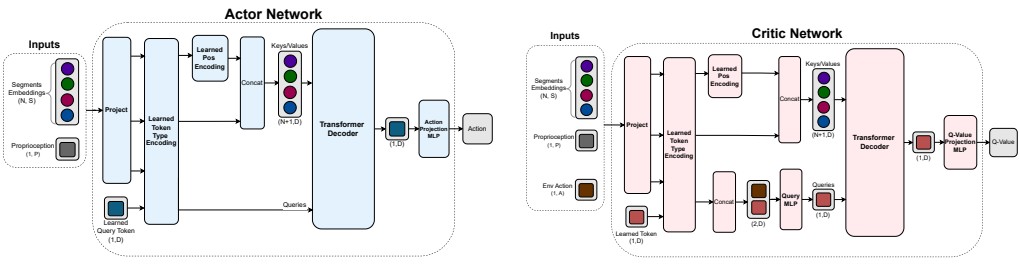

Figure 5: SegDAC actor critic networks architectures.

to other approaches. The keys and values come from projecting the set of segment tokens, the proprioception token and the learned token into a shared hidden space. The decoder attends over this set and produces a single output token that is mapped to the Q-value through a projection head. The actor uses the same design, but the learned query token replaces the action input, and the output token is projected to the action space. Because the computation operates directly on segment embeddings, the model processes far fewer tokens than patch-based encoders and can focus on the objects that matter without supervision. The output projection can also be swapped to support other RL algorithms such as DQN or TD3 without modifying the rest of the architecture.

The full actor-critic architecture is shown in Figure 5.

## 5 EXPERIMENTS

We evaluate SegDAC on a new visual generalization benchmark built on ManiSkill3 (Tao et al., 2024), covering 8 manipulation tasks with three difficulty levels (easy, medium, hard) and 12 visual perturbations. This benchmark uses clear criteria rather than random sampling, enabling systematic evaluation. We compare SegDAC with state-of-the-art baselines on sample efficiency and generalization, and further analyze its robustness to segment variability and its segment-level attention patterns. We also provide extensive quantitative and qualitative results, case studies and failure analyses in the appendix (Section A.5), showing how SegDAC performs under strong visual perturbations.

**Training and Evaluation Protocol** Following prior work in model-free visual RL (Hansen et al., 2021; Almuzairee et al., 2024), all agents were trained for 1M environment steps without perturbations across 8 tasks (Franka Panda arm and Unitree G1 humanoid). Policies were evaluated every 10K steps on unperturbed seeds (10 rollouts, 5 seeds). Hyperparameters were shared across tasks, with only text inputs differing. We present an analysis on the training speed of our method against SAM-G and FTD in appendix K.

**Generalization Benchmark** To assess visual generalization, we extended ManiSkill3 with 8 tasks, 3 difficulty levels, and up to 12 perturbation types per task. Although DMC-GB (Hansen et al., 2021) has been widely used, Yuan et al. (2023) showed that it does not fully capture the demands of realistic environments. ManiSkill3 offers a stronger basis for robotics simulation, and our benchmark increases the difficulty further by introducing a wide range of visual and semantic perturbations. Each condition was tested with 5 seeds and 50 rollouts, and we report interquartile mean (IQM) returns with 95% confidence intervals following Agarwal et al. (2022). Additional details, including visual examples of the perturbations and our scores aggregation, are provided in Appendix A.

### 5.1 VISUAL GENERALIZATION BENCHMARK AND RESULTS

Inspired by The Colosseum benchmark (Pumacay et al., 2024), we define the following taxonomy of scene entities:

**Manipulation Object (MO):** The task-critical object directly grasped or moved by the robot's end-effector to complete the main objective.

**Receiver Object (RO):** An essential object not directly manipulated, but necessary for task completion by interacting with the MO.

**Primary Interaction Surface (Table):** The main horizontal surface that supports or anchors inter-

actions between MOs and ROs.

**Background Elements:** All other visual components not classified above (e.g., walls, peripheral floor, skybox, distant scenery).

**Visual perturbation categories** include: (1) camera (pose, field of view), (2) lighting (direction, color), (3) color (MO, RO, table, background) and (4) texture (MO, RO, table, background).

**Difficulty levels.** Table 2 summarizes the three difficulty levels used in our visual generalization benchmark.

**Easy** involves minor visual changes that remain close to the default configuration, maintaining high similarity. These serve as smoke tests to verify robustness against small, non-disruptive variations.

**Medium** introduces substantial visual changes that reduce similarity to the default view while remaining task-valid. Perturbations are designed to avoid semantic conflicts between scene entities (for example, avoiding textures or colors that could alter an object's meaning). These settings are visually distinct yet realistic and solvable.

**Hard** applies aggressive out-of-distribution perturbations that introduce both visual and semantic challenges. Configurations differ strongly from the default view, with extreme changes in camera angles, lighting, colors, or textures that may confuse perception (for example, a cube textured like the target). They are intentionally difficult and not optimized to facilitate task completion.

For extended details on benchmark definitions as well as quantitative and qualitative results across tasks and perturbations, we refer the reader to Appendix A.

**Results:** Figure 7 reports IQM returns normalized by each task's maximum steps, grouped by perturbation category. We test six visual RL baselines: SAC-AE (Yarats et al., 2020), DrQ-v2 (Yarats et al., 2021), SAM-G (Wang et al., 2023), MaDi (Grooten et al., 2023), SADA (Almuzairee et al., 2024) and SMG (Zhang et al., 2024a). SAM-G uses ground-truth masks and reference frames to build point prompts, examples are in Appendix 43.

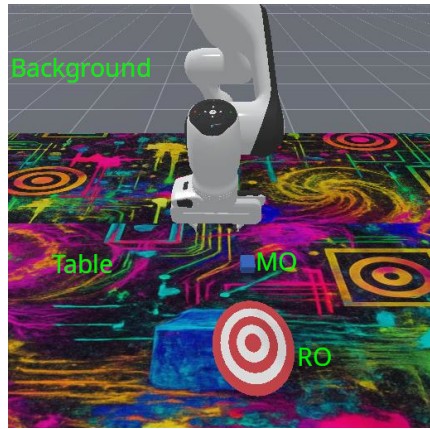

Figure 6: Hard visual perturbation in the PushCube task, with scene entities labeled using our taxonomy.

Table 2: Difficulty levels.

| Difficulty | Visual change | Semantic perturbation |
|---|---|---|
| Easy | ✓ Low | ✗ None |
| Medium | ✓ Moderate | ✗ None |
| Hard | ✓ High | ✓ Present |

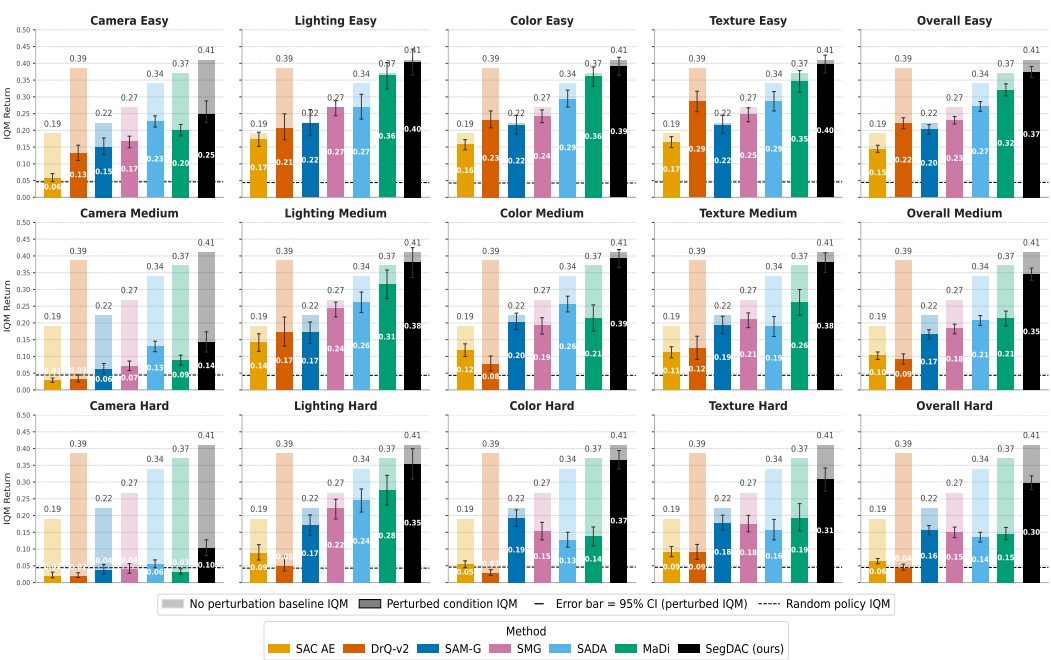

Figure 7: Visual generalization results grouped by categories.

Across categories and difficulty levels, SegDAC shows higher robustness than all baselines. Most methods perform reasonably under easy perturbations, but their performance drops sharply in the medium and hard settings. In contrast, SegDAC maintains stronger returns across all categories, including camera, lighting, color, and texture perturbations. For example, in the hard setting, MaDi loses roughly 60% of its performance, whereas SegDAC drops by only about 27%.

SAM-G, SMG, MaDi, and SADA are state-of-the-art methods for visual generalization, while DrQ-v2 is state-of-the-art in sample efficiency. Although the visual generalization methods degrade less than DrQ-v2, SegDAC still achieves about 2x their performance. All methods struggle in the camera-hard setting, highlighting the sensitivity of single-view architectures to large viewpoint shifts.

### 5.1.1 HARD MANIPULATION OBJECT (MO) TEXTURE AND HARD TABLE COLOR

Tables 3 and 4 highlight two hard perturbations designed to conflict with important parts of the scene. MO Texture replaces the manipulation object's texture with patterns that directly challenge its semantic identity and can conflict with other key scene elements (e.g., cube fragments, targets, chaotic textures), causing large drops for most baselines. SegDAC performs best, often reaching 2∼3x SAM-G's return. SAM-based methods tend to handle these perturbations better, while other approaches may overfit to specific colors instead of learning the object's semantics or geometry. Table Color matches the table color to the manipulation object, creating strong visual interference and exposing color overfitting. Examples for `PullCubeTool` and `PushCube` appear in Figure 8, with additional cases in Appendix A.

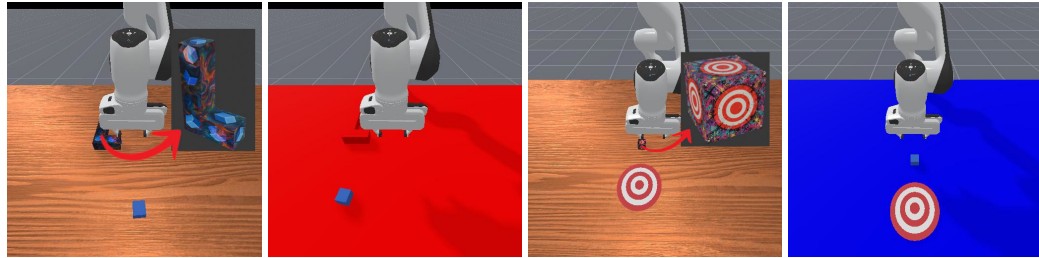

Figure 8: Hard Manipulation Object (MO) Texture and Table Color for `PullCubeTool` and `PushCube`.

Table 3: Hard Mo Texture Test

| Task | SAC AE | DrQ-v2 | SAM-G | SMG | SADA | MaDi | SegDAC |
|---|---|---|---|---|---|---|---|
| LiftPegUpright | 0.20±0.01 (-5.1%) | 0.22±0.01 (-53.9%) | 0.25±0.03 (-39.7%) | 0.22±0.01 (+0.0%) | 0.22±0.02 (-1.2%) | 0.21±0.01 (-4.4%) | **0.28±0.08 (-32.3%)** |
| PickCube | 0.06±0.02 (-69.8%) | 0.01±0.00 (-97.0%) | 0.11±0.01 (+3.3%) | 0.03±0.01 (-90.5%) | 0.02±0.01 (-94.4%) | 0.01±0.00 (-98.1%) | **0.15±0.01 (-55.2%)** |
| PokeCube | 0.08±0.01 (-54.2%) | 0.03±0.01 (-92.1%) | 0.07±0.01 (-77.6%) | 0.16±0.05 (-25.1%) | 0.02±0.02 (-95.1%) | 0.04±0.09 (-89.4%) | **0.22±0.03 (-45.0%)** |
| PullCube | 0.12±0.04 (-66.5%) | 0.10±0.03 (-71.4%) | 0.12±0.01 (-0.7%) | 0.27±0.07 (-29.8%) | 0.10±0.04 (-79.0%) | 0.26±0.12 (-46.2%) | **0.43±0.05 (-14.4%)** |
| PullCubeTool | 0.13±0.04 (-40.8%) | 0.12±0.08 (-81.2%) | 0.16±0.05 (-30.0%) | 0.11±0.05 (-54.3%) | 0.02±0.05 (-94.0%) | 0.03±0.02 (-94.6%) | **0.45±0.08 (-38.9%)** |
| PushCube | 0.13±0.02 (-55.9%) | 0.02±0.02 (-94.9%) | 0.23±0.06 (-54.3%) | 0.21±0.02 (-38.6%) | 0.08±0.03 (-82.7%) | 0.15±0.07 (-67.7%) | **0.45±0.06 (-0.4%)** |
| PlaceAppleInBowl | 0.05±0.02 (-32.0%) | 0.05±0.03 (-70.3%) | 0.06±0.03 (-2.8%) | 0.07±0.04 (-34.4%) | 0.04±0.04 (-66.6%) | 0.06±0.02 (-54.9%) | **0.29±0.18 (-7.4%)** |
| TransportBox | 0.05±0.03 (+1.6%) | 0.18±0.07 (-28.9%) | 0.27±0.01 (+0.3%) | 0.27±0.01 (+0.4%) | 0.23±0.09 (-5.2%) | 0.26±0.01 (-0.1%) | **0.28±0.01 (-0.1%)** |

Table 4: Hard Table Color Test

| Task | SAC AE | DrQ-v2 | SAM-G | SMG | SADA | MaDi | SegDAC |
|---|---|---|---|---|---|---|---|
| LiftPegUpright | 0.19±0.00 (-11.7%) | 0.19±0.01 (-60.0%) | **0.41±0.13 (-3.2%)** | 0.22±0.01 (-3.1%) | 0.22±0.01 (-0.5%) | 0.19±0.00 (-13.0%) | 0.38±0.14 (-8.1%) |
| PickCube | 0.02±0.01 (-88.5%) | 0.00±0.00 (-98.9%) | 0.09±0.01 (-11.3%) | 0.03±0.01 (-89.9%) | 0.03±0.01 (-92.2%) | 0.02±0.01 (-95.2%) | **0.11±0.01 (-67.9%)** |
| PokeCube | 0.02±0.01 (-90.5%) | 0.02±0.01 (-95.7%) | 0.27±0.06 (-15.0%) | 0.09±0.02 (-56.9%) | 0.12±0.04 (-68.9%) | 0.03±0.03 (-92.6%) | 0.20±0.03 (-49.9%) |
| PullCube | 0.02±0.02 (-94.0%) | 0.02±0.02 (-95.9%) | 0.12±0.01 (-0.9%) | 0.16±0.05 (-59.6%) | 0.27±0.04 (-43.9%) | 0.09±0.04 (-81.0%) | **0.35±0.07 (-31.1%)** |
| PullCubeTool | 0.03±0.02 (-85.9%) | 0.03±0.02 (-95.9%) | 0.11±0.05 (-52.5%) | 0.09±0.06 (-60.6%) | 0.02±0.03 (-95.5%) | 0.03±0.03 (-94.9%) | **0.14±0.04 (-80.9%)** |
| PushCube | 0.04±0.03 (-87.0%) | 0.04±0.03 (-91.0%) | 0.40±0.06 (-21.2%) | 0.05±0.02 (-86.7%) | 0.17±0.08 (-64.1%) | 0.06±0.03 (-86.3%) | **0.60±0.15 (+33.3%)** |
| PlaceAppleInBowl | 0.00±0.01 (-93.4%) | 0.01±0.01 (-92.7%) | 0.06±0.02 (-15.9%) | 0.02±0.01 (-84.4%) | 0.04±0.05 (-67.5%) | 0.01±0.02 (-95.5%) | **0.25±0.09 (-21.2%)** |
| TransportBox | 0.07±0.05 (+45.2%) | 0.07±0.10 (-73.6%) | 0.12±0.08 (-56.3%) | 0.17±0.08 (-37.8%) | 0.19±0.10 (-20.7%) | 0.12±0.06 (-55.8%) | **0.28±0.01 (-0.6%)** |

In both tables, **bold** values indicate the best IQM return for each task, and underlined values indicate the smallest performance drop relative to the model's unperturbed baseline. Complete results for all eight tasks, twelve perturbations and all difficulty levels (easy, medium, hard) are provided in Appendix A.6.

### 5.2 ROBUSTNESS TO IMAGE SEGMENTS VARIABILITY

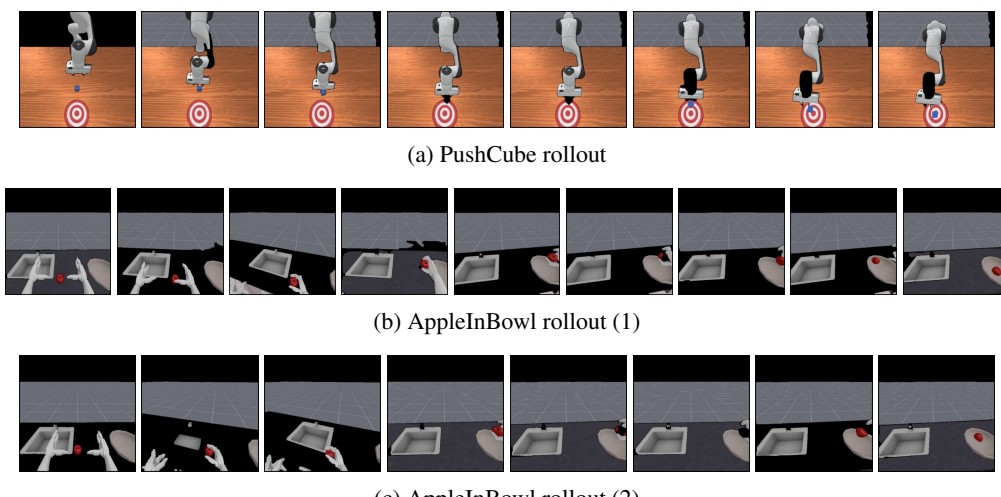

(a) PushCube rollout

(b) AppleInBowl rollout (1)

(c) AppleInBowl rollout (2)

Figure 10: Rollouts showing variability in detected segments.

Figure 10 shows examples where the set of detected segments changes over time, both in the number of segments and in their size and granularity. For visualization, all detected segments are stitched into a single frame, while SegDAC processes them as a sequence of individual segments. Black regions indicate frames where SAM failed to produce a mask.

Across several frames and trajectories, the background may appear late, task-critical segments like the cube or apple may disappear temporarily, and parts of the robot arm may merge or split into finer or coarser regions. Despite these changes, SegDAC

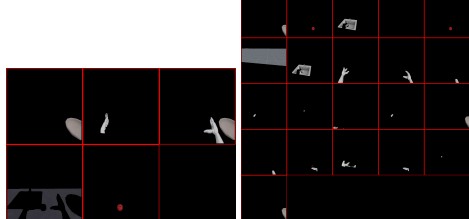

Figure 9: Variability in the number of segments detected by SAM at different time steps in the UnitreeG1PlaceAppleInBowl-v1 task (5 vs 21 segments).

maintains stable behaviour and continues to make progress on the task. This robustness is important for real-world applicability, since such variability naturally arises from lighting changes, viewpoint shifts, partial occlusions, and ambiguous boundaries.

**We believe SegDAC is the first model-free online visual RL approach relying on images to learn directly from a dynamic sequence of segment embeddings that vary across time, rather than reconstructing images from detected segments or relying on fixed pre-computed representations.** We hypothesize that this robustness mainly comes from two factors. First, the policy is trained on segment sets whose identity, number, size, and granularity change over time, which exposes it to natural variability in object-centric representations. Second, our Segment Embeddings Extraction Module produces embeddings that capture precise local information while also incorporating contextual information from SAM's self-attention. Together, these properties encourage the policy to use the contextual structure present across segments rather than depend on only a single segment. Additional examples, including temporary occlusions and finer variations in segment granularity and attention are shown in Appendix G.

## 5.3 SEGMENT ATTENTION ANALYSIS

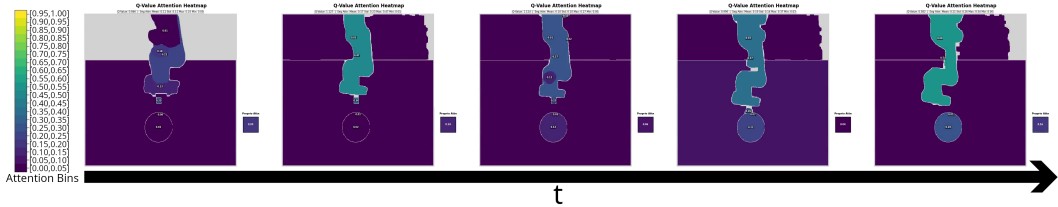

Figure 11: Critic attention on visual segments and proprioception during Q-value prediction.

Figure 11, along with extensive qualitative examples in Appendix G, demonstrates that SegDAC selectively attends to task-relevant objects while ignoring irrelevant segments such as the background or the table. Without history or frame stacking, the model adapts its attention over time, it focuses on the robot arm and cube at the start, then shifts toward the target after grasping the cube.

We observe that SegDAC does not maintain uniform attention on the robot throughout the trajectory. In some tasks, the model places more weight on proprioceptive features, while in others it emphasizes the robot arm. The goal is often ignored initially, especially when the first sub-task is to reach or grasp the manipulation object. Once the object is secured and needs to be moved, the model reallocates attention to the goal.

This behavior suggests that allowing the model to learn its own attention distribution leads to more adaptive and flexible decision-making compared to approaches that impose rigid or handcrafted attention patterns.

## 5.4 Sample Efficiency

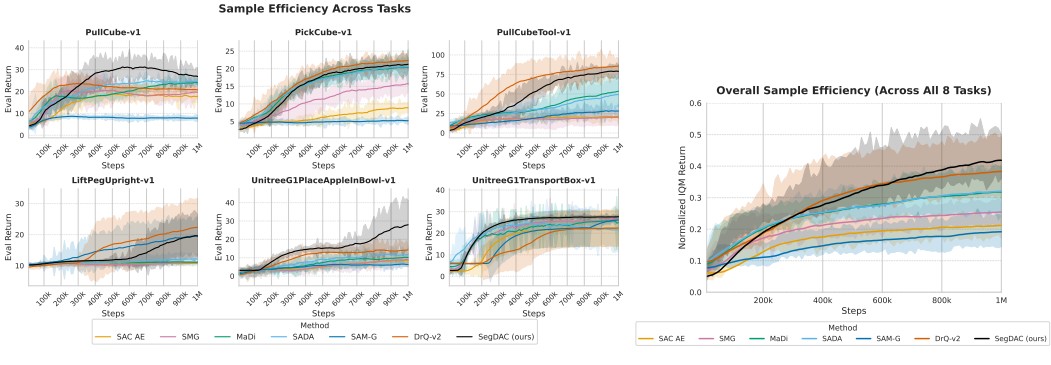

(a) Per-task sample efficiency.  (b) Aggregated IQM returns.

Figure 12: Sample efficiency curves, aggregated using 5 seeds.

To assess the sample efficiency of SegDAC, we compare it to the same baselines used in the visual generalization benchmark. Figure 12 shows that SegDAC matches the sample efficiency of the state-of-the-art DrQ-v2 across all tasks and outperforms it in three out of eight tasks. Moreover, SegDAC consistently achieves higher sample efficiency than the visual generalization baselines (MaDi, SADA, SMG, SAM-G), which often perform well on some tasks but collapse on others. While methods that excel in sample efficiency often struggle with visual generalization, and methods strong in visual generalization typically sacrifice efficiency, SegDAC challenges this trade-off and demonstrates that it is possible to achieve strong performance in both dimensions. Refer to the Appendix J for sample efficiency plots on additional tasks.

## 5.5 Case Study & Failure Cases

Figure 21 shows the Pick Cube task under the challenging hard table texture, which contains complex visual patterns along with a red cube and a green target blended into the table texture. This creates strong ambiguity, since the policy may confuse the textured regions with the actual cube. MaDi consistently fails in this setting and often produces unstable or extreme actions. In contrast, SegDAC succeeds on some rollouts (shown on left) and fails on others (shown on right), but the failures are more controlled. They usually come from missed grasp attempts rather than erratic behaviour. More examples are provided in Appendix A.5. We provide ablations in Appendix B.

## 6 Discussion

We introduced SegDAC, an actor-critic architecture that leverages pre-trained vision models and text-grounded segmentation to process a variable set of segments. Our results show that strong visual generalization is achievable without frame stacking, data augmentation, auxiliary losses, human labels, ground-truth masks, or reference frames, and that learning directly from segment embeddings without image reconstruction is both stable and effective. While we focus on short-horizon online RL, extending SegDAC to longer horizons and real-world settings is a natural direction. Overall, SegDAC offers a fresh direction for visual RL and improves visual generalization, sample efficiency, and robustness even when parts of the image are missing.

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

## A VISUAL GENERALIZATION BENCHMARK

### A.1 VISUAL PERTURBATION CATEGORIES

| Category | Perturbations |
|----------|---------------|
| Camera | camera pose, camera FOV |
| Lighting | lighting direction, lighting color |
| Color | MO color, RO color, table color, ground color |
| Texture | MO texture, RO texture, table texture, ground texture |

Table 5: Perturbation categories and their corresponding visual perturbation tests.

Table 5 lists the visual perturbation categories used in Figure 7.

### A.2 MODEL SELECTION

For all methods, we tested the final model (for each seed) after 1 million training steps. Early stopping had no significant effect on results, so using the final checkpoint ensured consistency and simplicity across tests.

### A.3 SCORES AGGREGATION

Scores are computed using normalized returns, where each return is divided by the maximum number of steps for the task, yielding values in $[0, 1]$. To compute benchmark returns, we first evaluate baseline performance on the unperturbed tasks by running 50 rollouts per task and per model seed. Each average return (over 50 rollouts) is computed for all $m = 8$ tasks and $n = 5$ seeds, yielding $m \times n = 40$ scores. We then compute the IQM and 95% confidence intervals using stratified bootstrap with 50,000 replications, following Agarwal et al. (2022).

For individual perturbation results (e.g., Table 7), we set $m = 1$ (a single task) and $n = 5$ (seeds), with each score averaged over 50 rollouts. For category-level scores (e.g., Camera Easy, Lighting Easy), $m$ is the total number of perturbation-task combinations in that category. For instance, both the Camera and Lighting categories include two perturbations (Pose/FOV and Direction/Color, respectively) applied to 8 tasks, resulting in $m = 2 \times 8 = 16$.

Some perturbation categories involve a receiving object (RO), such as RO Texture and RO Color in the Texture and Color categories. Tasks that do not include a RO (e.g., LiftPegUpright) cannot be evaluated on these specific perturbations, reducing $m$ by one for those tasks. As a result, the total number of perturbation-task pairs varies across categories depending on RO availability. For both Texture and Color, we obtain $m = 29$ after summing over all tasks while accounting for the presence or absence of ROs. A summary of the number of valid perturbation-task pairs per category, accounting for RO presence, is shown in Table 6.

For the overall scores, we aggregated results across all tasks and all perturbations for each difficulty level, yielding a total of $80 + 80 + 145 + 145 = 450$ test cases per difficulty. We then computed the IQM over these 450 results, providing a robust and reliable evaluation.

| Perturbation Category | # Tests ($m \times n$) |
|---|---|
| Camera | $16 \times 5 = 80$ |
| Lighting | $16 \times 5 = 80$ |
| Color | $29 \times 5 = 145$ |
| Texture | $29 \times 5 = 145$ |

Table 6: Number of test scores per visual perturbation category, **for each test** we perform **50 rollouts** to compute performance metrics.

### A.4 DIFFICULTY TEXTURES EXAMPLES

To illustrate the impact of benchmark difficulty on scene appearance, we include Blender renders of selected visual perturbations and full-scene previews for each task. Not all Blender renders are shown, the complete set is available at (anonymized for review). The textures and colors are also task-specific, for example, in the Pick Cube task, the default cube is red and the target is green. Accordingly, the texture perturbations vary in difficulty: the easy texture resembles metallic red (similar to the default), the medium texture is unrelated to either red or green, and the hard texture uses green dots resembling the target to induce confusion. Figure 13 shows how cube appearance

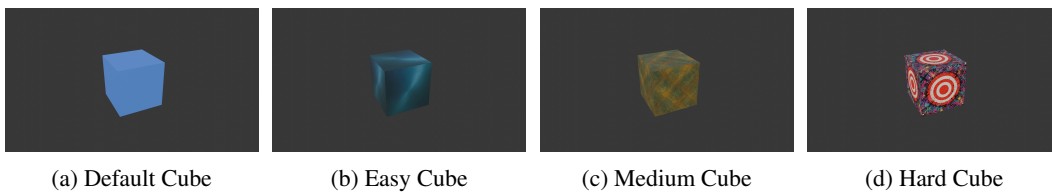

(a) Default Cube      (b) Easy Cube      (c) Medium Cube      (d) Hard Cube

Figure 13: Blender renders for the cube, used in tasks like push/pull/poke cube.

changes with perturbation difficulty. The default cube (used when no perturbations are applied) is a solid light blue. The easy texture remains blue with minor variations, such as darker shades and simple patterns, maintaining high visual similarity with the default cube. The medium texture introduces green and yellow-brown tones with slightly more complex patterns, it does not share a lot (if any) visual similarity with the default cube while avoiding conflicts with other scene elements (e.g: robot arm, table, or target). The hard texture is visually and semantically disruptive: it features a chaotic mix of colors and complex patterns, with each cube face displaying a target identical to the task's actual target. This creates a semantic clash, making it harder for the policy to recognize the cube. These examples demonstrate that our difficulty levels correspond to meaningful and progressively more challenging visual changes.

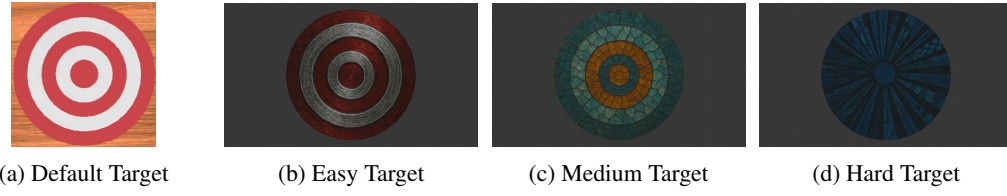

(a) Default Target      (b) Easy Target      (c) Medium Target      (d) Hard Target

Figure 14: Blender renders for the target.

Figure 14 illustrates the different target textures across difficulty levels. The easy texture closely resembles the default, with red and white nested rings and a slight metallic finish to introduce subtle variation. The medium texture uses entirely different colors and more complex patterns, but retains the same nested ring structure and number of rings. The hard texture is significantly different: it replaces the nested rings with inward-pointing stripes, resembling a dartboard. The color scheme

also changes drastically (e.g., blue instead of red), making the target visually and semantically harder to recognize.

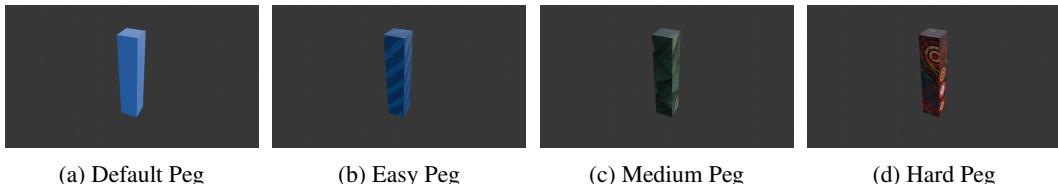

(a) Default Peg     (b) Easy Peg     (c) Medium Peg     (d) Hard Peg

Figure 15: Blender renders for the peg used in the poke cube task.

Figure 15 shows the peg textures across difficulty levels. While multiple tasks include a peg, their textures may differ, the renders shown here are for the poke cube task. The easy texture shares strong visual similarity with the default view, as both are blue. The medium texture uses green, a neutral color that does not resemble any other object in the scene, and features a more complex pattern. The hard texture is visually and semantically distinct, with no similarity to the default. It includes repeated target symbols similar to those used for the task, increasing ambiguity and making the object harder to identify correctly.

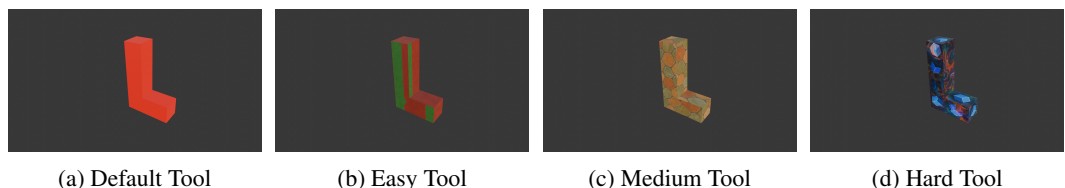

(a) Default Tool     (b) Easy Tool     (c) Medium Tool     (d) Hard Tool

Figure 16: Blender renders for the l-shaped tool used in the pull cube tool task.

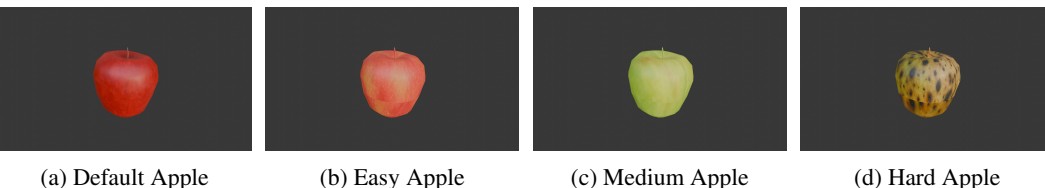

(a) Default Apple     (b) Easy Apple     (c) Medium Apple     (d) Hard Apple

Figure 17: Blender renders for the apple used in the place apple in the bowl task.

Figure 17 shows the apple textures across difficulty levels. The default apple is a simple red apple. The easy texture is a lighter red with faint orange areas, maintaining strong visual similarity. The medium texture is green, representing a common apple variety, but differing in color. The hard texture shows a brownish, rotten apple that is both visually distinct and semantically less typical, representing a rare case that may be underrepresented in real-world data or training distributions.

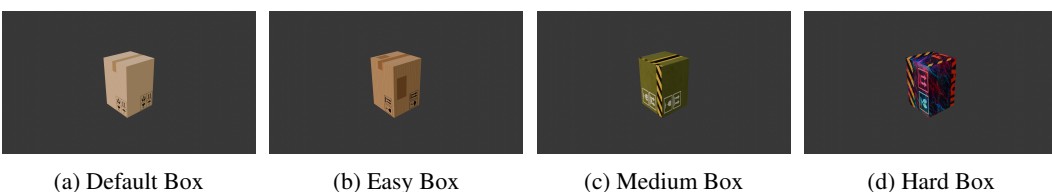

(a) Default Box     (b) Easy Box     (c) Medium Box     (d) Hard Box

Figure 18: Blender renders for the box used in the transport box task.

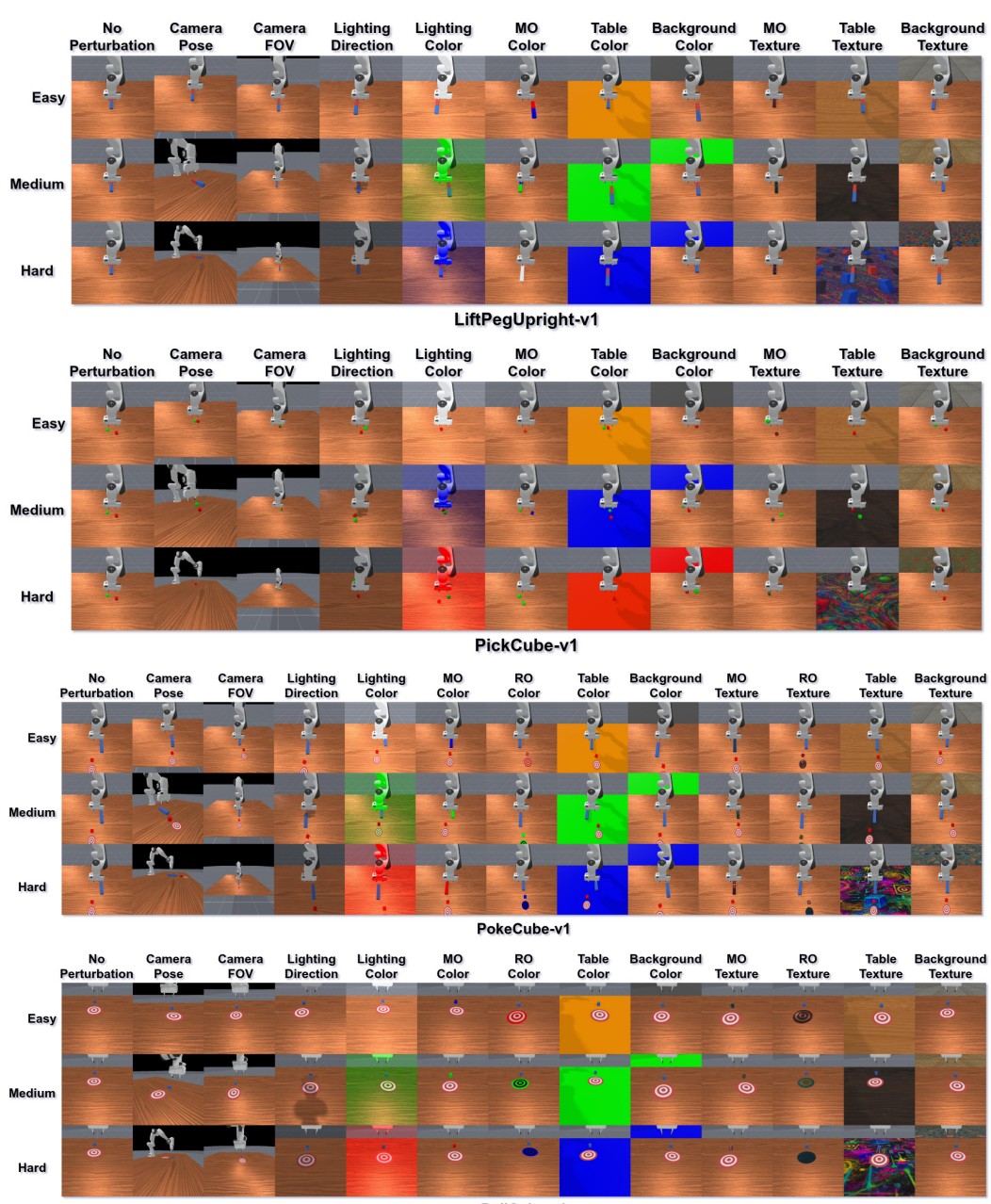

Figure 19: (Part 1) Visual perturbations for all tasks and difficulties.

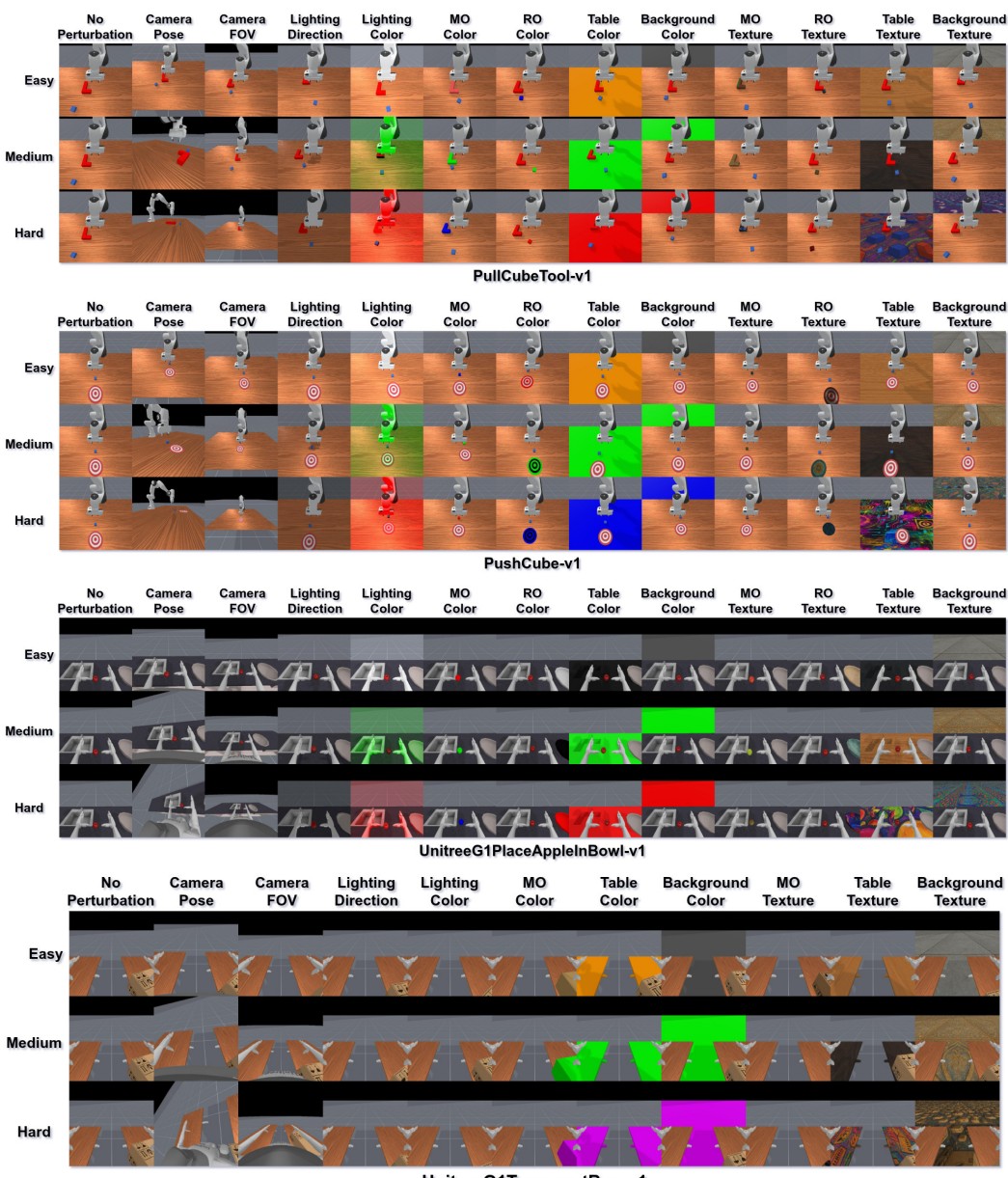

Figure 20: (Part 2) Visual perturbations for all tasks and difficulties.

Figures 19 and 20 show all visual perturbations for each task alongside their default (unperturbed) views. As difficulty increases, observations become more out-of-distribution and challenging. Notably, our camera pose perturbations are more aggressive than those typically used in other benchmarks, making them especially difficult for policies to handle. While no benchmark is perfect, we believe that exposing policies to strong perturbations provides a clearer understanding of their true robustness and generalization capabilities.

## A.5 CASE STUDY & FAILURE CASES

Below are qualitative rollout examples comparing SegDAC to MaDi across various tasks and perturbations. We also include cases with camera pose perturbations, as they were the most challenging for all methods.

**MaDi**

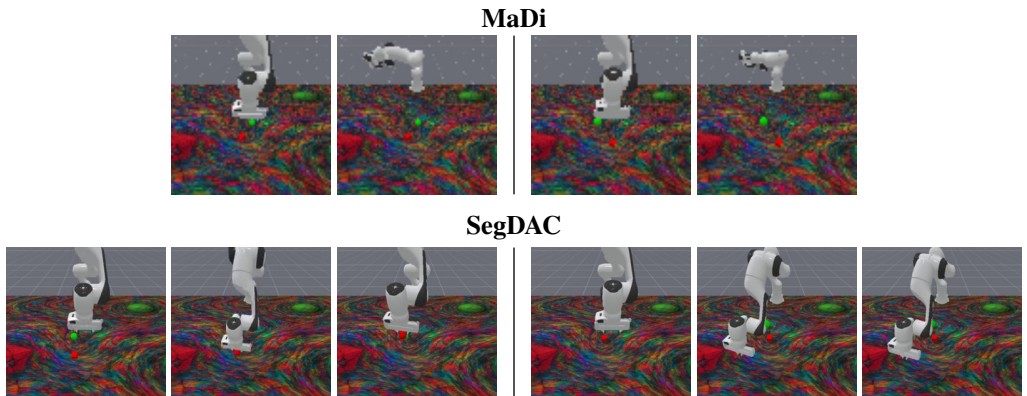

**SegDAC**

Figure 21: 2 Rollouts for *PickCube* - Hard Table Texture (top: MaDi, bottom: SegDAC).

**MaDi**

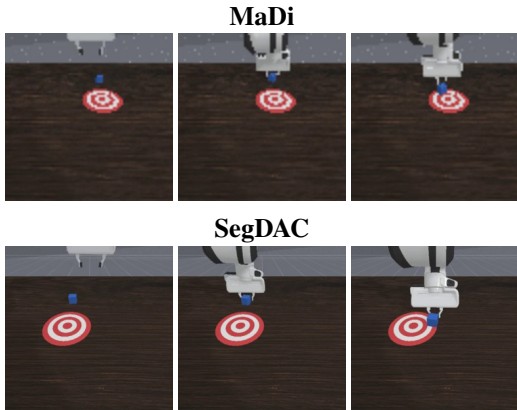

**SegDAC**

Figure 22: Rollout for *PullCube* - Medium Table Texture (top: MaDi, bottom: SegDAC).

Figure 22 shows that both MaDi and SegDAC are able to successfully complete the Pull Cube task under the medium table texture perturbation (success is reaching the first white ring), regardless of the target's initial position. This highlights that, in some cases, policies can still succeed under medium-level perturbations and do not always collapse when faced with moderate visual changes.

**MaDi**

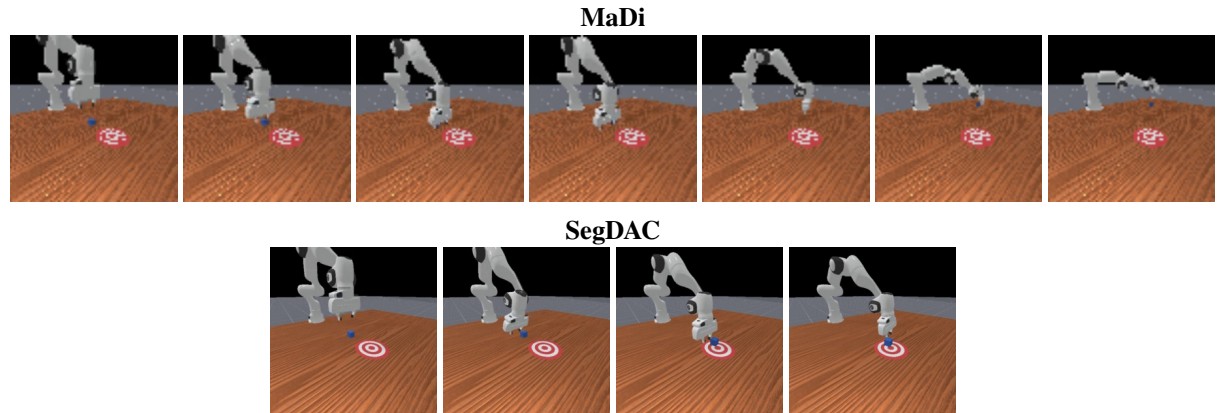

**SegDAC**

Figure 23: Rollout for *PushCube* - Medium Camera Pose (top: MaDi, bottom: SegDAC).

Figure 23 shows a rollout of the Push Cube task under the medium camera pose perturbation for both MaDi and SegDAC. MaDi initially behaves reasonably by reaching toward the cube, but fails to position the arm correctly. This leads to accumulating errors and eventually to unstable actions that push the cube away from the target. In contrast, SegDAC handles the perturbation more effectively and often succeeds, showing more stable and goal-directed behavior.

**MaDi**

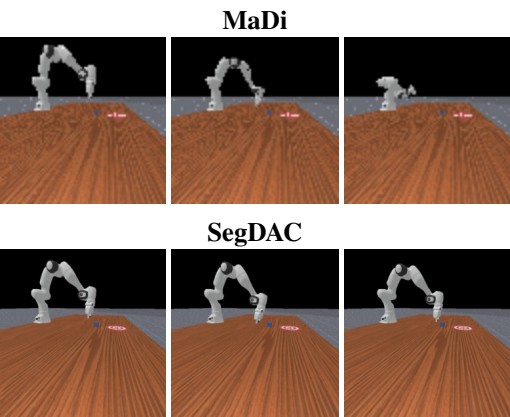

**SegDAC**

Figure 24: Rollout for *PushCube* - Hard Camera Pose (top: MaDi, bottom: SegDAC).

Figure 24 illustrates the difficulty of the hard camera pose perturbation for both policies. MaDi consistently collapses and produces erratic, non-sensical actions. While SegDAC does not succeed either, it shows more structured behavior, making small back-and-forth motions in an attempt to push the cube. However, it lacks the precision needed and repeatedly misses the cube by a small margin.

**MaDi**

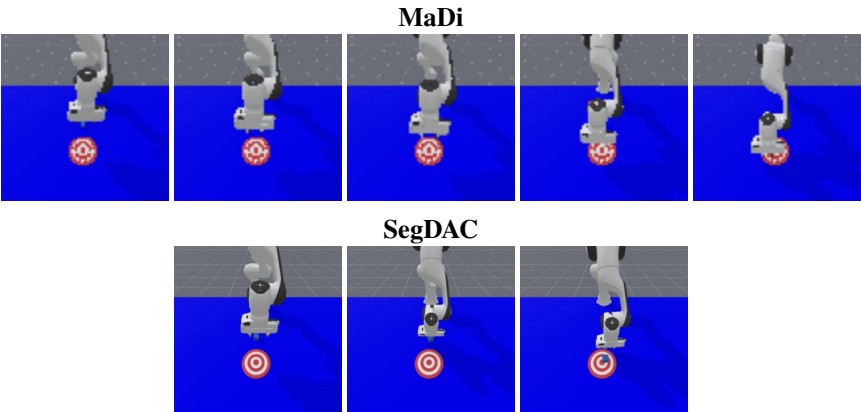

**SegDAC**

Figure 25: Rollout for *PushCube* - Hard Table Color (top: MaDi, bottom: SegDAC).

Figure 25 shows a rollout of the Push Cube task under the hard table color perturbation. In this case, the table is blue, the same color as the cube, which appears to confuse MaDi. The policy behaves as if the cube is already between the gripper and moves toward the target, while the actual cube remains at its initial position. This issue does not occur with SegDAC, which accurately identifies the cube and pushes it to the center of the target. This highlights the strength of object-centric approaches. While MaDi reasons at the pixel level, SegDAC decomposes the scene into segments and leverages bounding box coordinates of the cube segment to guide the policy more effectively.

## A.6 VISUAL GENERALIZATION PER PERTURBATION RESULTS

Table 7 and the subsequent tables report IQM returns and 95% confidence intervals for each visual perturbation, across all tasks, methods, and difficulty levels. The method with the highest absolute IQM return is shown in **bold**, while the method with the best relative improvement over its no-perturbation baseline is underlined. We also include the relative delta between the baseline (no perturbation) IQM return and the return under the perturbed setting. For example, a value of (-2.5%) indicates a 2.5% performance drop compared to the no perturbation case.

Table 7: Easy Camera Fov Test

| Task | SAC AE | DrQ-v2 | SAM-G | SMG | SADA | MaDi | SegDAC |
|---|---|---|---|---|---|---|---|
| LiftPegUpright | 0.20±0.00 (-7.8%) | 0.23±0.09 (-50.3%) | 0.25±0.02 (-40.6%) | 0.22±0.01 (-0.1%) | 0.22±0.01 (-1.1%) | 0.22±0.01 (-0.3%) | **0.27±0.08 (-34.4%)** |
| PickCube | 0.07±0.01 (-60.1%) | 0.12±0.04 (-74.0%) | 0.09±0.01 (-11.5%) | 0.17±0.02 (-47.1%) | **0.26±0.08 (-32.0%)** | 0.14±0.04 (-68.9%) | 0.21±0.02 (-36.5%) |
| PokeCube | 0.06±0.02 (-60.8%) | 0.15±0.03 (-66.2%) | 0.10±0.03 (-67.9%) | 0.12±0.04 (-44.3%) | 0.14±0.01 (-63.4%) | 0.11±0.01 (-72.9%) | **0.18±0.01 (-55.0%)** |
| PullCube | 0.09±0.03 (-76.5%) | 0.09±0.06 (-76.0%) | 0.14±0.01 (+12.4%) | 0.18±0.04 (-54.0%) | **0.42±0.14 (-13.6%)** | **0.42±0.17 (-15.6%)** | 0.32±0.12 (-35.7%) |
| PullCubeTool | 0.05±0.03 (-76.8%) | 0.12±0.02 (-80.6%) | 0.18±0.05 (-18.8%) | 0.22±0.09 (-5.6%) | 0.26±0.10 (-28.3%) | 0.21±0.03 (-65.9%) | **0.63±0.13 (-14.2%)** |
| PushCube | 0.12±0.03 (-59.0%) | 0.17±0.04 (-62.6%) | 0.15±0.02 (-71.0%) | 0.18±0.02 (-47.7%) | 0.24±0.04 (-47.5%) | 0.16±0.02 (-65.3%) | **0.59±0.10 (+30.3%)** |
| PlaceAppleInBowl | 0.04±0.01 (-52.5%) | 0.11±0.05 (-32.6%) | 0.04±0.03 (-38.5%) | 0.04±0.02 (-60.8%) | 0.10±0.04 (-14.2%) | 0.09±0.04 (-29.3%) | **0.21±0.11 (-32.8%)** |
| TransportBox | 0.03±0.02 (-31.8%) | 0.26±0.07 (-0.7%) | 0.26±0.02 (-3.9%) | 0.25±0.09 (-5.8%) | 0.22±0.10 (-9.7%) | 0.26±0.01 (-0.3%) | **0.27±0.01 (-2.5%)** |

Table 8: Easy Camera Pose Test

| Task | SAC AE | DrQ-v2 | SAM-G | SMG | SADA | MaDi | SegDAC |
|---|---|---|---|---|---|---|---|
| LiftPegUpright | 0.19±0.00 (-11.9%) | 0.23±0.09 (-50.5%) | **0.25±0.04 (-39.9%)** | 0.22±0.01 (-0.8%) | 0.22±0.04 (-1.9%) | 0.20±0.02 (-9.1%) | 0.19±0.00 (-54.0%) |
| PickCube | 0.05±0.02 (-74.9%) | 0.07±0.10 (-85.2%) | 0.08±0.01 (-22.7%) | 0.11±0.05 (-64.5%) | 0.25±0.06 (-34.5%) | **0.27±0.05 (-39.5%)** | 0.12±0.05 (-65.5%) |
| PokeCube | 0.03±0.01 (-83.8%) | 0.02±0.01 (-94.6%) | 0.12±0.01 (-62.4%) | 0.11±0.02 (-46.8%) | **0.22±0.04 (-42.1%)** | 0.16±0.01 (-60.2%) | 0.13±0.02 (-66.7%) |
| PullCube | 0.05±0.03 (-86.1%) | 0.02±0.02 (-94.2%) | 0.13±0.01 (+4.1%) | 0.14±0.04 (-62.9%) | 0.24±0.05 (-51.2%) | 0.24±0.02 (-51.2%) | **0.35±0.13 (-30.4%)** |
| PullCubeTool | 0.02±0.01 (-88.7%) | 0.15±0.07 (-76.4%) | 0.17±0.04 (-24.0%) | 0.17±0.06 (-28.4%) | 0.22±0.04 (-39.5%) | 0.20±0.05 (-67.6%) | **0.59±0.18 (-20.7%)** |
| PushCube | 0.06±0.05 (-80.1%) | 0.08±0.06 (-81.4%) | 0.34±0.06 (-32.7%) | 0.17±0.02 (-49.7%) | **0.36±0.04 (-21.5%)** | 0.30±0.09 (-36.0%) | 0.10±0.11 (-77.3%) |
| PlaceAppleInBowl | 0.03±0.02 (-59.1%) | 0.15±0.02 (-11.3%) | 0.05±0.02 (-18.7%) | 0.10±0.04 (-14.0%) | 0.12±0.04 (-2.4%) | 0.12±0.06 (-3.5%) | **0.22±0.08 (-29.7%)** |
| TransportBox | 0.04±0.04 (-12.6%) | 0.26±0.07 (-1.2%) | 0.26±0.01 (-2.2%) | 0.22±0.02 (-17.2%) | 0.22±0.10 (-10.5%) | 0.26±0.02 (-0.5%) | **0.28±0.01 (-1.3%)** |

Table 9: Easy Ground Color Test

| Task | SAC AE | DrQ-v2 | SAM-G | SMG | SADA | MaDi | SegDAC |
|---|---|---|---|---|---|---|---|
| LiftPegUpright | 0.21±0.00 (-0.7%) | 0.36±0.12 (-23.0%) | **0.43±0.15 (+2.5%)** | 0.22±0.00 (+0.4%) | 0.22±0.03 (+0.4%) | 0.23±0.01 (+2.5%) | 0.39±0.15 (-5.5%) |
| PickCube | 0.12±0.01 (-33.7%) | 0.13±0.09 (-72.5%) | 0.10±0.01 (-5.4%) | 0.33±0.07 (+4.8%) | 0.41±0.03 (+4.7%) | **0.43±0.06 (-2.4%)** | 0.35±0.04 (+3.1%) |
| PokeCube | 0.16±0.02 (-0.5%) | 0.20±0.12 (-53.3%) | 0.31±0.05 (-2.7%) | 0.21±0.14 (-0.8%) | 0.37±0.07 (-3.8%) | **0.41±0.01 (-0.1%)** | 0.34±0.06 (-12.1%) |
| PullCube | 0.23±0.04 (-38.7%) | 0.13±0.10 (-63.6%) | 0.13±0.01 (+2.6%) | 0.38±0.08 (-3.2%) | 0.45±0.18 (-8.5%) | 0.46±0.06 (-6.8%) | **0.51±0.06 (+1.4%)** |
| PullCubeTool | 0.19±0.05 (-12.2%) | 0.36±0.35 (-42.6%) | 0.23±0.07 (+5.6%) | 0.23±0.10 (+1.1%) | 0.36±0.14 (-1.8%) | 0.61±0.20 (-1.4%) | **0.69±0.14 (-7.3%)** |
| PushCube | 0.21±0.03 (-29.2%) | 0.32±0.16 (-28.9%) | **0.48±0.09 (-5.2%)** | 0.33±0.02 (-2.8%) | 0.46±0.02 (-0.1%) | 0.44±0.09 (-4.1%) | 0.44±0.04 (-1.8%) |
| PlaceAppleInBowl | 0.08±0.02 (+7.0%) | 0.15±0.02 (-8.5%) | 0.07±0.02 (+2.2%) | 0.08±0.04 (-24.0%) | 0.11±0.04 (-3.9%) | 0.12±0.05 (+1.0%) | **0.29±0.15 (-9.7%)** |
| TransportBox | 0.05±0.04 (+9.2%) | 0.24±0.08 (-5.7%) | 0.27±0.01 (+0.1%) | 0.26±0.02 (-1.4%) | 0.22±0.10 (-11.9%) | 0.26±0.00 (+0.1%) | **0.28±0.01 (-0.1%)** |

### Table 10: Easy Ground Texture Test

| Task | SAC AE | DrQ-v2 | SAM-G | SMG | SADA | MaDi | SegDAC |
|------|--------|--------|-------|-----|------|------|--------|
| LiftPegUpright | 0.21±0.01 (-1.3%) | **0.46±0.18 (-2.8%)** | 0.41±0.14 (-2.7%) | 0.22±0.00 (+0.3%) | 0.23±0.03 (+1.6%) | 0.22±0.01 (+1.6%) | 0.42±0.15 (+1.9%) |
| PickCube | 0.10±0.04 (-45.5%) | **0.44±0.03 (-4.8%)** | 0.11±0.01 (+6.4%) | 0.32±0.07 (+0.9%) | 0.40±0.03 (+3.8%) | 0.43±0.06 (-1.9%) | 0.35±0.05 (+5.2%) |
| PokeCube | 0.15±0.02 (-10.8%) | **0.43±0.06 (-0.4%)** | 0.32±0.06 (+1.4%) | 0.21±0.13 (+0.3%) | 0.37±0.07 (-3.9%) | 0.41±0.01 (-1.1%) | 0.40±0.05 (+0.8%) |
| PullCube | 0.25±0.03 (-32.4%) | 0.37±0.05 (-0.1%) | 0.12±0.01 (-1.0%) | 0.35±0.05 (-10.7%) | 0.42±0.15 (-13.2%) | 0.48±0.03 (-2.5%) | **0.53±0.06 (+4.9%)** |
| PullCubeTool | 0.19±0.06 (-15.4%) | 0.62±0.17 (-0.6%) | 0.27±0.08 (+20.3%) | 0.23±0.10 (-1.7%) | 0.34±0.17 (-8.3%) | 0.61±0.21 (-1.9%) | **0.74±0.16 (-0.6%)** |
| PushCube | 0.21±0.03 (-30.1%) | 0.34±0.22 (-24.6%) | **0.56±0.09 (+11.5%)** | 0.34±0.03 (-0.5%) | 0.45±0.01 (-1.8%) | 0.46±0.07 (+0.2%) | 0.46±0.04 (+2.9%) |
| PlaceAppleInBowl | 0.07±0.02 (-3.9%) | 0.05±0.04 (-66.8%) | 0.07±0.02 (+4.4%) | 0.11±0.04 (-1.9%) | 0.12±0.05 (-2.7%) | 0.12±0.00 (+0.2%) | **0.30±0.17 (-4.7%)** |
| TransportBox | 0.03±0.04 (-41.3%) | 0.26±0.07 (+1.1%) | 0.27±0.01 (-0.3%) | 0.18±0.12 (-32.8%) | 0.21±0.10 (-13.1%) | 0.26±0.00 (-0.0%) | **0.28±0.01 (-0.4%)** |

### Table 11: Easy Lighting Color Test

| Task | SAC AE | DrQ-v2 | SAM-G | SMG | SADA | MaDi | SegDAC |
|------|--------|--------|-------|-----|------|------|--------|
| LiftPegUpright | 0.21±0.01 (-0.6%) | 0.21±0.02 (-54.6%) | **0.42±0.15 (-0.0%)** | 0.22±0.01 (-0.5%) | 0.21±0.01 (-4.2%) | 0.22±0.02 (+2.3%) | 0.41±0.15 (+0.2%) |
| PickCube | 0.16±0.04 (-13.6%) | 0.07±0.04 (-84.8%) | 0.10±0.01 (-7.2%) | 0.33±0.05 (+5.4%) | 0.21±0.07 (-46.4%) | **0.43±0.06 (-3.0%)** | 0.24±0.05 (-28.9%) |
| PokeCube | 0.16±0.02 (-5.7%) | 0.10±0.05 (-76.9%) | 0.26±0.07 (-16.6%) | 0.22±0.12 (+0.8%) | 0.17±0.04 (-55.1%) | **0.41±0.01 (-0.8%)** | 0.36±0.03 (-7.8%) |
| PullCube | 0.30±0.04 (-19.7%) | 0.09±0.05 (-76.1%) | 0.13±0.02 (+0.3%) | 0.40±0.07 (+2.7%) | 0.44±0.14 (-9.0%) | 0.49±0.05 (-1.0%) | **0.51±0.05 (+0.8%)** |
| PullCubeTool | 0.24±0.07 (+8.2%) | 0.09±0.07 (-85.9%) | 0.27±0.06 (+22.4%) | 0.23±0.11 (+0.4%) | 0.19±0.10 (-47.4%) | 0.61±0.19 (-2.2%) | **0.72±0.12 (-2.9%)** |
| PushCube | 0.26±0.02 (-12.8%) | 0.16±0.06 (-64.5%) | **0.49±0.13 (-3.3%)** | 0.34±0.03 (-1.7%) | 0.37±0.11 (-18.8%) | 0.44±0.07 (-4.7%) | 0.48±0.02 (+5.8%) |
| PlaceAppleInBowl | 0.08±0.03 (+12.3%) | 0.04±0.06 (-75.6%) | 0.07±0.02 (+2.4%) | 0.12±0.04 (+11.1%) | 0.11±0.06 (-9.9%) | 0.12±0.04 (-6.0%) | **0.32±0.17 (+0.0%)** |
| TransportBox | 0.05±0.03 (+3.3%) | 0.26±0.07 (-0.6%) | 0.27±0.01 (+0.2%) | 0.27±0.01 (+0.4%) | 0.24±0.09 (-0.7%) | 0.26±0.01 (+0.1%) | **0.28±0.01 (+0.0%)** |

### Table 12: Easy Lighting Direction Test

| Task | SAC AE | DrQ-v2 | SAM-G | SMG | SADA | MaDi | SegDAC |
|------|--------|--------|-------|-----|------|------|--------|
| LiftPegUpright | 0.21±0.01 (-0.2%) | **0.46±0.18 (-1.7%)** | 0.42±0.14 (-0.7%) | 0.22±0.00 (-0.4%) | 0.22±0.03 (+0.9%) | 0.23±0.02 (+2.5%) | 0.40±0.15 (-2.6%) |
| PickCube | 0.17±0.03 (-8.8%) | **0.42±0.04 (-8.4%)** | 0.10±0.01 (-7.6%) | 0.32±0.05 (+2.0%) | 0.38±0.04 (-1.0%) | 0.41±0.05 (-6.0%) | 0.33±0.04 (-2.9%) |
| PokeCube | 0.16±0.03 (-1.8%) | 0.39±0.04 (-10.5%) | 0.29±0.09 (-8.5%) | 0.21±0.13 (+0.0%) | 0.38±0.08 (-1.9%) | **0.41±0.02 (+0.8%)** | 0.40±0.03 (+2.9%) |
| PullCube | 0.17±0.01 (-54.7%) | 0.26±0.05 (-30.3%) | 0.13±0.01 (+2.5%) | 0.38±0.09 (-1.6%) | **0.55±0.24 (+12.0%)** | 0.54±0.07 (+9.6%) | 0.52±0.09 (+4.0%) |
| PullCubeTool | 0.19±0.06 (-11.0%) | 0.37±0.22 (-40.4%) | 0.22±0.08 (-0.2%) | 0.22±0.10 (-5.2%) | 0.33±0.14 (-9.4%) | 0.62±0.21 (-1.2%) | **0.71±0.12 (-4.6%)** |
| PushCube | 0.27±0.03 (-7.7%) | 0.46±0.12 (-0.1%) | **0.49±0.11 (-2.8%)** | 0.35±0.01 (+1.4%) | 0.48±0.02 (+4.7%) | 0.43±0.07 (-6.9%) | 0.45±0.04 (+0.0%) |
| PlaceAppleInBowl | 0.08±0.02 (+4.2%) | 0.14±0.03 (-12.0%) | 0.07±0.02 (+8.5%) | 0.11±0.05 (+0.6%) | 0.12±0.04 (+0.2%) | 0.11±0.05 (-8.2%) | **0.32±0.17 (+1.7%)** |
| TransportBox | 0.05±0.03 (+0.8%) | 0.26±0.07 (-0.5%) | 0.27±0.01 (+0.1%) | 0.27±0.01 (+0.3%) | 0.24±0.09 (-0.3%) | 0.26±0.01 (+0.1%) | **0.28±0.01 (-0.1%)** |

### Table 13: Easy Mo Color Test

| Task | SAC AE | DrQ-v2 | SAM-G | SMG | SADA | MaDi | SegDAC |
|------|--------|--------|-------|-----|------|------|--------|
| LiftPegUpright | 0.21±0.01 (-1.3%) | 0.27±0.09 (-42.1%) | 0.37±0.13 (-12.4%) | 0.22±0.00 (-0.5%) | 0.22±0.02 (-0.8%) | 0.22±0.01 (-0.1%) | **0.40±0.15 (-2.5%)** |
| PickCube | 0.14±0.03 (-22.0%) | 0.25±0.06 (-45.0%) | 0.11±0.01 (+2.5%) | 0.25±0.09 (-19.8%) | 0.31±0.06 (-20.1%) | **0.43±0.06 (-2.9%)** | 0.28±0.06 (-17.4%) |
| PokeCube | 0.13±0.02 (-24.1%) | 0.17±0.01 (-60.0%) | 0.16±0.02 (-48.7%) | 0.19±0.12 (-9.3%) | 0.30±0.05 (-22.1%) | 0.39±0.01 (-4.4%) | **0.40±0.03 (+2.2%)** |
| PullCube | 0.24±0.04 (-34.2%) | 0.22±0.02 (-39.7%) | 0.12±0.01 (-4.7%) | 0.36±0.08 (-7.4%) | 0.34±0.12 (-31.3%) | 0.45±0.02 (-7.6%) | **0.48±0.08 (-3.7%)** |
| PullCubeTool | 0.14±0.02 (-38.2%) | 0.26±0.16 (-58.9%) | 0.24±0.09 (+6.2%) | 0.17±0.07 (-24.7%) | 0.31±0.13 (-14.2%) | 0.50±0.15 (-19.1%) | **0.81±0.17 (+9.8%)** |
| PushCube | 0.20±0.03 (-32.0%) | 0.23±0.16 (-48.7%) | 0.30±0.07 (-40.8%) | 0.26±0.04 (-22.6%) | 0.33±0.13 (-29.1%) | 0.43±0.07 (-6.4%) | **0.45±0.05 (-0.2%)** |
| PlaceAppleInBowl | 0.07±0.01 (-7.7%) | 0.08±0.04 (-53.4%) | 0.06±0.03 (-6.6%) | 0.10±0.04 (-7.7%) | 0.09±0.04 (-21.7%) | 0.13±0.04 (+4.5%) | **0.32±0.17 (+1.0%)** |
| TransportBox | 0.05±0.03 (-1.0%) | 0.26±0.07 (-0.6%) | 0.27±0.01 (-0.5%) | 0.27±0.01 (+0.5%) | 0.24±0.09 (-2.2%) | 0.26±0.01 (-0.0%) | **0.28±0.01 (-0.4%)** |

### Table 14: Easy Mo Texture Test

| Task | SAC AE | DrQ-v2 | SAM-G | SMG | SADA | MaDi | SegDAC |
|------|--------|--------|-------|-----|------|------|--------|
| LiftPegUpright | 0.20±0.01 (-5.7%) | 0.22±0.02 (-52.8%) | **0.31±0.06 (-27.0%)** | 0.22±0.01 (+0.5%) | 0.22±0.02 (-2.6%) | 0.20±0.01 (-6.8%) | **0.31±0.09 (-23.8%)** |
| PickCube | 0.14±0.01 (-22.9%) | **0.24±0.05 (-48.0%)** | 0.11±0.01 (+2.9%) | 0.06±0.07 (-82.4%) | 0.17±0.05 (-57.3%) | 0.23±0.08 (-46.6%) | 0.16±0.02 (-50.9%) |
| PokeCube | 0.12±0.01 (-25.2%) | 0.17±0.04 (-61.8%) | 0.17±0.03 (-46.2%) | 0.22±0.13 (+4.9%) | **0.38±0.03 (-8.1%)** | 0.23±0.06 (-8.2%) | 0.36±0.03 (-8.2%) |
| PullCube | 0.25±0.03 (-32.6%) | 0.27±0.05 (-27.1%) | 0.13±0.01 (+1.3%) | 0.35±0.05 (-10.9%) | 0.37±0.02 (-24.3%) | **0.47±0.05 (-5.1%)** | 0.46±0.10 (-8.5%) |
| PullCubeTool | 0.13±0.05 (-42.0%) | 0.12±0.07 (-80.5%) | 0.23±0.10 (+3.7%) | 0.17±0.06 (-27.5%) | 0.11±0.14 (-68.7%) | 0.23±0.08 (-63.7%) | **0.35±0.08 (-53.1%)** |
| PushCube | 0.18±0.02 (-38.9%) | 0.18±0.10 (-59.5%) | 0.32±0.10 (-36.8%) | 0.28±0.04 (-19.0%) | 0.19±0.04 (-57.9%) | 0.22±0.16 (-51.9%) | **0.48±0.08 (+7.3%)** |
| PlaceAppleInBowl | 0.08±0.02 (+8.5%) | 0.15±0.02 (-7.6%) | 0.06±0.02 (-11.2%) | 0.12±0.05 (+7.2%) | 0.12±0.04 (-1.2%) | 0.12±0.05 (-5.3%) | **0.29±0.17 (-8.2%)** |
| TransportBox | 0.05±0.03 (+4.5%) | 0.26±0.07 (-1.1%) | 0.27±0.01 (+0.0%) | 0.27±0.01 (+0.1%) | 0.24±0.09 (-3.5%) | 0.26±0.01 (+0.1%) | **0.28±0.01 (+0.0%)** |

### Table 15: Easy Ro Color Test

| Task | SAC AE | DrQ-v2 | SAM-G | SMG | SADA | MaDi | SegDAC |
|------|--------|--------|-------|-----|------|------|--------|
| LiftPegUpright | N/A | N/A | N/A | N/A | N/A | N/A | N/A |
| PickCube | N/A | N/A | N/A | N/A | N/A | N/A | N/A |
| PokeCube | 0.17±0.03 (+0.8%) | **0.42±0.05 (-3.7%)** | 0.32±0.13 (+0.9%) | 0.22±0.01 (+3.5%) | 0.38±0.05 (-1.2%) | 0.40±0.01 (-2.3%) | 0.40±0.02 (+1.8%) |
| PullCube | 0.34±0.06 (-8.7%) | 0.29±0.08 (-20.3%) | 0.12±0.01 (-5.9%) | 0.39±0.08 (+1.0%) | 0.41±0.11 (-16.6%) | 0.50±0.05 (+1.1%) | **0.51±0.09 (+1.0%)** |
| PullCubeTool | 0.18±0.06 (-15.4%) | 0.48±0.21 (-22.4%) | 0.22±0.09 (-2.9%) | 0.23±0.10 (-1.9%) | 0.30±0.16 (-18.6%) | 0.60±0.18 (-3.5%) | **0.70±0.14 (-5.9%)** |
| PushCube | 0.26±0.03 (-13.1%) | 0.23±0.22 (-48.8%) | **0.48±0.09 (-4.5%)** | 0.33±0.02 (-2.3%) | 0.43±0.04 (-6.8%) | 0.43±0.07 (-6.3%) | 0.44±0.04 (-2.9%) |
| PlaceAppleInBowl | 0.08±0.03 (+6.9%) | 0.11±0.02 (-32.4%) | 0.07±0.02 (+0.8%) | 0.10±0.05 (-7.5%) | 0.12±0.04 (+0.2%) | 0.12±0.05 (-0.7%) | **0.29±0.14 (-7.0%)** |
| TransportBox | N/A | N/A | N/A | N/A | N/A | N/A | N/A |

### Table 16: Easy Ro Texture Test

| Task | SAC AE | DrQ-v2 | SAM-G | SMG | SADA | MaDi | SegDAC |
|------|--------|--------|-------|-----|------|------|--------|
| LiftPegUpright | N/A | N/A | N/A | N/A | N/A | N/A | N/A |
| PickCube | N/A | N/A | N/A | N/A | N/A | N/A | N/A |
| PokeCube | 0.17±0.04 (+0.9%) | 0.35±0.04 (-19.5%) | 0.31±0.08 (-2.3%) | 0.22±0.13 (+5.3%) | 0.38±0.07 (-2.1%) | **0.41±0.01 (-0.0%)** | 0.40±0.04 (+2.0%) |
| PullCube | 0.35±0.03 (-6.0%) | 0.14±0.12 (-63.0%) | 0.13±0.02 (+2.1%) | 0.38±0.11 (-3.5%) | **0.70±0.05 (+43.4%)** | 0.58±0.07 (+17.4%) | 0.55±0.07 (+9.1%) |
| PullCubeTool | 0.20±0.07 (-6.8%) | 0.57±0.26 (-8.2%) | 0.24±0.07 (+7.2%) | 0.23±0.10 (-1.1%) | 0.35±0.13 (-3.6%) | 0.63±0.20 (+1.2%) | **0.72±0.12 (-2.3%)** |
| PushCube | 0.24±0.03 (-19.4%) | 0.33±0.23 (-28.4%) | 0.47±0.14 (-6.9%) | 0.33±0.02 (-3.8%) | **0.57±0.07 (+22.9%)** | 0.56±0.13 (+19.8%) | 0.49±0.05 (+10.0%) |
| PlaceAppleInBowl | 0.06±0.02 (-20.5%) | 0.06±0.04 (-63.9%) | 0.06±0.01 (-2.8%) | 0.10±0.05 (-6.5%) | 0.11±0.04 (-3.2%) | 0.12±0.05 (-3.2%) | **0.23±0.12 (-25.7%)** |
| TransportBox | N/A | N/A | N/A | N/A | N/A | N/A | N/A |

### Table 17: Easy Table Color Test

| Task | SAC AE | DrQ-v2 | SAM-G | SMG | SADA | MaDi | SegDAC |
|---|---|---|---|---|---|---|---|
| LiftPegUpright | 0.21±0.00 (-2.3%) | 0.21±0.02 (-55.3%) | **0.38±0.12 (-9.2%)** | 0.22±0.01 (-0.6%) | 0.22±0.03 (-1.3%) | 0.21±0.00 (-6.2%) | 0.28±0.08 (-32.2%) |
| PickCube | 0.06±0.05 (-68.1%) | 0.05±0.11 (-88.7%) | 0.11±0.02 (+0.6%) | 0.21±0.05 (-31.7%) | 0.26±0.07 (-32.7%) | **0.43±0.06 (-2.1%)** | 0.19±0.04 (-43.0%) |
| PokeCube | 0.14±0.02 (-16.5%) | 0.30±0.14 (-32.1%) | 0.31±0.06 (-1.5%) | 0.19±0.10 (-9.2%) | 0.28±0.07 (-28.4%) | **0.40±0.01 (-3.5%)** | 0.24±0.02 (-38.6%) |
| PullCube | 0.14±0.07 (-63.2%) | 0.30±0.22 (-18.0%) | 0.13±0.02 (+2.3%) | 0.25±0.10 (-35.8%) | **0.42±0.03 (-13.3%)** | 0.39±0.14 (-21.6%) | 0.42±0.05 (-15.7%) |
| PullCubeTool | 0.19±0.06 (-14.6%) | **0.61±0.17 (-1.4%)** | 0.28±0.09 (+24.9%) | 0.14±0.07 (-40.8%) | 0.35±0.18 (-5.1%) | 0.43±0.28 (-31.6%) | 0.48±0.12 (-35.4%) |
| PushCube | 0.24±0.06 (-19.2%) | 0.39±0.25 (-14.8%) | 0.49±0.08 (-3.1%) | 0.32±0.01 (-7.8%) | 0.45±0.05 (-3.4%) | 0.45±0.08 (-2.2%) | **0.66±0.08 (+46.3%)** |
| PlaceAppleInBowl | 0.06±0.04 (-13.1%) | 0.09±0.04 (-44.1%) | 0.05±0.03 (-19.7%) | 0.06±0.02 (-42.0%) | 0.10±0.05 (-15.9%) | 0.12±0.06 (-6.3%) | **0.30±0.16 (-5.0%)** |
| TransportBox | 0.06±0.03 (+17.4%) | 0.19±0.11 (-25.8%) | 0.26±0.01 (-1.5%) | 0.27±0.01 (+0.4%) | 0.24±0.09 (-3.7%) | 0.26±0.00 (-0.2%) | **0.28±0.01 (-0.5%)** |

### Table 18: Easy Table Texture Test

| Task | SAC AE | DrQ-v2 | SAM-G | SMG | SADA | MaDi | SegDAC |
|---|---|---|---|---|---|---|---|
| LiftPegUpright | 0.21±0.00 (+0.2%) | 0.40±0.19 (-15.7%) | **0.41±0.14 (-1.7%)** | 0.22±0.01 (-0.3%) | 0.22±0.05 (+0.6%) | 0.23±0.01 (+2.5%) | 0.39±0.13 (-5.4%) |
| PickCube | 0.16±0.07 (-13.8%) | 0.33±0.15 (-27.8%) | 0.10±0.00 (-4.7%) | 0.29±0.07 (-6.6%) | 0.35±0.07 (-9.3%) | **0.42±0.05 (-5.1%)** | 0.34±0.03 (+1.5%) |
| PokeCube | 0.16±0.01 (-5.3%) | 0.40±0.05 (-6.9%) | 0.32±0.08 (+1.5%) | 0.22±0.12 (+1.0%) | 0.36±0.08 (-6.9%) | **0.41±0.02 (+0.1%)** | 0.38±0.01 (-2.7%) |
| PullCube | 0.33±0.04 (-10.9%) | 0.37±0.03 (+1.1%) | 0.13±0.01 (+4.3%) | 0.41±0.13 (+5.3%) | **0.50±0.12 (+2.3%)** | 0.49±0.05 (+0.3%) | 0.50±0.08 (-0.8%) |
| PullCubeTool | 0.22±0.07 (-1.4%) | 0.41±0.24 (-33.8%) | 0.26±0.07 (+17.5%) | 0.23±0.10 (-2.4%) | 0.29±0.08 (-20.8%) | 0.60±0.20 (-4.0%) | **0.61±0.18 (-17.2%)** |
| PushCube | 0.26±0.05 (-11.7%) | 0.45±0.23 (-2.0%) | **0.52±0.06 (+2.8%)** | 0.33±0.01 (-2.9%) | 0.45±0.09 (-3.2%) | 0.45±0.06 (-2.6%) | 0.48±0.06 (+6.9%) |
| PlaceAppleInBowl | 0.07±0.02 (-0.8%) | 0.13±0.02 (-22.5%) | 0.06±0.03 (-12.2%) | 0.07±0.03 (-35.5%) | 0.10±0.04 (-14.3%) | 0.11±0.07 (-7.6%) | **0.28±0.16 (-10.0%)** |
| TransportBox | 0.06±0.04 (+27.8%) | 0.24±0.07 (-6.9%) | 0.27±0.01 (+0.0%) | 0.27±0.01 (+0.0%) | 0.24±0.09 (-1.3%) | 0.26±0.00 (-0.8%) | **0.28±0.01 (-0.3%)** |

### Table 19: Medium Camera Fov Test

| Task | SAC AE | DrQ-v2 | SAM-G | SMG | SADA | MaDi | SegDAC |
|---|---|---|---|---|---|---|---|
| LiftPegUpright | 0.19±0.00 (-11.7%) | 0.19±0.01 (-59.3%) | 0.20±0.01 (-53.6%) | **0.22±0.01 (-1.7%)** | 0.21±0.01 (-3.6%) | 0.21±0.00 (-5.5%) | 0.22±0.00 (-47.6%) |
| PickCube | 0.02±0.02 (-88.9%) | 0.02±0.01 (-96.7%) | 0.03±0.01 (-74.9%) | 0.04±0.02 (-86.4%) | **0.11±0.02 (-71.7%)** | 0.08±0.02 (-82.8%) | 0.06±0.02 (-81.7%) |
| PokeCube | 0.01±0.01 (-92.3%) | 0.02±0.01 (-95.8%) | 0.05±0.03 (-83.7%) | 0.08±0.03 (-64.5%) | **0.11±0.02 (-71.6%)** | 0.09±0.02 (-78.7%) | 0.09±0.01 (-75.8%) |
| PullCube | 0.05±0.02 (-86.9%) | 0.03±0.02 (-92.8%) | 0.14±0.02 (+8.0%) | 0.09±0.03 (-76.2%) | **0.19±0.03 (-60.1%)** | 0.15±0.03 (-69.0%) | 0.18±0.02 (-64.9%) |
| PullCubeTool | 0.02±0.01 (-90.0%) | 0.03±0.05 (-95.2%) | 0.06±0.02 (-72.4%) | 0.09±0.06 (-60.4%) | 0.15±0.05 (-58.7%) | 0.10±0.02 (-83.2%) | **0.30±0.10 (-59.3%)** |
| PushCube | 0.02±0.02 (-91.7%) | 0.02±0.01 (-96.4%) | 0.04±0.00 (-92.9%) | 0.05±0.01 (-85.7%) | **0.16±0.03 (-65.4%)** | 0.06±0.02 (-87.5%) | 0.08±0.03 (-82.6%) |
| PlaceAppleInBowl | 0.01±0.01 (-80.9%) | **0.03±0.01 (-82.1%)** | 0.03±0.01 (-58.1%) | 0.03±0.01 (-72.1%) | 0.03±0.04 (-71.5%) | 0.03±0.02 (-77.5%) | 0.03±0.04 (-89.6%) |
| TransportBox | 0.03±0.03 (-32.9%) | 0.18±0.07 (-31.8%) | 0.13±0.05 (-51.4%) | 0.16±0.06 (-41.4%) | 0.16±0.08 (-33.6%) | 0.20±0.06 (-21.8%) | **0.27±0.01 (-1.8%)** |

### Table 20: Medium Camera Pose Test

| Task | SAC AE | DrQ-v2 | SAM-G | SMG | SADA | MaDi | SegDAC |
|---|---|---|---|---|---|---|---|
| LiftPegUpright | 0.19±0.01 (-10.9%) | 0.19±0.01 (-59.0%) | 0.19±0.00 (-54.2%) | 0.21±0.02 (-3.8%) | 0.19±0.01 (-13.3%) | 0.20±0.02 (-10.1%) | **0.23±0.01 (-45.3%)** |
| PickCube | 0.03±0.01 (-84.4%) | 0.03±0.01 (-94.5%) | 0.04±0.03 (-65.8%) | 0.06±0.02 (-80.0%) | 0.11±0.02 (-71.8%) | 0.07±0.02 (-84.8%) | **0.12±0.04 (-64.7%)** |
| PokeCube | 0.02±0.02 (-86.4%) | 0.02±0.03 (-95.9%) | 0.02±0.03 (-79.6%) | 0.07±0.02 (-67.5%) | **0.11±0.03 (-72.8%)** | 0.08±0.02 (-79.6%) | 0.10±0.01 (-74.6%) |
| PullCube | 0.04±0.01 (-90.5%) | 0.02±0.02 (-95.9%) | 0.12±0.02 (-7.6%) | 0.08±0.04 (-78.8%) | **0.16±0.04 (-66.9%)** | 0.06±0.03 (-87.5%) | 0.07±0.02 (-85.4%) |
| PullCubeTool | 0.02±0.02 (-91.1%) | 0.01±0.01 (-99.1%) | **0.05±0.01 (-77.9%)** | 0.04±0.04 (-82.7%) | 0.03±0.01 (-93.1%) | 0.01±0.01 (-98.8%) | 0.01±0.01 (-98.4%) |
| PushCube | 0.04±0.01 (-85.6%) | 0.02±0.01 (-94.8%) | 0.08±0.01 (-84.9%) | 0.08±0.02 (-76.4%) | 0.10±0.01 (-78.8%) | 0.10±0.01 (-78.6%) | **0.34±0.06 (-24.7%)** |
| PlaceAppleInBowl | 0.01±0.01 (-86.5%) | 0.05±0.03 (-70.6%) | 0.02±0.01 (-71.3%) | 0.02±0.01 (-81.4%) | **0.08±0.06 (-34.5%)** | 0.03±0.02 (-77.6%) | 0.05±0.08 (-84.3%) |
| TransportBox | 0.04±0.03 (-29.4%) | 0.06±0.02 (-75.9%) | 0.07±0.03 (-75.6%) | 0.04±0.04 (-83.5%) | 0.14±0.08 (-44.7%) | 0.10±0.06 (-61.8%) | **0.27±0.02 (-3.2%)** |

### Table 21: Medium Ground Color Test

| Task | SAC AE | DrQ-v2 | SAM-G | SMG | SADA | MaDi | SegDAC |
|---|---|---|---|---|---|---|---|
| LiftPegUpright | 0.21±0.00 (-1.1%) | 0.21±0.02 (-55.3%) | 0.38±0.15 (-10.4%) | 0.22±0.00 (+0.3%) | 0.21±0.04 (-3.0%) | 0.22±0.02 (-0.9%) | **0.39±0.16 (-5.1%)** |
| PickCube | 0.04±0.02 (-80.1%) | 0.02±0.01 (-96.7%) | 0.11±0.01 (+0.6%) | 0.27±0.04 (-13.4%) | 0.35±0.04 (-8.4%) | **0.43±0.06 (-1.5%)** | 0.33±0.05 (-2.2%) |
| PokeCube | 0.12±0.05 (-28.7%) | 0.09±0.07 (-79.9%) | 0.29±0.08 (-7.7%) | 0.21±0.13 (-3.7%) | 0.32±0.08 (-18.0%) | **0.41±0.02 (-0.3%)** | 0.38±0.03 (-4.3%) |
| PullCube | 0.18±0.05 (-50.9%) | 0.02±0.08 (-95.7%) | 0.13±0.01 (+1.9%) | 0.20±0.11 (-49.6%) | 0.33±0.10 (-31.7%) | 0.34±0.14 (-30.1%) | **0.55±0.06 (+10.5%)** |
| PullCubeTool | 0.16±0.08 (-25.4%) | 0.09±0.12 (-85.4%) | 0.26±0.06 (+19.2%) | 0.24±0.10 (+1.9%) | 0.29±0.07 (-20.5%) | 0.47±0.16 (-24.6%) | **0.59±0.12 (-20.8%)** |
| PushCube | 0.23±0.02 (-23.4%) | 0.03±0.05 (-92.4%) | **0.52±0.09 (+3.7%)** | 0.33±0.04 (-4.6%) | 0.38±0.11 (-17.0%) | 0.43±0.09 (-7.3%) | 0.47±0.03 (+4.0%) |
| PlaceAppleInBowl | 0.08±0.03 (+1.1%) | 0.01±0.04 (-95.9%) | 0.08±0.02 (+16.2%) | 0.05±0.05 (-55.4%) | 0.10±0.04 (-15.0%) | 0.10±0.04 (-21.2%) | **0.30±0.16 (-4.4%)** |
| TransportBox | 0.04±0.02 (-20.3%) | 0.05±0.08 (-81.0%) | 0.27±0.01 (-0.6%) | 0.16±0.13 (-41.7%) | 0.23±0.10 (-6.3%) | 0.26±0.01 (-1.1%) | **0.28±0.01 (-0.1%)** |

### Table 22: Medium Ground Texture Test

| Task | SAC AE | DrQ-v2 | SAM-G | SMG | SADA | MaDi | SegDAC |
|---|---|---|---|---|---|---|---|
| LiftPegUpright | 0.21±0.01 (-1.3%) | **0.42±0.18 (-10.7%)** | 0.40±0.15 (-5.7%) | 0.22±0.00 (-0.1%) | 0.22±0.04 (+1.2%) | 0.22±0.01 (-0.1%) | 0.39±0.15 (-4.2%) |
| PickCube | 0.07±0.01 (-59.5%) | 0.39±0.03 (-15.8%) | 0.10±0.01 (-6.0%) | 0.28±0.06 (-9.6%) | 0.36±0.04 (-16.8%) | **0.43±0.06 (-1.6%)** | 0.33±0.03 (-3.1%) |
| PokeCube | 0.13±0.02 (-22.4%) | 0.40±0.09 (-7.7%) | 0.32±0.08 (+2.1%) | 0.22±0.13 (+3.7%) | 0.25±0.09 (-36.3%) | **0.40±0.01 (-2.2%)** | 0.39±0.02 (-1.5%) |
| PullCube | 0.15±0.04 (-58.8%) | 0.31±0.02 (-16.0%) | 0.13±0.02 (+2.8%) | 0.19±0.10 (-51.6%) | 0.24±0.04 (-50.6%) | 0.26±0.06 (-46.6%) | **0.51±0.06 (+1.6%)** |
| PullCubeTool | 0.19±0.04 (-13.7%) | 0.56±0.13 (-9.6%) | 0.24±0.10 (+7.0%) | 0.23±0.10 (-9.1%) | 0.30±0.15 (-16.8%) | 0.58±0.19 (-7.0%) | **0.72±0.14 (-3.3%)** |
| PushCube | 0.14±0.03 (-53.3%) | 0.43±0.22 (-6.7%) | **0.55±0.09 (+9.3%)** | 0.32±0.04 (-7.2%) | 0.33±0.08 (-27.7%) | 0.44±0.08 (-4.4%) | 0.49±0.06 (+9.1%) |
| PlaceAppleInBowl | 0.01±0.01 (-87.4%) | 0.00±0.01 (-97.4%) | 0.06±0.02 (-3.9%) | 0.07±0.06 (-38.7%) | 0.10±0.04 (-15.8%) | 0.06±0.05 (-47.2%) | **0.30±0.16 (-6.4%)** |
| TransportBox | 0.04±0.04 (-26.2%) | 0.11±0.10 (-58.6%) | 0.26±0.02 (-1.8%) | 0.06±0.06 (-78.7%) | 0.09±0.05 (-62.5%) | 0.11±0.06 (-58.4%) | **0.28±0.01 (-1.1%)** |

### Table 23: Medium Lighting Color Test

| Task | SAC AE | DrQ-v2 | SAM-G | SMG | SADA | MaDi | SegDAC |
|---|---|---|---|---|---|---|---|
| LiftPegUpright | 0.21±0.00 (-2.0%) | 0.19±0.00 (-60.2%) | 0.22±0.06 (-47.2%) | 0.22±0.00 (+0.5%) | 0.23±0.03 (+2.5%) | 0.22±0.01 (-0.6%) | **0.43±0.16 (+4.3%)** |
| PickCube | 0.05±0.01 (-70.7%) | 0.01±0.01 (-98.3%) | 0.06±0.02 (-46.8%) | 0.18±0.05 (-43.7%) | 0.10±0.05 (-73.0%) | **0.19±0.02 (-56.1%)** | 0.16±0.02 (-51.9%) |
| PokeCube | 0.06±0.04 (-60.8%) | 0.06±0.05 (-85.8%) | 0.09±0.03 (-70.9%) | 0.19±0.11 (-9.1%) | 0.18±0.07 (-53.1%) | **0.38±0.03 (-8.3%)** | 0.30±0.03 (-24.1%) |
| PullCube | 0.20±0.06 (-47.0%) | 0.05±0.03 (-86.8%) | 0.13±0.02 (+3.6%) | 0.33±0.04 (-14.9%) | 0.29±0.11 (-41.4%) | 0.37±0.05 (-24.0%) | **0.49±0.08 (-1.9%)** |
| PullCubeTool | 0.12±0.04 (-45.6%) | 0.06±0.03 (-90.4%) | 0.18±0.04 (-20.4%) | 0.18±0.08 (-20.4%) | 0.32±0.15 (-12.5%) | 0.15±0.12 (-76.3%) | **0.55±0.13 (-26.0%)** |
| PushCube | 0.18±0.03 (-39.7%) | 0.03±0.03 (-92.9%) | 0.08±0.01 (-84.2%) | 0.26±0.04 (-22.9%) | 0.32±0.05 (-31.0%) | 0.48±0.10 (+2.9%) | **0.53±0.06 (+18.1%)** |
| PlaceAppleInBowl | 0.05±0.01 (-32.0%) | 0.01±0.01 (-91.0%) | 0.06±0.01 (-3.4%) | 0.04±0.01 (-60.4%) | 0.10±0.04 (-17.4%) | 0.07±0.04 (-41.7%) | **0.28±0.14 (-11.2%)** |
| TransportBox | 0.05±0.03 (+4.9%) | 0.26±0.07 (-0.6%) | 0.27±0.01 (-0.1%) | 0.27±0.01 (+0.3%) | 0.25±0.09 (+0.4%) | 0.26±0.01 (-0.0%) | **0.28±0.01 (+0.4%)** |

## Table 24: Medium Lighting Direction Test

| Task | SAC AE | DrQ-v2 | SAM-G | SMG | SADA | MaDi | SegDAC |
|---|---|---|---|---|---|---|---|
| LiftPegUpright | 0.21±0.01 (-0.6%) | 0.23±0.03 (-51.0%) | 0.40±0.15 (-4.1%) | 0.22±0.00 (-0.1%) | 0.22±0.04 (-0.6%) | 0.21±0.01 (-2.4%) | **0.42±0.16 (+1.3%)** |
| PickCube | 0.16±0.03 (-14.0%) | 0.36±0.10 (-21.2%) | 0.11±0.01 (+6.4%) | 0.29±0.06 (-6.7%) | 0.35±0.05 (-8.7%) | **0.43±0.05 (-2.7%)** | 0.24±0.04 (-28.2%) |
| PokeCube | 0.15±0.01 (-7.2%) | 0.30±0.09 (-30.0%) | 0.27±0.08 (-12.8%) | 0.23±0.13 (+7.0%) | 0.36±0.07 (-7.3%) | **0.41±0.02 (-0.3%)** | 0.36±0.03 (-7.0%) |
| PullCube | 0.28±0.01 (-24.2%) | 0.34±0.10 (-6.8%) | 0.14±0.02 (+9.4%) | 0.39±0.07 (+0.4%) | 0.49±0.07 (+1.4%) | 0.49±0.03 (-1.3%) | **0.50±0.07 (+0.5%)** |
| PullCubeTool | 0.22±0.06 (+1.4%) | 0.47±0.20 (-25.1%) | 0.23±0.05 (+3.2%) | 0.22±0.10 (-2.9%) | 0.30±0.10 (-18.7%) | 0.58±0.20 (-6.3%) | **0.71±0.12 (-4.0%)** |
| PushCube | 0.25±0.04 (-16.8%) | **0.51±0.24 (+12.7%)** | 0.49±0.10 (-2.4%) | 0.32±0.04 (-6.9%) | 0.46±0.05 (-1.3%) | 0.46±0.10 (-1.1%) | 0.48±0.05 (+6.8%) |
| PlaceAppleInBowl | 0.07±0.02 (+0.3%) | 0.14±0.03 (-13.5%) | 0.06±0.02 (-5.6%) | 0.10±0.04 (-7.3%) | 0.12±0.04 (+0.8%) | 0.12±0.04 (+0.0%) | **0.29±0.15 (-7.3%)** |
| TransportBox | 0.05±0.03 (+0.3%) | 0.26±0.07 (-0.4%) | 0.27±0.01 (-0.3%) | 0.27±0.01 (+0.5%) | 0.25±0.09 (+0.0%) | 0.26±0.01 (-0.0%) | **0.28±0.01 (+0.2%)** |

## Table 25: Medium Mo Color Test

| Task | SAC AE | DrQ-v2 | SAM-G | SMG | SADA | MaDi | SegDAC |
|---|---|---|---|---|---|---|---|
| LiftPegUpright | 0.20±0.01 (-5.4%) | 0.23±0.04 (-51.3%) | 0.30±0.09 (-29.4%) | 0.22±0.01 (+0.7%) | 0.22±0.01 (-0.2%) | 0.21±0.01 (-4.8%) | **0.41±0.15 (-0.7%)** |
| PickCube | 0.11±0.03 (-37.7%) | 0.21±0.12 (-53.3%) | 0.11±0.02 (+6.1%) | 0.10±0.06 (-68.3%) | 0.22±0.09 (-43.6%) | 0.02±0.01 (-96.3%) | **0.27±0.05 (-18.8%)** |
| PokeCube | 0.07±0.01 (-55.0%) | 0.02±0.02 (-96.1%) | 0.18±0.03 (-41.8%) | 0.19±0.10 (-11.5%) | 0.17±0.03 (-55.4%) | 0.07±0.15 (-83.2%) | **0.40±0.04 (+1.5%)** |
| PullCube | 0.15±0.05 (-60.0%) | 0.08±0.02 (-77.8%) | 0.13±0.02 (+2.5%) | 0.30±0.07 (-22.0%) | 0.46±0.12 (-5.4%) | 0.10±0.06 (-79.1%) | **0.47±0.07 (-6.4%)** |
| PullCubeTool | 0.09±0.02 (-59.0%) | 0.03±0.04 (-94.8%) | 0.18±0.04 (-17.7%) | 0.18±0.08 (-21.6%) | 0.25±0.17 (-30.6%) | 0.12±0.05 (-80.1%) | **0.49±0.06 (-34.1%)** |
| PushCube | 0.15±0.01 (-48.7%) | 0.01±0.00 (-96.7%) | 0.34±0.16 (-31.8%) | 0.26±0.02 (-22.7%) | 0.41±0.16 (-11.3%) | 0.09±0.03 (-80.1%) | **0.47±0.03 (+4.7%)** |
| PlaceAppleInBowl | 0.01±0.00 (-80.0%) | 0.01±0.00 (-95.2%) | 0.06±0.02 (-3.7%) | 0.06±0.04 (-44.9%) | 0.04±0.04 (-64.2%) | 0.06±0.03 (-52.2%) | **0.29±0.16 (-8.3%)** |
| TransportBox | 0.05±0.03 (-1.0%) | 0.16±0.10 (-39.0%) | 0.27±0.01 (+0.3%) | 0.26±0.01 (-2.2%) | 0.23±0.09 (-5.5%) | 0.26±0.01 (-0.3%) | **0.28±0.01 (+0.3%)** |

## Table 26: Medium Mo Texture Test

| Task | SAC AE | DrQ-v2 | SAM-G | SMG | SADA | MaDi | SegDAC |
|---|---|---|---|---|---|---|---|
| LiftPegUpright | 0.20±0.01 (-5.5%) | 0.20±0.00 (-58.6%) | 0.22±0.01 (-46.4%) | 0.22±0.01 (-0.1%) | 0.21±0.01 (-5.2%) | 0.20±0.00 (-10.2%) | **0.30±0.09 (-27.5%)** |
| PickCube | 0.05±0.02 (-71.1%) | 0.01±0.00 (-97.3%) | **0.11±0.02 (+6.4%)** | 0.03±0.01 (-89.6%) | 0.02±0.02 (-94.3%) | 0.01±0.00 (-98.1%) | 0.10±0.02 (-70.2%) |
| PokeCube | 0.08±0.01 (-52.4%) | 0.07±0.02 (-84.4%) | 0.07±0.03 (-77.8%) | 0.19±0.10 (-12.7%) | 0.06±0.02 (-85.6%) | 0.16±0.10 (-60.8%) | **0.26±0.05 (-32.4%)** |
| PullCube | 0.10±0.01 (-72.0%) | 0.08±0.01 (-77.4%) | 0.12±0.01 (-1.2%) | 0.28±0.06 (-29.1%) | 0.08±0.04 (-83.1%) | 0.12±0.07 (-76.6%) | **0.47±0.07 (-6.8%)** |
| PullCubeTool | 0.07±0.02 (-68.1%) | 0.02±0.01 (-96.5%) | 0.14±0.03 (-37.4%) | 0.10±0.05 (-54.6%) | 0.04±0.07 (-89.1%) | 0.04±0.03 (-93.0%) | **0.20±0.08 (-73.1%)** |
| PushCube | 0.11±0.01 (-62.1%) | 0.02±0.00 (-96.5%) | 0.21±0.03 (-58.8%) | 0.20±0.02 (-40.5%) | 0.06±0.02 (-86.8%) | 0.10±0.04 (-78.9%) | **0.46±0.05 (+1.9%)** |
| PlaceAppleInBowl | 0.03±0.01 (-60.1%) | 0.04±0.02 (-78.1%) | 0.06±0.03 (-12.0%) | 0.04±0.02 (-59.9%) | 0.03±0.03 (-73.4%) | 0.04±0.02 (-67.8%) | **0.29±0.17 (-6.7%)** |
| TransportBox | 0.05±0.04 (+4.1%) | 0.25±0.08 (-4.9%) | 0.27±0.01 (-0.3%) | 0.27±0.01 (+0.4%) | 0.23±0.09 (-5.4%) | 0.26±0.01 (-0.0%) | **0.28±0.01 (-0.3%)** |

## Table 27: Medium Ro Color Test

| Task | SAC AE | DrQ-v2 | SAM-G | SMG | SADA | MaDi | SegDAC |
|---|---|---|---|---|---|---|---|
| LiftPegUpright | N/A | N/A | N/A | N/A | N/A | N/A | N/A |
| PickCube | N/A | N/A | N/A | N/A | N/A | N/A | N/A |
| PokeCube | 0.16±0.02 (-5.1%) | 0.30±0.06 (-30.2%) | 0.31±0.08 (-1.0%) | 0.21±0.12 (-1.5%) | 0.36±0.07 (-8.0%) | **0.41±0.02 (-0.9%)** | 0.38±0.03 (-3.0%) |
| PullCube | 0.30±0.05 (-18.4%) | 0.16±0.09 (-57.3%) | 0.12±0.01 (-5.8%) | 0.38±0.09 (-2.3%) | **0.53±0.19 (+9.2%)** | 0.51±0.11 (+3.8%) | 0.70±0.11 (-5.1%) |
| PullCubeTool | 0.21±0.06 (-12.6%) | 0.54±0.26 (-12.6%) | 0.21±0.06 (-4.1%) | 0.23±0.10 (+0.5%) | 0.33±0.21 (-10.4%) | 0.63±0.20 (+0.6%) | **0.70±0.11 (-5.1%)** |
| PushCube | 0.25±0.03 (-14.7%) | 0.35±0.23 (-24.3%) | **0.49±0.13 (-2.1%)** | 0.32±0.03 (-7.6%) | 0.45±0.10 (-2.2%) | 0.49±0.07 (+6.1%) | 0.45±0.05 (+0.4%) |
| PlaceAppleInBowl | 0.08±0.02 (+12.4%) | 0.14±0.03 (-17.0%) | 0.07±0.02 (-1.0%) | 0.11±0.06 (-4.0%) | 0.12±0.04 (-2.9%) | 0.11±0.05 (-6.7%) | **0.25±0.09 (-19.5%)** |
| TransportBox | N/A | N/A | N/A | N/A | N/A | N/A | N/A |

## Table 28: Medium Ro Texture Test

| Task | SAC AE | DrQ-v2 | SAM-G | SMG | SADA | MaDi | SegDAC |
|---|---|---|---|---|---|---|---|
| LiftPegUpright | N/A | N/A | N/A | N/A | N/A | N/A | N/A |
| PickCube | N/A | N/A | N/A | N/A | N/A | N/A | N/A |
| PokeCube | 0.17±0.02 (+3.2%) | 0.32±0.04 (-26.8%) | 0.32±0.08 (+2.6%) | 0.23±0.12 (+6.9%) | 0.37±0.08 (-5.1%) | **0.41±0.01 (-1.0%)** | 0.39±0.03 (-1.4%) |
| PullCube | 0.35±0.04 (-6.4%) | 0.09±0.08 (-75.1%) | 0.10±0.03 (-22.5%) | 0.36±0.08 (-7.2%) | **0.60±0.11 (+22.6%)** | 0.43±0.09 (-12.3%) | 0.51±0.06 (+2.6%) |
| PullCubeTool | 0.20±0.07 (-9.9%) | 0.50±0.23 (-18.8%) | 0.25±0.08 (+11.0%) | 0.23±0.10 (+0.7%) | 0.33±0.17 (-10.8%) | 0.67±0.19 (+7.2%) | **0.70±0.14 (-5.9%)** |
| PushCube | 0.24±0.04 (-19.3%) | 0.28±0.23 (-38.1%) | 0.45±0.08 (-9.8%) | 0.32±0.03 (-7.5%) | **0.52±0.08 (+13.6%)** | 0.46±0.07 (-1.2%) | 0.50±0.04 (+12.0%) |
| PlaceAppleInBowl | 0.07±0.03 (-1.4%) | 0.14±0.03 (-14.0%) | 0.07±0.02 (-1.0%) | 0.10±0.06 (-8.4%) | 0.12±0.04 (+0.6%) | 0.12±0.05 (+1.3%) | **0.22±0.12 (-29.0%)** |
| TransportBox | N/A | N/A | N/A | N/A | N/A | N/A | N/A |

## Table 29: Medium Table Color Test

| Task | SAC AE | DrQ-v2 | SAM-G | SMG | SADA | MaDi | SegDAC |
|---|---|---|---|---|---|---|---|
| LiftPegUpright | 0.21±0.01 (-3.3%) | 0.19±0.00 (-60.0%) | **0.39±0.10 (-7.8%)** | 0.22±0.01 (-2.5%) | 0.22±0.03 (+0.6%) | 0.20±0.01 (-10.9%) | 0.38±0.13 (-6.7%) |
| PickCube | 0.04±0.00 (-79.6%) | 0.00±0.00 (-99.1%) | 0.11±0.02 (-0.6%) | 0.05±0.02 (-82.8%) | 0.08±0.02 (-80.0%) | 0.06±0.04 (-85.6%) | **0.17±0.04 (-48.9%)** |
| PokeCube | 0.08±0.03 (-53.4%) | 0.05±0.05 (-89.3%) | 0.21±0.04 (-33.4%) | 0.10±0.06 (-51.1%) | **0.29±0.05 (-25.9%)** | 0.23±0.12 (-44.9%) | 0.28±0.02 (-28.4%) |
| PullCube | 0.12±0.09 (-66.4%) | 0.03±0.03 (-90.7%) | 0.12±0.01 (-2.5%) | 0.24±0.08 (-39.5%) | 0.36±0.06 (-26.2%) | 0.36±0.20 (-27.0%) | **0.42±0.06 (-15.9%)** |
| PullCubeTool | 0.12±0.09 (-44.9%) | 0.06±0.04 (-90.9%) | 0.28±0.09 (+26.8%) | 0.08±0.05 (-66.7%) | 0.19±0.10 (-48.7%) | 0.09±0.02 (-85.6%) | **0.46±0.07 (-37.6%)** |
| PushCube | 0.13±0.05 (-56.0%) | 0.05±0.03 (-89.5%) | 0.33±0.04 (-34.9%) | 0.07±0.05 (-78.7%) | 0.31±0.03 (-33.4%) | 0.12±0.09 (-74.3%) | **0.63±0.12 (+39.3%)** |
| PlaceAppleInBowl | 0.02±0.02 (-66.4%) | 0.03±0.02 (-81.0%) | 0.04±0.02 (-34.8%) | 0.02±0.01 (-83.1%) | 0.09±0.04 (-21.7%) | 0.03±0.02 (-79.0%) | **0.28±0.15 (-11.1%)** |
| TransportBox | 0.03±0.04 (-37.4%) | 0.04±0.04 (-84.6%) | 0.23±0.04 (-15.8%) | 0.18±0.10 (-32.6%) | 0.21±0.09 (-15.7%) | 0.19±0.08 (-27.4%) | **0.28±0.01 (-0.7%)** |

## Table 30: Medium Table Texture Test

| Task | SAC AE | DrQ-v2 | SAM-G | SMG | SADA | MaDi | SegDAC |
|---|---|---|---|---|---|---|---|
| LiftPegUpright | 0.21±0.01 (-2.5%) | 0.19±0.00 (-60.2%) | **0.43±0.14 (+3.5%)** | 0.22±0.01 (-1.2%) | 0.22±0.04 (-2.3%) | 0.22±0.01 (-1.1%) | 0.41±0.15 (-1.3%) |
| PickCube | 0.11±0.02 (-39.8%) | 0.02±0.01 (-96.4%) | 0.09±0.01 (-12.7%) | 0.20±0.04 (-35.3%) | 0.15±0.05 (-61.2%) | **0.30±0.08 (-30.9%)** | 0.23±0.06 (-32.8%) |
| PokeCube | 0.12±0.02 (-25.6%) | 0.03±0.02 (-92.2%) | 0.32±0.07 (+2.3%) | 0.18±0.10 (-14.5%) | 0.17±0.03 (-55.3%) | **0.37±0.07 (-8.7%)** | 0.32±0.04 (-18.8%) |
| PullCube | 0.06±0.03 (-82.4%) | 0.01±0.00 (-97.0%) | 0.12±0.01 (-3.8%) | 0.32±0.07 (-19.2%) | 0.37±0.08 (-23.4%) | 0.41±0.07 (-17.0%) | **0.49±0.04 (-2.3%)** |
| PullCubeTool | 0.06±0.03 (-72.1%) | 0.05±0.09 (-92.3%) | 0.24±0.11 (+6.4%) | 0.13±0.07 (-45.5%) | 0.09±0.10 (-75.0%) | 0.54±0.19 (-13.4%) | **0.64±0.13 (-13.3%)** |
| PushCube | 0.19±0.04 (-36.0%) | 0.02±0.02 (-95.3%) | **0.45±0.05 (-10.9%)** | 0.28±0.06 (-17.9%) | 0.21±0.06 (-54.5%) | 0.40±0.08 (-13.0%) | 0.43±0.07 (-3.6%) |
| PlaceAppleInBowl | 0.01±0.00 (-90.7%) | 0.02±0.01 (-90.5%) | 0.05±0.04 (-17.3%) | 0.02±0.01 (-84.2%) | 0.04±0.02 (-64.5%) | 0.01±0.02 (-88.8%) | **0.29±0.14 (-8.6%)** |
| TransportBox | 0.05±0.03 (+2.9%) | 0.14±0.04 (-45.6%) | 0.26±0.01 (-4.3%) | 0.26±0.01 (-4.2%) | 0.17±0.10 (-28.8%) | 0.23±0.03 (-13.0%) | **0.28±0.01 (+0.2%)** |

## Table 31: Hard Camera Fov Test

| Task | SAC AE | DrQ-v2 | SAM-G | SMG | SADA | MaDi | SegDAC |
|---|---|---|---|---|---|---|---|
| LiftPegUpright | 0.19±0.00 (-12.7%) | 0.19±0.01 (-59.9%) | 0.19±0.00 (-55.2%) | **0.22±0.01 (-1.7%)** | 0.20±0.02 (-9.5%) | 0.20±0.01 (-10.2%) | 0.20±0.01 (-51.8%) |
| PickCube | 0.02±0.01 (-91.6%) | 0.02±0.00 (-95.0%) | 0.01±0.00 (-91.0%) | 0.01±0.01 (-96.1%) | 0.06±0.03 (-84.0%) | 0.03±0.03 (-93.0%) | **0.07±0.02 (-78.7%)** |
| PokeCube | 0.02±0.02 (-89.9%) | 0.02±0.01 (-95.2%) | 0.05±0.03 (-82.8%) | **0.07±0.03 (-68.3%)** | 0.06±0.03 (-85.1%) | 0.02±0.02 (-94.2%) | 0.05±0.01 (-86.3%) |
| PullCube | 0.06±0.03 (-83.6%) | 0.01±0.00 (-96.4%) | **0.12±0.04 (-7.4%)** | 0.09±0.01 (-75.9%) | 0.09±0.03 (-81.2%) | 0.05±0.02 (-89.7%) | 0.11±0.01 (-77.1%) |
| PullCubeTool | 0.02±0.03 (-92.8%) | 0.02±0.03 (-96.3%) | 0.03±0.02 (-86.3%) | 0.06±0.07 (-74.5%) | 0.03±0.03 (-91.9%) | 0.05±0.02 (-92.2%) | **0.16±0.04 (-78.6%)** |
| PushCube | 0.01±0.01 (-95.6%) | 0.02±0.03 (-95.4%) | 0.02±0.01 (-95.8%) | 0.02±0.01 (-93.7%) | 0.07±0.02 (-85.2%) | 0.04±0.02 (-91.4%) | **0.12±0.05 (-72.5%)** |
| PlaceAppleInBowl | 0.01±0.00 (-90.2%) | 0.01±0.01 (-91.0%) | 0.02±0.01 (-76.3%) | 0.01±0.01 (-88.0%) | 0.02±0.01 (-81.8%) | 0.02±0.02 (-85.7%) | **0.03±0.01 (-90.8%)** |
| TransportBox | 0.03±0.03 (-46.0%) | 0.10±0.03 (-62.7%) | 0.09±0.04 (-65.2%) | 0.05±0.07 (-82.0%) | 0.11±0.04 (-54.0%) | 0.04±0.01 (-84.6%) | **0.28±0.01 (-0.6%)** |

## Table 32: Hard Camera Pose Test

| Task | SAC AE | DrQ-v2 | SAM-G | SMG | SADA | MaDi | SegDAC |
|---|---|---|---|---|---|---|---|
| LiftPegUpright | 0.19±0.00 (-11.6%) | 0.19±0.00 (-59.7%) | 0.19±0.01 (-53.9%) | **0.22±0.01 (-1.0%)** | 0.20±0.01 (-10.0%) | 0.19±0.00 (-13.4%) | 0.20±0.01 (-52.1%) |
| PickCube | 0.01±0.01 (-92.1%) | 0.02±0.01 (-96.7%) | 0.01±0.00 (-90.9%) | 0.02±0.01 (-92.3%) | 0.03±0.01 (-93.1%) | 0.03±0.02 (-94.2%) | **0.07±0.01 (-79.4%)** |
| PokeCube | 0.01±0.01 (-92.1%) | 0.01±0.01 (-97.0%) | 0.03±0.01 (-90.7%) | 0.05±0.04 (-76.8%) | **0.06±0.02 (-84.9%)** | 0.03±0.02 (-93.1%) | 0.05±0.00 (-87.2%) |
| PullCube | 0.02±0.01 (-94.5%) | 0.01±0.01 (-98.3%) | **0.07±0.03 (-41.3%)** | 0.02±0.03 (-95.5%) | 0.05±0.02 (-88.9%) | 0.02±0.01 (-95.3%) | 0.05±0.04 (-89.1%) |
| PullCubeTool | 0.02±0.03 (-92.3%) | 0.01±0.01 (-98.1%) | 0.00±0.01 (-97.8%) | 0.03±0.05 (-86.7%) | 0.03±0.02 (-90.9%) | 0.01±0.01 (-97.6%) | **0.09±0.05 (-87.3%)** |
| PushCube | 0.04±0.03 (-86.7%) | 0.01±0.01 (-97.1%) | 0.02±0.01 (-96.0%) | 0.02±0.01 (-94.0%) | 0.02±0.01 (-96.6%) | 0.02±0.02 (-95.2%) | **0.10±0.06 (-78.3%)** |
| PlaceAppleInBowl | 0.01±0.00 (-93.1%) | 0.01±0.01 (-94.5%) | 0.02±0.01 (-76.6%) | 0.01±0.01 (-90.1%) | **0.03±0.04 (-71.0%)** | 0.01±0.02 (-90.1%) | 0.03±0.02 (-90.9%) |
| TransportBox | 0.03±0.03 (-42.1%) | 0.02±0.03 (-91.0%) | 0.08±0.04 (-68.6%) | 0.03±0.05 (-87.9%) | 0.06±0.03 (-74.1%) | 0.03±0.03 (-90.4%) | **0.28±0.01 (-1.1%)** |

## Table 33: Hard Ground Color Test

| Task | SAC AE | DrQ-v2 | SAM-G | SMG | SADA | MaDi | SegDAC |
|---|---|---|---|---|---|---|---|
| LiftPegUpright | 0.20±0.01 (-4.6%) | 0.19±0.01 (-59.7%) | 0.35±0.15 (-16.3%) | 0.22±0.00 (+0.3%) | 0.21±0.01 (-6.5%) | 0.22±0.01 (+0.4%) | **0.38±0.14 (-8.5%)** |
| PickCube | 0.02±0.01 (-89.6%) | 0.01±0.01 (-96.8%) | 0.11±0.02 (+2.8%) | 0.27±0.05 (-12.4%) | 0.09±0.07 (-75.6%) | **0.38±0.06 (-14.2%)** | 0.36±0.03 (+6.0%) |
| PokeCube | 0.08±0.02 (-50.3%) | 0.01±0.01 (-97.0%) | 0.27±0.09 (-15.0%) | 0.23±0.13 (+7.1%) | 0.10±0.03 (-74.2%) | 0.34±0.12 (-16.7%) | **0.35±0.04 (-10.3%)** |
| PullCube | 0.01±0.01 (-96.2%) | 0.02±0.03 (-94.1%) | 0.13±0.01 (+2.8%) | 0.27±0.12 (-30.1%) | 0.38±0.08 (-21.2%) | 0.23±0.21 (-52.3%) | **0.52±0.08 (+3.2%)** |
| PullCubeTool | 0.07±0.04 (-66.3%) | 0.06±0.06 (-90.4%) | 0.27±0.08 (+22.1%) | 0.22±0.10 (-4.9%) | 0.06±0.06 (-84.0%) | 0.46±0.17 (-25.5%) | **0.60±0.14 (-18.5%)** |
| PushCube | 0.08±0.05 (-74.7%) | 0.02±0.01 (-96.7%) | 0.54±0.10 (+8.5%) | 0.31±0.04 (-10.1%) | 0.34±0.10 (-26.3%) | 0.29±0.13 (-38.4%) | 0.47±0.04 (+5.2%) |
| PlaceAppleInBowl | 0.00±0.00 (-96.1%) | 0.00±0.00 (-97.4%) | 0.06±0.02 (-2.2%) | 0.04±0.03 (-65.8%) | 0.06±0.03 (-50.9%) | 0.03±0.06 (-76.1%) | **0.30±0.16 (-4.0%)** |
| TransportBox | 0.05±0.05 (+4.5%) | 0.01±0.02 (-95.3%) | 0.27±0.01 (+0.2%) | 0.21±0.11 (-21.6%) | 0.23±0.10 (-8.0%) | 0.26±0.02 (-0.6%) | **0.28±0.01 (-0.4%)** |

## Table 34: Hard Ground Texture Test

| Task | SAC AE | DrQ-v2 | SAM-G | SMG | SADA | MaDi | SegDAC |
|---|---|---|---|---|---|---|---|
| LiftPegUpright | 0.21±0.01 (-0.9%) | 0.20±0.01 (-57.0%) | 0.37±0.14 (-12.0%) | 0.22±0.01 (+0.1%) | 0.21±0.02 (-4.6%) | 0.22±0.01 (+0.0%) | **0.40±0.15 (-2.1%)** |
| PickCube | 0.07±0.02 (-61.4%) | 0.10±0.06 (-78.9%) | 0.10±0.01 (-4.9%) | 0.31±0.06 (-1.5%) | 0.36±0.05 (-6.8%) | **0.43±0.07 (-3.0%)** | 0.32±0.02 (-4.2%) |
| PokeCube | 0.10±0.03 (-40.4%) | 0.14±0.10 (-67.0%) | 0.32±0.08 (+2.0%) | 0.22±0.13 (+4.7%) | 0.27±0.09 (-30.8%) | **0.40±0.01 (-2.1%)** | 0.38±0.04 (-3.3%) |
| PullCube | 0.13±0.04 (-64.5%) | 0.14±0.06 (-61.1%) | 0.12±0.01 (-1.7%) | 0.15±0.13 (-61.5%) | 0.42±0.16 (-14.8%) | 0.31±0.09 (-37.8%) | **0.55±0.05 (+9.6%)** |
| PullCubeTool | 0.09±0.04 (-56.6%) | 0.08±0.19 (-86.8%) | 0.21±0.06 (-6.1%) | 0.23±0.10 (-0.4%) | 0.23±0.07 (-37.2%) | 0.55±0.19 (-12.1%) | **0.68±0.12 (-8.7%)** |
| PushCube | 0.12±0.04 (-59.1%) | 0.25±0.21 (-46.1%) | **0.49±0.06 (-1.5%)** | 0.30±0.03 (-11.8%) | 0.39±0.07 (-15.6%) | 0.47±0.09 (+1.9%) | 0.49±0.07 (+9.4%) |
| PlaceAppleInBowl | 0.01±0.00 (-89.1%) | 0.00±0.01 (-97.1%) | 0.07±0.02 (-0.6%) | 0.05±0.06 (-52.2%) | 0.01±0.01 (-89.3%) | 0.01±0.02 (-88.9%) | **0.27±0.14 (-14.7%)** |
| TransportBox | 0.04±0.03 (-12.7%) | 0.10±0.11 (-62.2%) | **0.27±0.01 (-0.8%)** | 0.05±0.05 (-79.8%) | 0.08±0.05 (-66.1%) | 0.07±0.06 (-71.6%) | 0.27±0.01 (-1.6%) |

## Table 35: Hard Lighting Color Test

| Task | SAC AE | DrQ-v2 | SAM-G | SMG | SADA | MaDi | SegDAC |
|---|---|---|---|---|---|---|---|
| LiftPegUpright | 0.22±0.01 (+0.6%) | 0.20±0.01 (-58.6%) | 0.25±0.09 (-39.5%) | 0.22±0.00 (-0.1%) | 0.22±0.04 (-2.2%) | 0.20±0.01 (-8.9%) | **0.39±0.15 (-4.4%)** |
| PickCube | 0.02±0.01 (-91.2%) | 0.01±0.02 (-98.0%) | 0.05±0.02 (-53.7%) | 0.08±0.03 (-74.8%) | 0.07±0.02 (-82.5%) | 0.07±0.03 (-83.7%) | **0.15±0.01 (-55.8%)** |
| PokeCube | 0.08±0.02 (-53.6%) | 0.03±0.01 (-93.2%) | 0.10±0.03 (-67.6%) | 0.19±0.03 (-39.5%) | 0.19±0.03 (-51.5%) | 0.21±0.04 (-49.2%) | **0.37±0.05 (-4.8%)** |
| PullCube | 0.02±0.01 (-93.4%) | 0.04±0.04 (-89.0%) | 0.11±0.01 (-9.7%) | 0.27±0.05 (-30.7%) | 0.23±0.10 (-53.7%) | 0.42±0.08 (-14.3%) | **0.49±0.09 (-2.1%)** |
| PullCubeTool | 0.01±0.02 (-94.2%) | 0.03±0.02 (-95.6%) | 0.17±0.03 (-23.7%) | 0.12±0.08 (-47.0%) | 0.17±0.12 (-52.3%) | 0.08±0.05 (-87.5%) | **0.57±0.15 (-23.3%)** |
| PushCube | 0.06±0.01 (-78.4%) | 0.04±0.02 (-90.4%) | 0.15±0.04 (-70.9%) | 0.24±0.05 (-30.7%) | 0.35±0.09 (-23.1%) | 0.36±0.09 (-21.6%) | **0.54±0.05 (+21.0%)** |
| PlaceAppleInBowl | 0.01±0.01 (-89.8%) | 0.01±0.00 (-95.2%) | 0.05±0.03 (-26.1%) | 0.03±0.02 (-69.6%) | 0.07±0.02 (-43.4%) | 0.01±0.02 (-93.0%) | **0.25±0.12 (-21.2%)** |
| TransportBox | 0.05±0.03 (+2.2%) | 0.26±0.07 (-0.3%) | 0.27±0.01 (+0.3%) | 0.27±0.01 (+0.3%) | 0.25±0.09 (+0.0%) | 0.26±0.01 (+0.1%) | **0.28±0.01 (-0.2%)** |

## Table 36: Hard Lighting Direction Test

| Task | SAC AE | DrQ-v2 | SAM-G | SMG | SADA | MaDi | SegDAC |
|---|---|---|---|---|---|---|---|
| LiftPegUpright | 0.21±0.01 (-1.9%) | 0.19±0.01 (-59.5%) | **0.40±0.12 (-5.0%)** | 0.22±0.00 (-0.5%) | 0.22±0.05 (-0.3%) | 0.22±0.01 (+2.3%) | 0.37±0.12 (-10.3%) |
| PickCube | 0.10±0.02 (-44.6%) | 0.02±0.01 (-94.6%) | 0.09±0.02 (-12.5%) | 0.31±0.05 (+0.4%) | 0.35±0.04 (-10.8%) | **0.40±0.08 (-8.9%)** | 0.19±0.02 (-43.1%) |
| PokeCube | 0.13±0.01 (-22.5%) | 0.07±0.04 (-84.9%) | 0.26±0.06 (-16.2%) | 0.21±0.12 (+0.4%) | 0.35±0.07 (-10.0%) | **0.40±0.02 (-2.2%)** | 0.20±0.05 (-49.3%) |
| PullCube | 0.17±0.04 (-54.5%) | 0.05±0.07 (-87.7%) | 0.14±0.01 (+13.9%) | 0.41±0.11 (+4.6%) | **0.52±0.19 (+7.1%)** | 0.45±0.04 (-8.7%) | 0.47±0.08 (-5.8%) |
| PullCubeTool | 0.17±0.04 (-24.2%) | 0.04±0.18 (-92.9%) | 0.22±0.08 (-1.2%) | 0.23±0.10 (+1.1%) | 0.32±0.19 (-11.5%) | 0.61±0.18 (-1.4%) | **0.64±0.09 (-13.7%)** |
| PushCube | 0.21±0.02 (-29.7%) | 0.05±0.04 (-89.4%) | 0.38±0.09 (-25.0%) | 0.33±0.01 (-3.5%) | **0.44±0.03 (-4.3%)** | 0.43±0.07 (-6.9%) | 0.44±0.05 (-2.7%) |
| PlaceAppleInBowl | 0.07±0.02 (-3.1%) | 0.01±0.01 (-91.3%) | 0.06±0.01 (-9.5%) | 0.07±0.05 (-36.3%) | 0.10±0.04 (-15.9%) | 0.11±0.04 (-9.0%) | **0.24±0.11 (-25.5%)** |
| TransportBox | 0.05±0.03 (+1.3%) | 0.26±0.07 (-0.7%) | 0.27±0.01 (-0.4%) | 0.27±0.01 (+0.1%) | 0.24±0.09 (-0.5%) | 0.26±0.01 (+0.0%) | **0.28±0.01 (+0.5%)** |

## Table 37: Hard Mo Color Test

| Task | SAC AE | DrQ-v2 | SAM-G | SMG | SADA | MaDi | SegDAC |
|---|---|---|---|---|---|---|---|
| LiftPegUpright | 0.21±0.01 (-1.6%) | 0.19±0.00 (-59.4%) | 0.22±0.01 (-47.5%) | 0.22±0.01 (-0.3%) | 0.20±0.01 (-10.3%) | 0.20±0.00 (-8.9%) | **0.39±0.12 (-5.6%)** |
| PickCube | 0.06±0.02 (-68.2%) | 0.02±0.01 (-96.6%) | 0.11±0.01 (+3.1%) | 0.04±0.02 (-88.3%) | 0.04±0.02 (-90.0%) | 0.01±0.01 (-98.1%) | **0.28±0.03 (-17.2%)** |
| PokeCube | 0.08±0.01 (-53.1%) | 0.01±0.01 (-97.1%) | 0.20±0.04 (-36.8%) | 0.07±0.04 (-69.2%) | 0.02±0.03 (-96.0%) | 0.01±0.00 (-97.2%) | **0.34±0.05 (-13.8%)** |
| PullCube | 0.10±0.04 (-73.3%) | 0.08±0.01 (-79.5%) | 0.13±0.02 (+2.1%) | 0.18±0.07 (-54.4%) | 0.09±0.04 (-81.1%) | 0.12±0.04 (-75.8%) | **0.48±0.07 (-3.9%)** |
| PullCubeTool | 0.04±0.02 (-80.4%) | 0.05±0.12 (-91.8%) | 0.26±0.09 (+18.7%) | 0.19±0.17 (-46.8%) | 0.19±0.17 (-20.0%) | 0.07±0.04 (-89.0%) | **0.44±0.10 (-40.2%)** |
| PushCube | 0.10±0.02 (-65.9%) | 0.02±0.01 (-96.4%) | 0.28±0.03 (-43.7%) | 0.04±0.03 (-88.7%) | 0.06±0.02 (-88.0%) | 0.08±0.03 (-82.7%) | **0.48±0.05 (+7.0%)** |
| PlaceAppleInBowl | 0.02±0.01 (-78.6%) | 0.01±0.00 (-92.9%) | 0.07±0.02 (+1.9%) | 0.05±0.04 (-53.8%) | 0.06±0.03 (-46.2%) | 0.06±0.02 (-51.9%) | **0.30±0.17 (-6.4%)** |
| TransportBox | 0.05±0.03 (+1.8%) | 0.18±0.11 (-30.1%) | 0.26±0.01 (-3.0%) | 0.27±0.01 (-0.2%) | 0.24±0.10 (-2.2%) | 0.26±0.01 (-0.9%) | **0.28±0.01 (-0.2%)** |

Table 38: Hard Ro Color Test

| Task | SAC AE | DrQ-v2 | SAM-G | SMG | SADA | MaDi | SegDAC |
|---|---|---|---|---|---|---|---|
| LiftPegUpright | N/A | N/A | N/A | N/A | N/A | N/A | N/A |
| PickCube | N/A | N/A | N/A | N/A | N/A | N/A | N/A |
| PokeCube | 0.11±0.03 (-34.2%) | 0.06±0.01 (-85.9%) | 0.30±0.08 (-3.2%) | 0.19±0.08 (-10.9%) | 0.20±0.06 (-49.0%) | 0.30±0.07 (-27.9%) | **0.39±0.05 (+0.0%)** |
| PullCube | 0.04±0.02 (-88.0%) | 0.01±0.01 (-96.4%) | 0.11±0.01 (-15.9%) | 0.32±0.05 (-16.8%) | 0.09±0.08 (-81.3%) | 0.09±0.12 (-82.1%) | 0.50±0.05 (-1.2%) |
| PullCubeTool | 0.14±0.05 (-35.2%) | 0.28±0.09 (-54.7%) | 0.22±0.05 (-1.8%) | 0.21±0.09 (-7.5%) | 0.23±0.14 (-38.2%) | 0.42±0.16 (-32.3%) | 0.67±0.18 (-9.0%) |
| PushCube | 0.06±0.01 (-78.9%) | 0.04±0.01 (-91.3%) | 0.44±0.10 (-11.7%) | 0.30±0.04 (-11.5%) | 0.04±0.02 (-90.7%) | 0.29±0.13 (-38.2%) | **0.46±0.04 (+1.7%)** |
| PlaceAppleInBowl | 0.04±0.02 (-52.6%) | 0.01±0.01 (-92.4%) | 0.06±0.02 (-7.3%) | 0.07±0.04 (-40.1%) | 0.11±0.04 (-11.0%) | 0.07±0.03 (-44.3%) | **0.27±0.12 (-15.6%)** |
| TransportBox | N/A | N/A | N/A | N/A | N/A | N/A | N/A |

Table 39: Hard Ro Texture Test

| Task | SAC AE | DrQ-v2 | SAM-G | SMG | SADA | MaDi | SegDAC |
|---|---|---|---|---|---|---|---|
| LiftPegUpright | N/A | N/A | N/A | N/A | N/A | N/A | N/A |
| PickCube | N/A | N/A | N/A | N/A | N/A | N/A | N/A |
| PokeCube | 0.16±0.02 (-4.3%) | 0.25±0.01 (-43.2%) | 0.31±0.07 (-1.4%) | 0.22±0.10 (+1.4%) | 0.36±0.08 (-6.8%) | **0.40±0.01 (-3.3%)** | 0.30±0.06 (-23.7%) |
| PullCube | 0.29±0.08 (-20.7%) | 0.06±0.07 (-84.8%) | 0.09±0.02 (-27.7%) | 0.37±0.09 (-5.2%) | **0.54±0.12 (+11.5%)** | 0.46±0.12 (-6.3%) | 0.52±0.11 (+3.5%) |
| PullCubeTool | 0.18±0.08 (-17.0%) | 0.39±0.18 (-37.5%) | 0.22±0.07 (+0.1%) | 0.23±0.10 (-0.4%) | 0.31±0.19 (-16.1%) | 0.61±0.19 (-1.2%) | **0.66±0.16 (-11.1%)** |
| PushCube | 0.27±0.05 (-10.7%) | 0.11±0.11 (-75.4%) | 0.46±0.08 (-9.1%) | 0.31±0.03 (-9.0%) | 0.48±0.10 (+4.3%) | **0.51±0.14 (+9.1%)** | 0.44±0.07 (-1.5%) |
| PlaceAppleInBowl | 0.08±0.02 (+2.9%) | 0.12±0.05 (-24.3%) | 0.07±0.02 (+4.6%) | 0.09±0.04 (-16.3%) | 0.13±0.04 (+5.7%) | 0.10±0.06 (-17.0%) | **0.26±0.10 (-16.9%)** |
| TransportBox | N/A | N/A | N/A | N/A | N/A | N/A | N/A |

Table 40: Hard Table Texture Test

| Task | SAC AE | DrQ-v2 | SAM-G | SMG | SADA | MaDi | SegDAC |
|---|---|---|---|---|---|---|---|
| LiftPegUpright | 0.19±0.01 (-10.9%) | 0.20±0.01 (-58.3%) | **0.22±0.02 (-48.1%)** | **0.22±0.01 (-3.2%)** | 0.20±0.01 (-8.7%) | 0.20±0.01 (-11.2%) | 0.20±0.01 (-51.7%) |
| PickCube | 0.03±0.01 (-83.7%) | 0.02±0.01 (-96.0%) | 0.10±0.01 (-2.7%) | **0.11±0.06 (-63.9%)** | 0.07±0.03 (-82.5%) | 0.05±0.04 (-89.4%) | 0.07±0.02 (-79.9%) |
| PokeCube | 0.07±0.05 (-54.9%) | 0.02±0.00 (-96.0%) | 0.25±0.05 (-21.5%) | 0.12±0.08 (-44.9%) | 0.08±0.02 (-78.2%) | 0.06±0.03 (-85.4%) | 0.15±0.04 (-62.3%) |
| PullCube | 0.03±0.02 (-93.0%) | 0.01±0.00 (-98.4%) | 0.13±0.01 (+2.6%) | 0.14±0.07 (-63.9%) | **0.16±0.04 (-67.7%)** | 0.12±0.04 (-75.2%) | 0.04±0.03 (-92.5%) |
| PullCubeTool | 0.07±0.07 (-67.7%) | 0.03±0.03 (-94.6%) | 0.16±0.03 (-29.2%) | 0.11±0.07 (-50.7%) | 0.05±0.03 (-86.5%) | **0.29±0.17 (-53.0%)** | 0.02±0.02 (-97.4%) |
| PushCube | 0.04±0.01 (-86.2%) | 0.03±0.02 (-92.4%) | **0.19±0.04 (-62.2%)** | 0.05±0.02 (-84.2%) | 0.08±0.02 (-83.7%) | 0.05±0.03 (-90.2%) | 0.10±0.06 (-78.2%) |
| PlaceAppleInBowl | 0.01±0.00 (-82.9%) | 0.01±0.02 (-91.9%) | 0.04±0.01 (-46.6%) | 0.02±0.04 (-80.2%) | 0.05±0.02 (-56.9%) | 0.01±0.02 (-91.3%) | **0.25±0.11 (-20.2%)** |
| TransportBox | 0.05±0.04 (+7.6%) | 0.15±0.08 (-41.3%) | 0.26±0.01 (-1.9%) | 0.24±0.03 (-11.5%) | 0.25±0.08 (+3.7%) | 0.24±0.04 (-9.3%) | **0.27±0.01 (-2.2%)** |

# B ABLATIONS

## B.1 TEXT INPUTS ABLATION

### B.1.1 DOES LEARNING TO IGNORE BACKGROUND IMPROVE VISUAL GENERALIZATION?

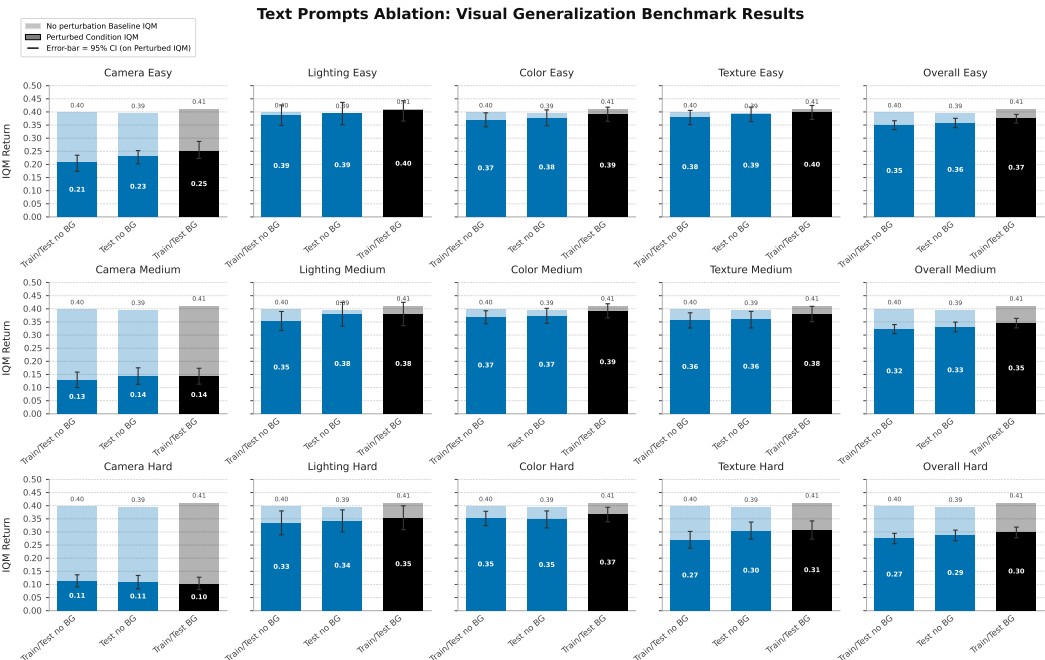

Figure 26: Visual generalization results when including or excluding the "background" text tags.

We conducted an ablation to evaluate whether including the "background" text tag improves visual generalization in SegDAC. In the "Train/Test no BG" setting (Figure 26), the model is trained and tested without the "background" tag. For example, in the Push Cube task, the input tags would be ["robot", "gripper", "small box", "target"] instead of ["background", "robot", "gripper", "small box", "target"]. In the "Test no BG" setting, the model is trained with the background tag but tested without it. The final configuration, shown in black in Figure 26, uses the background tag during both training and testing and corresponds to the final version of SegDAC.

The goal of this ablation is to determine whether training with background segments encourages the model to ignore irrelevant information more effectively, or if excluding them allows it to focus more directly on task-relevant objects. While YOLO-World and SAM can still introduce irrelevant segments even without the background tag, our experiments show that including the tag leads to a noticeably larger number of irrelevant segments.

Results in Figure 26 show that all configurations perform similarly, but "Train/Test BG" achieves about $\sim 9\%$ higher IQM returns. This suggests that including the background tag improves generalization by encouraging the model to focus less on distractions.

### B.1.2 TEXT INPUTS SENSITIVITY

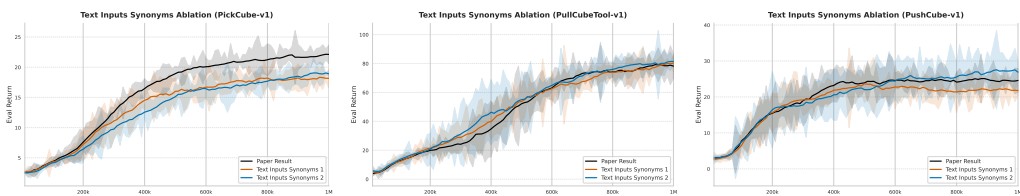

Figure 27: Sample efficiency when using different text inputs synonyms (using 3 seeds).

Table 41: Text inputs used for the synonym ablation for each task.

| Text Input Set | Pick Cube | Pull Cube Tool | Push Cube |
|---|---|---|---|
| Default | background, robot, gripper, small box, sphere | background, robot, gripper, small box, block | background, robot, gripper, small box, target |
| Synonyms 1 | background, robot, manipulator, box, target | background, robot, manipulator, box, tool, handle | background, robot, manipulator, box, goal |
| Synonyms 2 | background, bot, end-effector, cube, circle | background, robot, end effector, cube, peg, stick | background, bot, arm, cube, bullseye |

Figure 27 shows the sample efficiency curves when training with different sets of synonyms for the text inputs used by the YOLO-World detector, the exact text inputs are shown in table 41. Across all tasks we observe that the overall learning trend remains stable. Some tasks show small increases or decreases in absolute performance, but the shape of the curve is consistent and no collapse occurs. This indicates that SegDAC is not sensitive to the exact choice of words used to describe the objects. The method does not require heavy prompt engineering or tuning of phrasing, which is often a concern in work relying on large language models. YOLO-World operates on short, simple concept words and ignores terms that do not match the frame. Together these results suggest that the text-guided segmentation component of SegDAC is stable and easy to use in practice.

### B.1.3 SHARED TEXT INPUTS

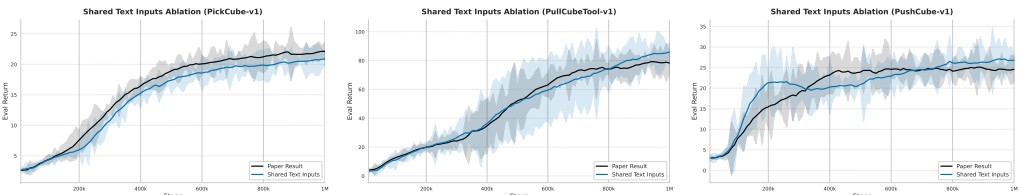

Figure 28: Sample efficiency when using a shared list of text inputs for all tasks (using 3 seeds).

We evaluate the robustness of SegDAC by using a single, shared vocabulary across all tasks. This contrasts with the default setting where each task uses a specific set of text inputs. Table 42 details the specific text inputs used. Although we evaluated this ablation on three specific tasks, the shared text inputs includes concepts required for all eight tasks presented in this paper. This design simulates a realistic scenario where a single shared vocabulary is used across a diverse set of tasks/environments.

Figure 28 presents the sample efficiency results. We observe that performance remains stable across all tasks when using the shared list. The shape of the learning curves is consistent with the baseline and no performance collapse occurs. This suggests that a single general vocabulary can be reused across different tasks.

We attribute this stability to two main factors. First, the text tags in YOLO-World are used to score potential regions. If a text input describes an object that is not visible in the scene, the detector generally does not output a bounding box. Consequently, the number of generated segments is primarily determined by the image content rather than the size of the text vocabulary. Second, while the object detector may produce a slightly different number of bounding boxes when the text inputs change, SegDAC is explicitly trained to handle a variable number of segments. This ensures the policy remains robust to fluctuations in the count or granularity of detected objects.

Table 42: Comparison of the shared text inputs versus the task-specific default lists.

| Configuration | Text Inputs |
|---|---|
| **Shared Text Inputs (Ablation)** | background, robot, robot hand, gripper, box, target, block, bar, apple, bowl, sphere, wooden peg |
| **Pick Cube (Default)** | background, robot, gripper, small box, sphere |
| **Pull Cube Tool (Default)** | background, robot, gripper, small box, block |
| **Push Cube (Default)** | background, robot, gripper, small box, target |

Furthermore, the number of resulting SAM masks remains modest (typically 5 to 25) even with a larger vocabulary. This keeps the computational cost significantly lower than a typical transformer encoder that would need to process thousands of patch embeddings. For these experiments, we maintained the same low object detection threshold used in the main results and did not perform specific parameter tuning for the shared text inputs setting.

## B.2 OBJECT-CENTRIC VS GLOBAL IMAGE REPRESENTATIONS

To assess whether object-centric representations are more effective than global image features, we compared SegDAC to an SAC baseline that uses a fixed-length global image representation. This baseline, referred to as SAC SAM Encoder, computes the mean of SAM's patch embeddings over the entire image, without using segmentation. The resulting vector is then processed by an MLP with the same number of layers as the projection head used in SegDAC to predict actions or Q-values.

Figure 29 shows the results. SegDAC outperforms SAC SAM Encoder on 7 out of 8 tasks and ties on the remaining one. This highlights a key limitation of directly using pre-trained encoders like SAM for global image representations in online visual RL. Even after attempting hyperparameter tuning for SAC SAM Encoder, we observed no improvement in performance.

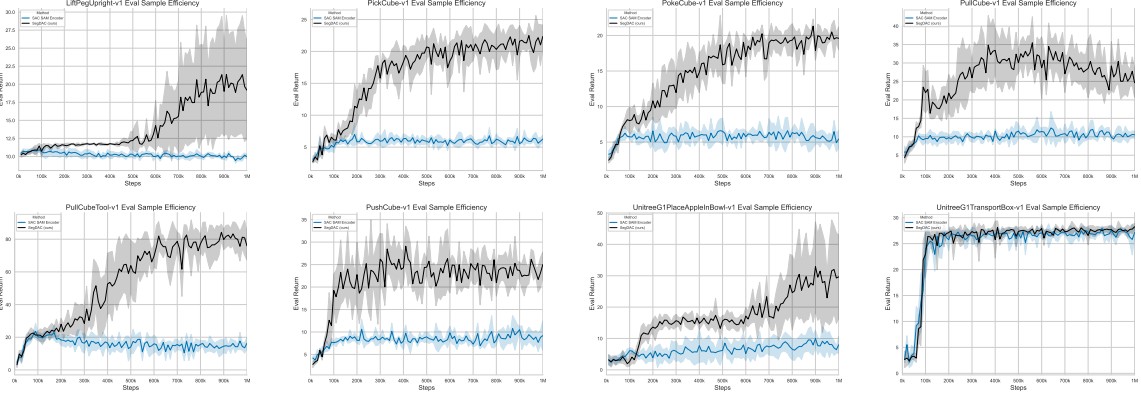

Figure 29: Sample efficiency comparison of SegDAC vs SAC SAM Encoder which uses a global image representation instead of the object-centric representations used by SegDAC.

## C    TASKS DEFINITIONS

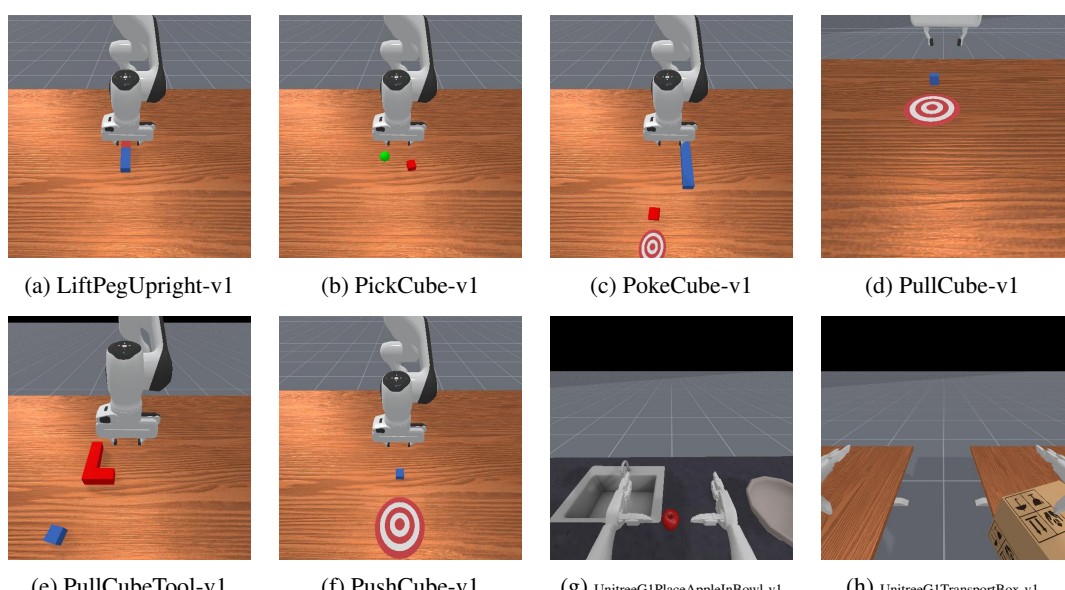

(a) LiftPegUpright-v1    (b) PickCube-v1    (c) PokeCube-v1    (d) PullCube-v1

(e) PullCubeTool-v1    (f) PushCube-v1    (g) UnitreeG1PlaceAppleInBowl-v1    (h) UnitreeG1TransportBox-v1

Figure 30: Default (no perturbation) scenes for each of the 8 tasks used for training.

To train SegDAC and the baselines, we used 8 tasks from ManiSkill3 (version `mani_skill==3.0.0b21`) with the default normalized dense reward. The tasks and their corresponding configurations are summarized in table 43 and visualized in figure 30.

| Task Name | Max Episode Length | Control Mode |
|---|---|---|
| LiftPegUpright-v1 | 50 | PD EE Delta Pos |
| PickCube-v1* | 50 | PD EE Delta Pos |
| PokeCube-v1 | 50 | PD EE Delta Pos |
| PullCube-v1 | 50 | PD EE Delta Pos |
| PullCubeTool-v1 | 100 | PD EE Delta Pos |
| PushCube-v1 | 50 | PD EE Delta Pos |
| UnitreeG1PlaceAppleInBowl-v1 | 100 | PD Joint Delta Pos |
| UnitreeG1TransportBox-v1 | 100 | PD Joint Delta Pos |

Table 43: Task configurations used for training SegDAC and baselines.

* For the PickCube-v1 task, we created a modified version that makes the visual target sphere visible. This allows vision-based policies to leverage goal information. We achieved this by commenting out the following line in the original task code:

```
# self._hidden_objects.append(self.goal_site)
```

## D    HYPERPARAMETERS

Each method was trained using its own hyperparameter configuration. Once a configuration was found to perform well for a given method, it was kept fixed across all tasks. For the baselines, we used the official implementations provided by the authors and used their default hyperparameters as starting points. In most cases, these settings were appropriate without modification. The only adjustment we made was to the discount factor, which we set to 0.80 for all methods, as we found this value worked better with ManiSkill3 and is also used in the framework's official baselines.

For SegDAC, text tags were provided manually per task by a human. However, automatic tagging is also feasible using a VLM or image text tagging models such as RAM++ (Zhang et al., 2023b). We opted for manual tags to reduce the scope of this research and to reduce computational overhead.

Note that due to computational constraints, we did not perform an extensive hyperparameter search for SegDAC, and we did not tune the parameters of YOLO-World or SAM.

Tables 45 and 46 summarize the key hyperparameters used for SegDAC and the baselines. We report only the most relevant parameters for reproducibility, omitting architectural defaults that are standard for each algorithm. For the complete list of hyperparameters, readers are invited to check the official code repository on the project page (anonymized for review).

Table 44: Text tags used by SegDAC for each task.

| Task | Text Tags |
|---|---|
| LiftPegUpright-v1 | background, robot, gripper, rectangle bar, wooden peg |
| PickCube-v1 | background, robot, gripper, small box, sphere |
| PushCube-v1 | background, robot, gripper, small box, target |
| PullCube-v1 | background, robot, gripper, small box, target |
| PokeCube-v1 | background, robot, gripper, small box, bar, target |
| PullCubeTool-v1 | background, robot, gripper, small box, block |
| UnitreeG1PlaceAppleInBowl-v1 | background, robot, robot hand, gripper, apple, bowl |
| UnitreeG1TransportBox-v1 | background, robot, robot hand, gripper, box |

The list of text tags used by SegDAC for each task is provided in Table 44.

| Hyperparameter | SegDAC |
|---|---|
| Actor learning rate | $3 \times 10^{-4}$ |
| Critic learning rate | $5 \times 10^{-4}$ |
| Entropy learning rate | $3 \times 10^{-4}$ |
| Optimizer | Adam |
| Gamma (discount) | 0.80 |
| Target update rate ($\tau$) | 0.01 |
| Actor update frequency | 1 |
| Critic update frequency | 1 |
| Target networks update frequency | 2 |
| Image resolution | 512 |
| Min log std | -10 |
| Max log std | 2 |
| Embedding dim | 128 |
| Transformer decoder layers | 6 |
| Transformer decoder heads | 8 |
| Transformer decoder hidden size | 1024 |
| Transformer dropout | 0.0 |
| Projection head | ResidualMLP |
| Projection head hidden layers | 4 |
| Projection head hidden size | 256 |
| Projection head norm | LayerNorm |
| Projection head pre-normalize input | true |
| Projection head activation | ReLU |
| YOLO-World | yolov8s-worldv2 |
| YOLO-World Confidence Threshold | 0.0001 |
| YOLO-World IoU Threshold | 0.01 |
| EfficientViT SAM | efficientvit-sam-l0 |
| Mask post-processing kernel size | 9 |
| Segment Embedding Min Pixel | 4 |
| Replay Buffer Max Size | 1,000,000 |

Table 45: Key hyperparameters used for training SegDAC.

Note that in SegDAC, the actor and critic each use their own transformer decoder and projection head, but they share the same architecture and hyperparameters. For clarity, we omit separate entries for each and report the shared configuration.

| Hyperparameter | SADA | MaDi | DrQ-v2 | SAC-AE | SAM-G | SMG |
|---|---|---|---|---|---|---|
| Actor Learning Rate | $5 \times 10^{-4}$ | $1 \times 10^{-3}$ | $1 \times 10^{-4}$ | $1 \times 10^{-4}$ | $1 \times 10^{-4}$ | $1 \times 10^{-3}$ |
| Critic Learning Rate | $5 \times 10^{-4}$ | $1 \times 10^{-3}$ | $1 \times 10^{-4}$ | $1 \times 10^{-3}$ | $1 \times 10^{-4}$ | $1 \times 10^{-3}$ |
| Entropy Learning Rate | $5 \times 10^{-4}$ | $1 \times 10^{-4}$ | – | $1 \times 10^{-4}$ | – | $1 \times 10^{-4}$ |
| Target Update Rate ($\tau$) | 0.01 | 0.01 | 0.01 | 0.01 | 0.01 | 0.01 |
| Actor Update Frequency | 2 | 2 | 2 | 1 | 4 | 2 |
| Critic Update Frequency | 2 | 1 | 2 | 1 | 4 | 2 |
| Target Networks Update Frequency | 2 | 2 | 2 | 2 | 4 | 2 |
| Replay Buffer Size | 1,000,000 | 1,000,000 | 1,000,000 | 1,000,000 | 1,000,000 | 1,000,000 |
| Gamma (discount) | 0.80 | 0.80 | 0.80 | 0.80 | 0.80 | 0.80 |
| Image Resolution | 84 | 84 | 84 | 84 | 84 | 84 |
| MLP Projection Layers | 4 | 3 | 4 | 3 | 4 | 3 |
| MLP Features Dim | 1024 | 1024 | 1024 | 1024 | 1024 | 1024 |
| Frame Stack | 3 | 3 | 3 | 3 | 3 | 3 |
| N-step Return | – | – | 3 | – | 3 | – |
| Optimizer | Adam | Adam | Adam | Adam | Adam | Adam |

Table 46: Key hyperparameters for baseline methods. Note that SAM-G doesn't support batching so we had to change the update frequency to match the update-to-data ratio of 0.25 used by every other methods.

## E SEGMENTATION DETAILS

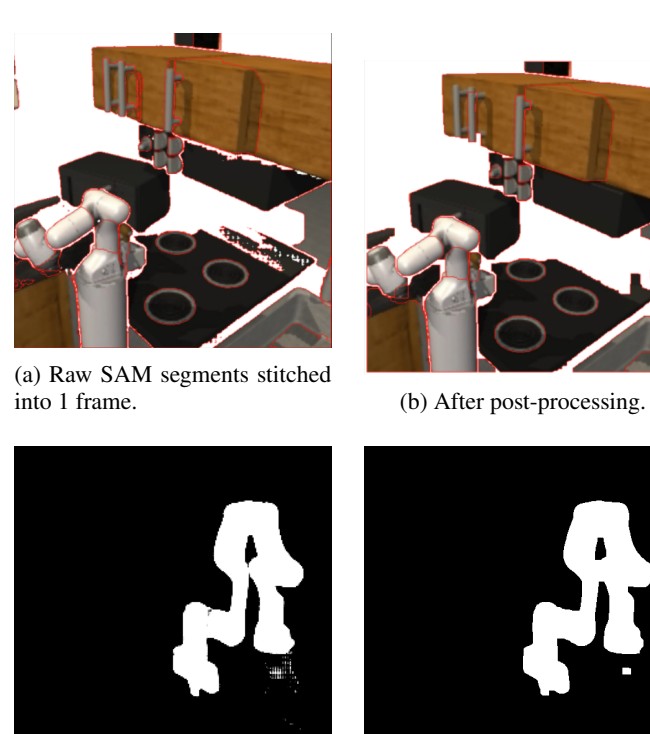

(a) Raw SAM segments stitched into 1 frame.

(b) After post-processing.

(c) Raw binary mask.

(d) Post-processed mask.

Figure 31: Effect of our lightweight post-processing on SAM outputs. Top: stitched overlays. Bottom: single mask.

To improve the quality of segments predicted by SAM, prior work such as SAM (Kirillov et al., 2023), SAM-G (Wang et al., 2023), and PerSAM (Zhang et al., 2023a) typically rely on heavy post-processing pipelines, often involving iterative mask refinement. In contrast, we adopt a lightweight and efficient approach based on morphological opening and closing operations (Sreedhar, 2012). These classical image processing techniques are fast and effective at removing small artifacts such as holes or sprinkles commonly found in raw segmentation masks. The impact of this post-processing is illustrated in Figure 31.

Our implementation of this post-processing procedure is written in pure PyTorch, as shown below:

```python
import torch
import torch.nn.functional as F

...
    def post_process_masks(self, masks: torch.Tensor, kernel_size: int)
        -> torch.Tensor:
        """
        masks: (N,1,H,W)
        """
        opened = self.apply_morphological_opening(
            masks.to(torch.float32), kernel_size=kernel_size
        )
        return self.apply_morphological_closing(opened,
            kernel_size=kernel_size).to(
            torch.uint8
        )

    def apply_morphological_opening(
```

```
         self, masks: torch.Tensor, kernel_size: int
     ) -> torch.Tensor:
         eroded = self.apply_erosion(masks, kernel_size)
         return self.apply_dilation(eroded, kernel_size)

     def apply_erosion(self, masks: torch.Tensor, kernel_size: int) ->
         torch.Tensor:
         return -self.max_pool2d_same_dim(-masks, kernel_size=kernel_size)

     def apply_dilation(self, masks: torch.Tensor, kernel_size: int) ->
         torch.Tensor:
         return self.max_pool2d_same_dim(masks, kernel_size=kernel_size)

     def max_pool2d_same_dim(self, masks: torch.Tensor, kernel_size: int):
         stride = 1
         dilation = 1
         pad_h_top, pad_h_bottom = self.compute_padding(
             kernel_size, stride, dilation)
         pad_w_left, pad_w_right = self.compute_padding(
             kernel_size, stride, dilation)

         padded_input = F.pad(
             masks, (pad_w_left, pad_w_right, pad_h_top, pad_h_bottom))

         return F.max_pool2d(
             padded_input, kernel_size=kernel_size, stride=stride,
                 dilation=dilation
         )

     def compute_padding(self, kernel_size, stride=1, dilation=1):
         padding_total = max(0, (kernel_size - 1) * dilation - stride + 1)
         pad_before = padding_total // 2
         pad_after = padding_total - pad_before
         return pad_before, pad_after

     def apply_morphological_closing(
         self, masks: torch.Tensor, kernel_size: int
     ) -> torch.Tensor:
         dilated = self.apply_dilation(masks, kernel_size)
         return self.apply_erosion(dilated, kernel_size)
```

# F    SEGMENT EMBEDDINGS EXTRACTION DETAILS

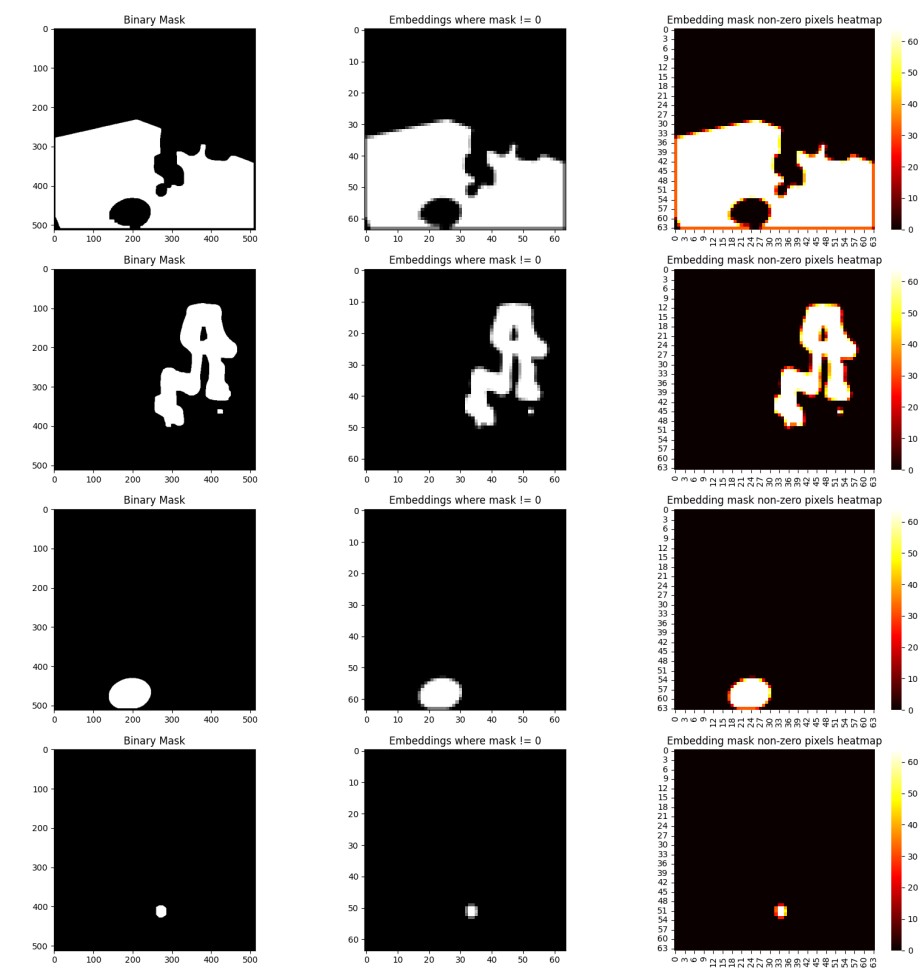

Figure 32: Left: binary mask for the selected segment. Middle: SAM encoder patch grid, showing only patches whose spatial footprint overlaps the mask. Right: per-patch mask–overlap heatmap, defined as the number (or fraction) of mask pixels inside each patch. Boundary patches have lower overlap (red) because they straddle multiple segments, while interior patches approach full coverage (white). This heatmap reflects spatial overlap only, not the contextual information carried by the patch embeddings.

Figure 32 illustrates the relationship between the binary masks produced by our grounded segmentation module (left), the corresponding **overlapping** patch embeddings (middle), and a heatmap showing the pixel count per patch (right). Most patch embeddings have a high pixel count, while those near the mask edges tend to overlap less exclusively with the segment.

# G    SEGMENT ATTENTION ANALYSIS

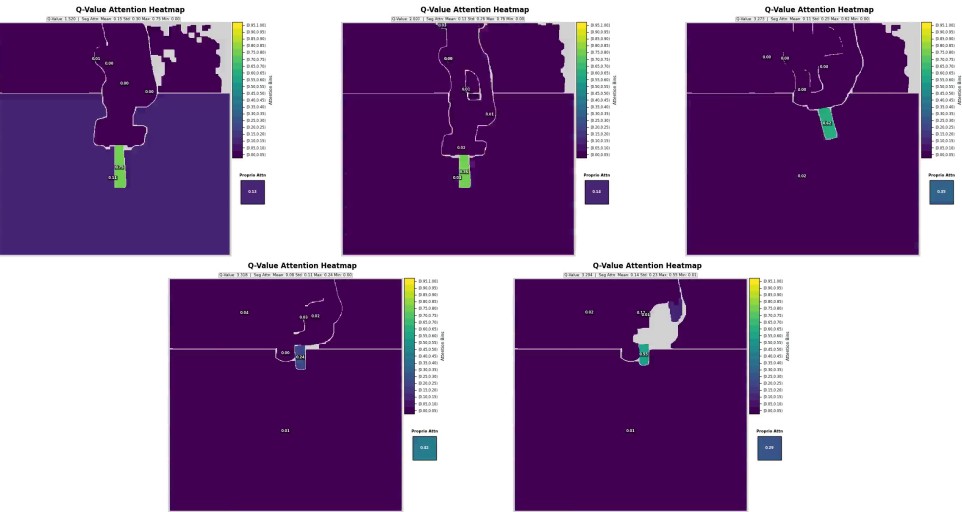

Figure 33:    Average segment attention from the critic when predicting Q-values in the LiftPegUpright-v1 task. The trajectory begins in the top-left frame. Grey regions indicate areas where SAM did not detect any segments.

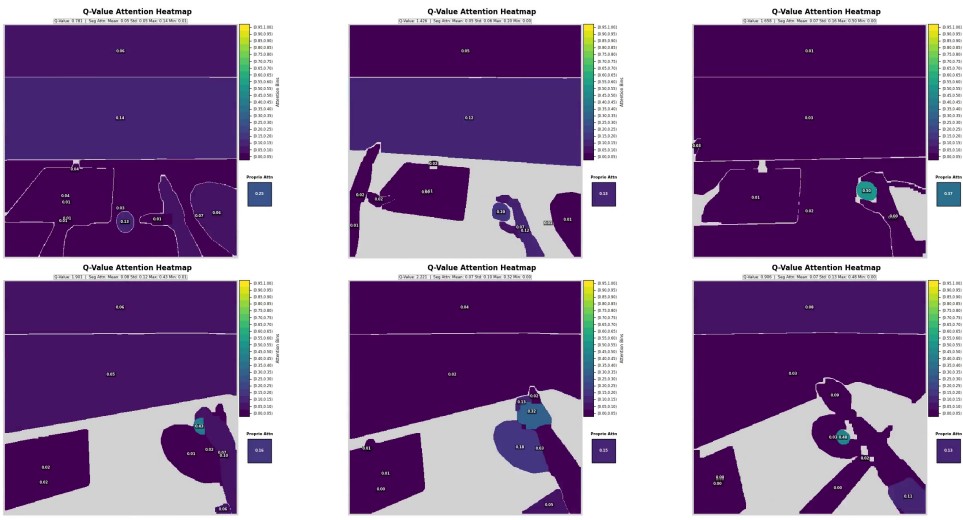

Figure 34:    Average segment attention from the critic when predicting Q-values in the UnitreeG1PlaceAppleInBowl-v1 task. The trajectory begins in the top-left frame. Grey regions indicate areas where SAM did not detect any segments.

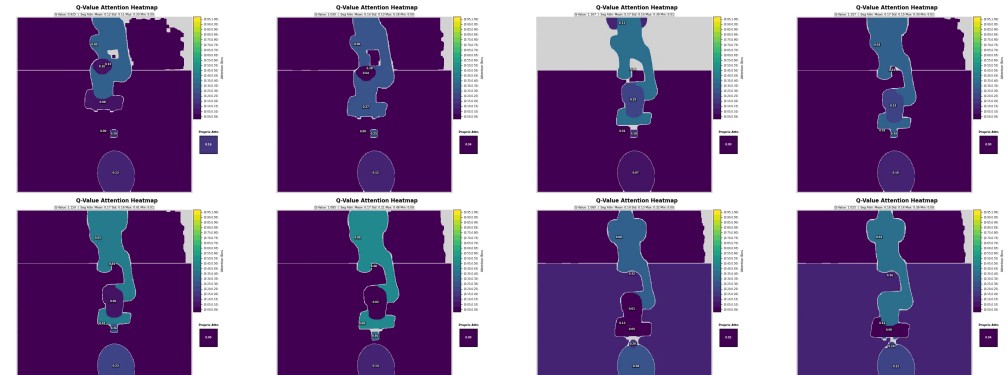

Figure 35: Average segment attention from the critic when predicting Q-values in the PushCube-v1 task. The trajectory begins in the top-left frame. Grey regions indicate areas where SAM did not detect any segments.

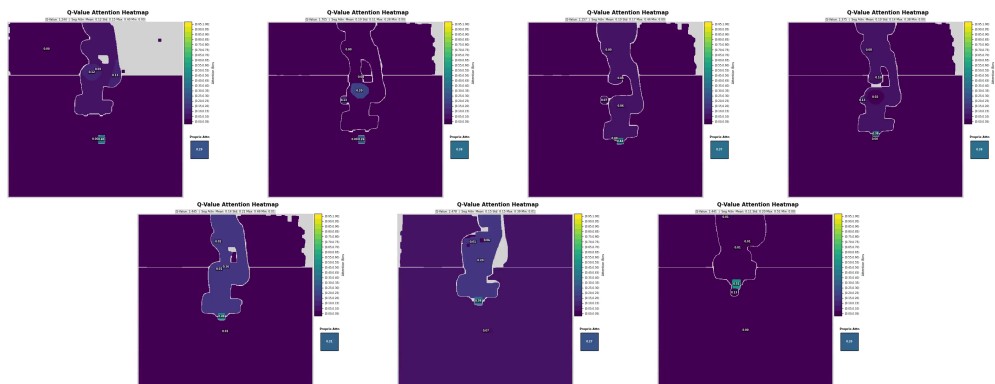

Figure 36: Average segment attention from the critic when predicting Q-values in the PickCube-v1 task. The trajectory begins in the top-left frame. Grey regions indicate areas where SAM did not detect any segments.

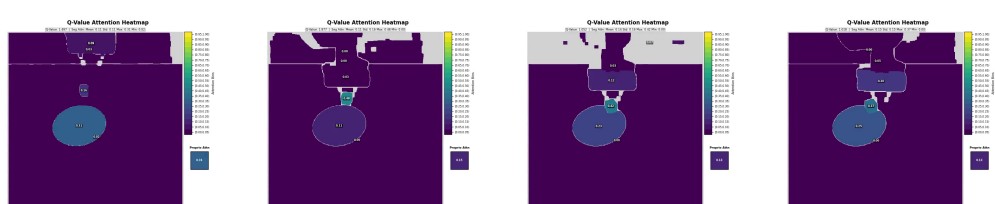

Figure 37: Average segment attention from the critic when predicting Q-values in the PullCube-v1 task. The trajectory begins on the left frame. Grey regions indicate areas where SAM did not detect any segments.

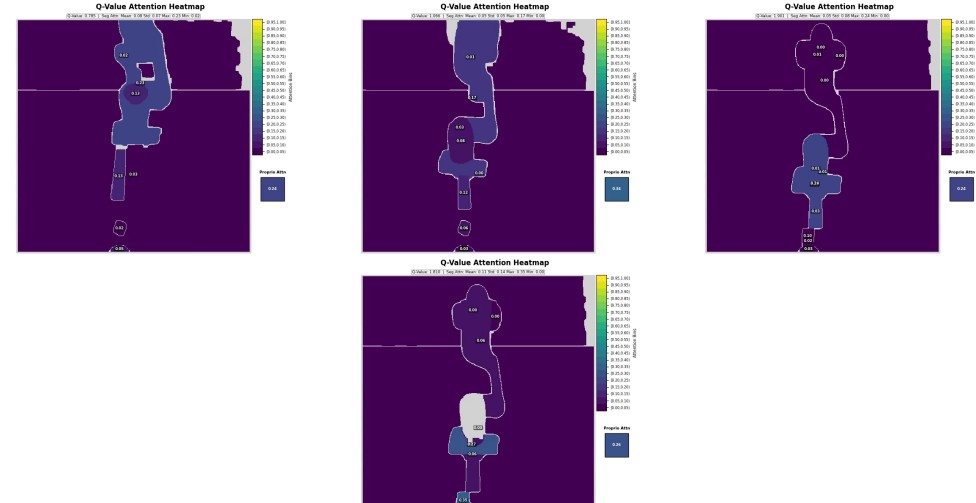

Figure 38: Average segment attention from the critic when predicting Q-values in the PokeCube-v1 task. The trajectory begins in the top-left frame. Grey regions indicate areas where SAM did not detect any segments.

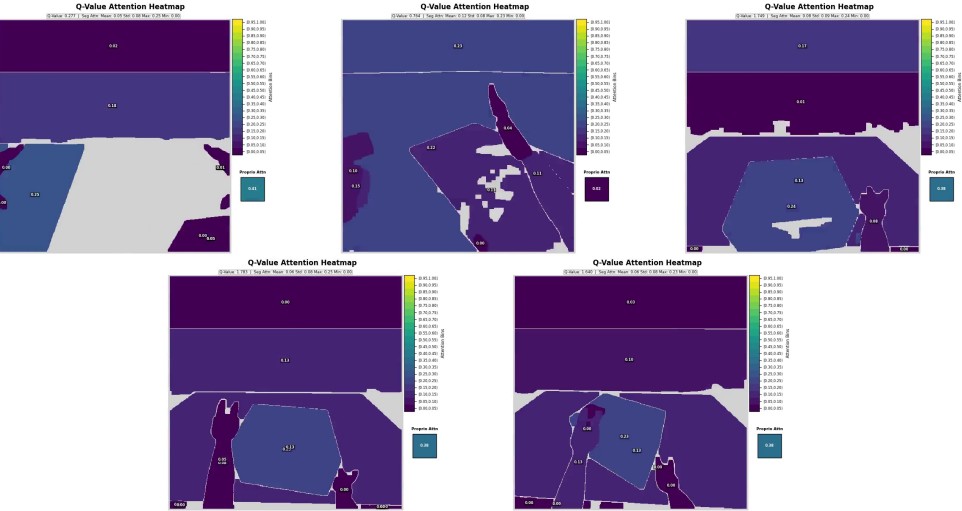

Figure 39: Average segment attention from the critic when predicting Q-values in the UnitreeG1TransportBox-v1 task. The trajectory begins in the top-left frame. Grey regions indicate areas where SAM did not detect any segments.

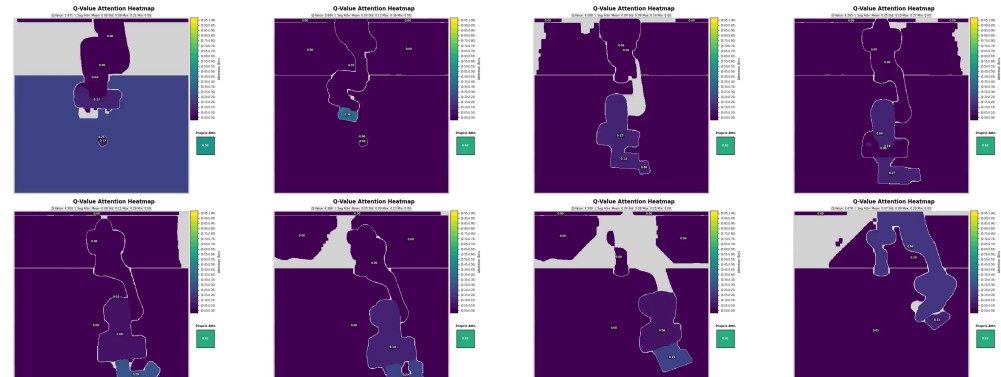

Figure 40: Average segment attention from the critic when predicting Q-values in the PullCubeTool-v1 task. The trajectory begins in the top-left frame. Grey regions indicate areas where SAM did not detect any segments.

Figures 33 to 40 show the average segment attention from the critic when predicting Q-values for all eight tasks. Grey regions indicate areas where SAM did not detect any segments.

We observe that SegDAC consistently attends to task-relevant segments, such as the peg in LiftPegUpright-v1, while ignoring background elements when they provide no useful information. For example, in LiftPegUpright-v1, the peg receives approximately 75% of the attention early in the trajectory. Once the peg is upright, the model shifts some attention to proprioceptive inputs, possibly to stabilize the peg.

SegDAC also demonstrates robustness to variability in detected segments. In PickCube-v1, early frames show the servo button segmented as a small circular region, which the critic uses to guide positioning. In later frames, when the full robot arm is detected instead, the critic adapts and reallocates attention accordingly to complete the task.

Across tasks, SegDAC often allocates significant attention to the manipulation object. For instance, the apple receives about 50% of the attention in UnitreeG1PlaceAppleInBowl-v1. Similarly, in LiftPegUpright-v1, the peg receives around 75% early on.

Figure 36 illustrates that SegDAC is robust to partial occlusions. At the beginning of the trajectory, the goal region is fully occluded by the robot arm, and the model focuses its attention on the cube, which aligns with the sub-goal of grasping it first. As the cube is picked up and the arm moves, the goal region becomes partially visible. Even when it becomes occluded again during the motion, SegDAC continues moving the arm toward the correct location. The goal region becomes visible near the end, and the model successfully completes the task. This shows that SegDAC can handle some level of dynamic occlusions and maintain task-relevant behavior throughout.

## H  IMPLEMENTATION DETAILS

### H.1  POLICY EVALUATION AND DATA COLLECTION

In this work, we use **evaluation** to refer to the policy's performance under the *no-perturbation* setting during training, which we use to track sample efficiency, samples used for evaluation are not seen during training. **Testing** refers to performance on the *visual generalization benchmark*, which includes various visual perturbations. This distinction separates standard in-distribution evaluation from robustness testing under distribution shifts.

During both evaluation and testing, stochastic policies were run in deterministic mode by taking the mean of their action distributions. We followed the ManiSkill3 evaluation protocol[1], which disables

---

[1]https://maniskill.readthedocs.io/en/v3.0.0b21/user_guide/
reinforcement_learning/setup.html#evaluation

early termination to ensure all trajectories have the same length. This makes performance metrics more comparable.

Evaluation metrics were averaged over 5 random seeds, each with 10 rollouts. Proper seeding was applied at the start of both training and evaluation scripts. We used 10 parallel environments to collect evaluation rollouts during training.

For training, we used 20 parallel GPU environments and performed 5 gradient updates per environment step with a batch size of 128, resulting in an update-to-data ratio of 0.25.

## H.2 ONLINE RL OPTIMIZATION

Training online RL agents with large models such as SAM on high-resolution images (512x512) is computationally expensive. To complete experiments within one day on a single L40s GPU, we applied various key optimizations.

**Prompt-Based Segmentation**  FTD relied on prompt-free segmentation, which we found to be significantly slower. In our tests, using bounding boxes as prompts for SAM yielded about a $10\times$ speedup compared to prompt-free mode. Prompt-based segmentation also provides semantic grounding, making it both faster and better aligned with our design.

**Segmentation Post-Processing**  SAM-G applied iterative refinement, repeatedly invoking SAM to improve mask quality. While effective, this was too slow for online RL. We instead use a lightweight post-processing step that runs in about 1 ms on an L40s GPU. Further details are given in Appendix E.

**Efficient Vision Backbone.**  We replaced the original SAM model (Kirillov et al., 2023) with EfficientViT-SAM (Zhang et al., 2024b), which offers comparable segmentation quality but significantly faster inference speed, approximately $48\times$ faster according to the authors. We also tested SAM 2 (Ravi et al., 2024), which improves encoder efficiency but remained significantly slower than EfficientViT-SAM in practice.

All RGB images were processed at a resolution of (3, 512, 512) in (C, H, W) format. This is significantly higher than standard visual RL settings, which often operate in the (3, 84, 84) to (3, 128, 128) range (Laskin et al., 2020; Kostrikov et al., 2021; Srinivas et al., 2020; Hansen & Wang, 2021; Yarats et al., 2021; Bertoin et al., 2023; Almuzairee et al., 2024; Grooten et al., 2023). We used the smallest checkpoint (efficientvit_sam_l0), which natively supports 512×512 images. Reducing the resolution further led to noticeable performance degradation with only marginal speedup, so we retained the native resolution.

We extended EfficientViT-SAM to support batched image embedding computation. This significantly improved throughput since image embedding is the most computationally expensive step. Most SAM variants, including the official implementations, do not support batching. Additionally, we tested prompt-free segmentation but found it to be too slow for our pipeline (10-100x slower), so we opted for guided segmentation using text inputs.

**Efficient Network Computation.**  To optimize network computation, we used `torch.vmap` (Paszke et al., 2019) to batch forward passes across multiple networks, such as the twin critics in SAC. This allowed us to compute their outputs in a single pass, improving efficiency.

**Replay Buffer Design.**  We implemented a custom replay buffer using `TensorDict` (Bou et al., 2023) to efficiently handle the variable number of segments per frame. Since the segment embedding extraction module has no trainable weights, we compute embeddings once during data collection and store them directly in the buffer rather than storing raw images.

This approach reduces both memory usage and compute load. Segment embeddings have dimension 128∼256, and a typical frame has between 4 and 25 segments, this is significantly smaller than the original (3×512×512) image. This also avoids running SAM during training, which would otherwise make online RL infeasible.

**Variable-Length Sequence Handling.** Since each environment step yields a variable number of segments, a batch of $B$ steps results in a total of $N_{total}$ segments. Padding each sequence to a fixed length would be inefficient and would require an additional per-task hyperparameter. To avoid this, we adopt sequence packing, inspired by recent techniques in large language models (Krell et al., 2022). All segment embeddings are concatenated into a single sequence of shape $(1, N_{total}, D)$, and a causal attention mask is applied to ensure that each segment only attends to other segments from the same time step. This enables efficient and correct attention computation without the overhead of padding. SegDAC is only limited by available memory for the maximum number of segments it can process.

## I PRIOR WORKS COMPARISON

Table 47: Pixel-based RL baselines vs SegDAC.

| Criterion | SAC-AE | DrQ-v2 | MaDi | SADA | SegDAC (ours) |
|---|---|---|---|---|---|
| Needs Human Labels | ✓ No | ✓ No | ✓ No | ✓ No | ✓ No |
| Leverages Pretrained Vision Models | ✗ No | ✗ No | ✗ No | ✗ No | ✓ Yes |
| Leverages Text | ✗ No | ✗ No | ✗ No | ✗ No | ✓ Yes |
| Object-centric representations | ✗ No | ✗ No | ✗ No | ✗ No | ✓ Yes |
| Needs Auxiliary Losses | ✗ Yes | ✓ No | ✓ No | ✓ No | ✓ No |
| Needs Data Augs (RL training) | ✓ No | ✗ Yes | ✗ Yes | ✗ Yes | ✓ No |

## J SAMPLE EFFICIENCY

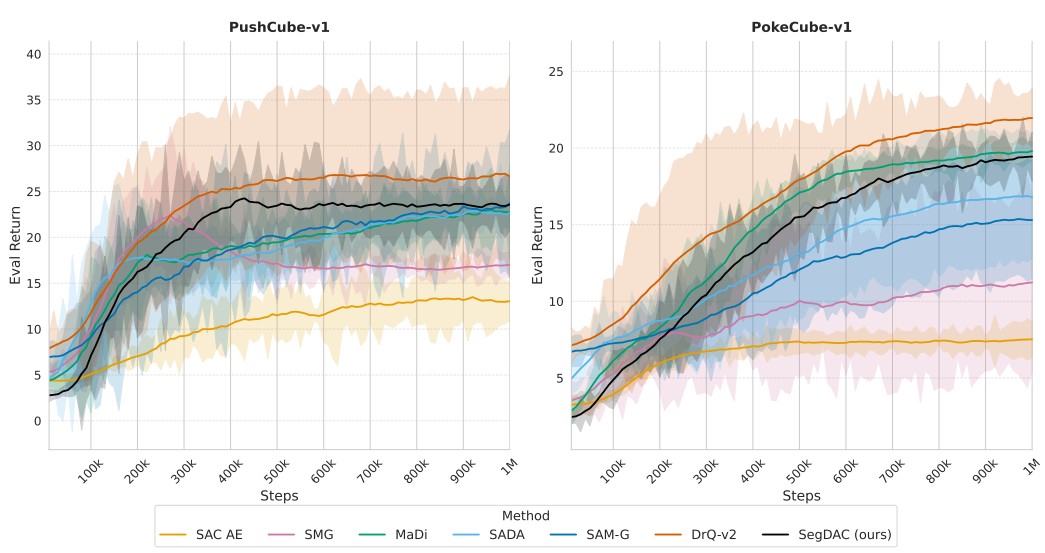

Figure 41: Evaluation return sample efficiency curves for the push/poke cube manipulation tasks.

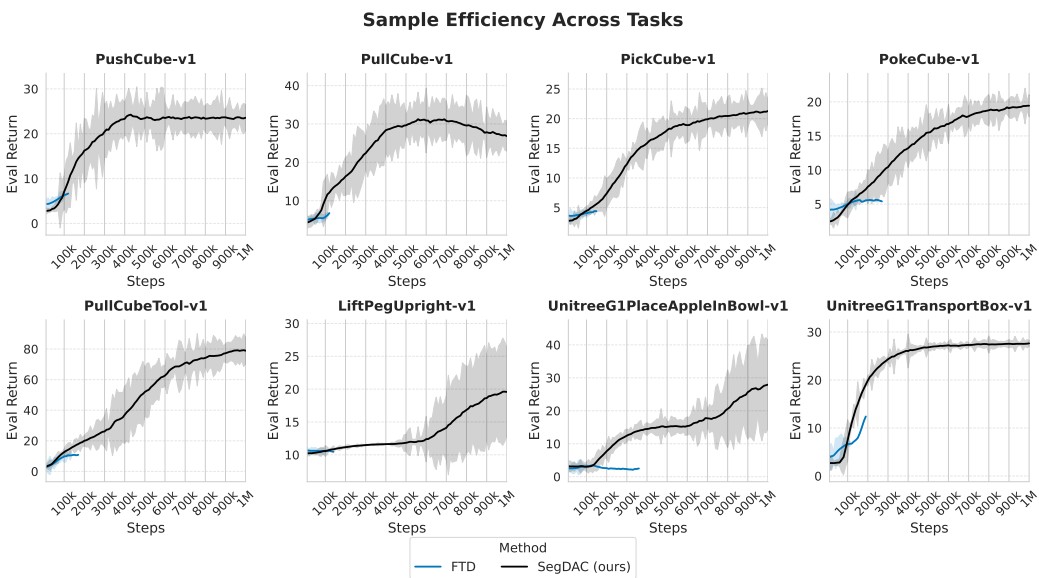

Figure 42: Evaluation return sample efficiency curves (partial results) comparing SegDAC to FTD, we could not complete trainings due to the large compute needed by FTD.

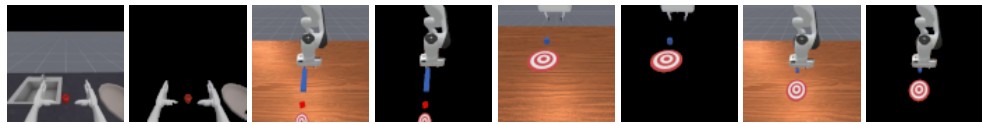

Figure 43: The reference frames and ground truth segmentation masks that SAM-G requires in order to operate (only some of the tasks are shown for brevity).

Figure 42 shows partial sample-efficiency results comparing SegDAC with FTD. Figure 43 illustrates the reference frames and ground-truth masks required by SAM-G (only some of the tasks are shown for brevity but the same is true for all tasks), whereas SegDAC does not rely on any ground-truth segmentation or reference frames. To produce these ground truth masks we used the first frame of a trajectory for each task then used SAM 3 playground platform and annotated each mask using 5-15 point prompts to guide SAM 3 until the segmentation mask was near perfect.

## K  SPEED ANALYSIS

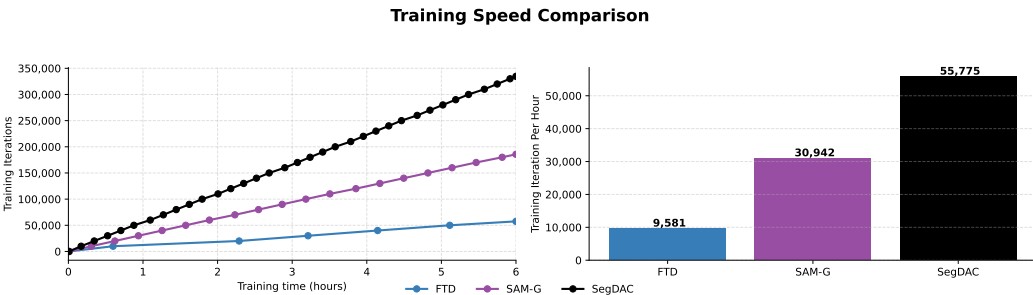

Figure 44: Training speed comparison of various methods that use SAM (includes env data collection + training).

Figure 44 shows the end to end training speed for methods that use SAM inside the RL loop. This includes data collection, SAM processing and all model updates (and periodic evaluation) . SegDAC

is noticeably faster than the previous SAM-based baselines even if we use a full transformer decoder instead of a light attention block on top of CNN features. FTD is slow because of its heavy auxiliary tasks and because it reconstructs RGB images from the segments. SAM-G also pays this cost since it learns from reconstructed images instead of working directly in latent space. FTD also relies on prompt-free segmentation which is much slower than text-guided segmentation.

SegDAC keeps the pipeline simple and efficient. We use a short list of concept words for YOLO-World, and we reuse the SAM encoder features directly without any reconstruction stage. Our object-centric embeddings are extracted with a fast procedure and we store them in the replay buffer instead of RGB images. This avoids re-running SAM during training and keeps the RL loop light. The transformer then processes a small set of segment embeddings which is far cheaper than operating on hundreds of patch embeddings, and this helps address the quadratic cost of attention. Prior SAM-based methods also rely on heavy mask refinement loops to improve masks over time. We avoid this and instead use a lightweight morphological post-processing step (see Appendix E) which takes only a few milliseconds on an L40S.

These design choices make it possible to use a full transformer architecture and still achieve better end to end speed. In practice this reduces the overhead per step and leads to faster overall training. SegDAC runs at almost twice the speed of SAM-G and FTD while keeping the full SAM perception stack.

## L    FROZEN WEIGHTS TRADE OFF

We use pretrained frozen weights because full end to end updates would be too slow and potentially unstable for online RL, especially since SAM must run at every step. Freezing perception keeps the features stable and makes training practical, but the features are not guaranteed to match the RL objective and may limit performance in some cases. Full finetuning could improve representations, yet in practice it often reduces stability, lowers sample efficiency and can damage pretrained encoders, so it is not obvious that it is always better. Pretrained features are also not perfect and joint representation learning can sometimes help, but this adds a lot of complexity in our setup because we would need to store RGB images in the replay buffer instead of segment embeddings and re-run SAM during training, which would be far too slow for online RL. In our work we found that using the SAM encoder to compute segment embeddings gave representations that were strong enough to make RL training efficient even with frozen weights. Our visual generalization experiments also show that learning representations from scratch during online RL can easily overfit the training environment. Partial finetuning is a promising middle ground and we see it as a natural direction for future work.

