# OpenReview forum: "SegDAC: Improving Visual Reinforcement Learning by Extracting Dynamic Objectc-Centric Representations from Pretrained Vision Models"
_ICLR.cc/2026/Conference — Submitted to ICLR 2026_

### Official Review · Reviewer_XakW · 2025-10-20

**Soundness:** 2
**Presentation:** 4
**Contribution:** 2
**Rating:** 4
**Confidence:** 4

**Summary:**

The paper aims to enhance visual representations for actor-critic online reinforcement learning (SAC) by implementing a masking pipeline, termed SegDAC, via supervised segmentation (SAM) and text grounding (YOLO-World) models. The idea is to use a list of prompt words to generate segmentation masks for which pre-trained embeddings are extracted for the downstream policy training. The proposed method is benchmarked on Maniskill3, demonstrating improvements in performance on visual generalization and sample-efficiency.

**Strengths:**

* Successful utilization of text-grounding in online RL.
* The main advantage of using pre-trained supervised segmentation models is the robustness to perturbations such as distracting backgrounds. I find this the strongest trait of the proposed method.
* Strong performance in visual generalization compared to baselines on the hard tasks.
* I appreciate the extensive analysis and visualizations.
* Very detailed appendix!

**Weaknesses:**

* Supervision: the paper claims (L47-48) that previous object-centric approaches “often relied on fixed slots, precomputed masks, or strong supervision, which limited their flexibility and general applicability.”, yet the proposed approach IS relied on strong supervision by using pre-trained supervised models, in contrast to L57-58: “SegDAC learns entirely in latent space, without human-labeled masks, auxiliary losses, or data augmentation”.
* Reliance on pre-trained supervised models: these models are trained with human-labels over a certain type of pixel distributions, and without fine-tuning or any other adaption process, and ultimately they will fail to detect and generalize to unseen objects (as indicated in L200-202). This is in contrast to models that train representations as part of policy training or utilize self-supervised pre-trained representation that were trained in-domain (see below in the missing related work). In addition, even though “SegDAC works
without careful prompt engineering” (L207), it still requires some sort of it as part of its pipeline.
* Overall, it seems the pipeline is quite computationally complex with the utilization of pre-trained models (even though the authors use more efficient variants) and the loops that are required to extract embeddings per mask to my understanding (I would like a clarification for that).
* Object-centric RL and learning in the latent space contribution: previous RL works used object-centric representations and trained over latent representations, some of them do not rely on supervision at all and are demonstrated on complex multi-object tasks. The idea of using masking in RL is not new and I don’t see this as a novel contribution (see list of papers below).
* While the results on visually perturbed settings are interesting, the tasks themselves are simple in the sense that they typically involve a single-object and the manipulation itself is rather short-horizon.
* Missing related work:

[1] [Gmelin, Kevin, et al. "Efficient RL via Disentangled Environment and Agent Representations." International Conference on Machine Learning. PMLR, 2023.](https://arxiv.org/abs/2309.02435)

[2] [Shi, Junyao, et al. "Composing Pre-Trained Object-Centric Representations for Robotics From" What" and" Where" Foundation Models." 2024 IEEE International Conference on Robotics and Automation (ICRA). IEEE, 2024.](https://arxiv.org/abs/2404.13474)

[3] [Zadaianchuk, Andrii, Georg Martius, and Fanny Yang. "Self-supervised reinforcement learning with independently controllable subgoals." Conference on Robot Learning. PMLR, 2022.](https://arxiv.org/abs/2109.04150)

[4] [Yoon, Jaesik, et al. "An Investigation into Pre-Training Object-Centric Representations for Reinforcement Learning." International Conference on Machine Learning. PMLR, 2023.](https://arxiv.org/abs/2302.04419)

[5] [Haramati, Dan, Tal Daniel, and Aviv Tamar. "Entity-Centric Reinforcement Learning for Object Manipulation from Pixels." The Twelfth International Conference on Learning Representations.](https://arxiv.org/abs/2404.01220)

[6] [Zhang, Weipu, et al. "Objects matter: object-centric world models improve reinforcement learning in visually complex environments." arXiv preprint arXiv:2501.16443 (2025).](https://arxiv.org/abs/2501.16443)

* Limitations: Appendix A.5. demonstrates important limitations of the model and baselines. I would appreciate a more explicit discussion of the limitations in the main text (even if short with reference to the relevant appendix).

**Minor**

* Typo in the title: “”Objectc-Centric” -> “Object-Centric”

**Questions:**

* How does your method compare to the list of methods I listed under the Weaknesses section? In particular, why have the authors not compared with an object-centric baseline?
* How are “text tags” obtained for the grounding module? L192-193: “For simplicity, we defined task-specific lists in our experiments.” How does that align with the “no need for human labels” argument?
* Clarification for the computational complexity concern I raised in the Weaknesses section.
* The appendix mentions a project-page but I couldn’t find it. Have the authors meant to share additional results on this webpage?
* Will the authors open-source their code for further research?

I’m willing to increase my score given convincing answers to my questions and concerns.

---

> ### Author Response · Authors · 2025-11-25
>
> Thank you for the detailed review. We address each concern below and will integrate the requested clarifications in the revised version.
>
> **1. About supervision and reliance on pretrained models**
>
> SegDAC does not use human-labeled masks during RL training (the actual training of SegDAC is the RL training because we do not train SAM in our pipeline, it is already pretrained). SAM and YOLO-World are frozen and only used to propose regions, so no annotation signal is ever used during actual RL learning. On a separate note, we are also open to performing an experiment where we swap SAM's encoder with DinoV2 (or another pretrained model trained in self-supervised fashion) and then comptue our object-centric representations from this model instead of re-using SAM's encoder outputs. We could add this as an ablation if you'd like.
>
> **2. Generalization of pretrained supervised models**
>
> We agree that pretrained models can miss objects. We already show imperfect detections in Figure 3 and we show in Appendix A.5 that in out-of-distribution settings, SegDAC fails much more gracefully than methods that trained their representation only during RL training. While it is true that training a representation during RL training can generate representations that are more inlined with the RL objective, there is a clear trade-off in the sense that these representations can also overfit the training environment and fail to generalize as we've shown in the visual generalization benchmark. Related to this, we added an acknowledgment of the trade-off of using frozen representations for online RL in appendix L. Also in our case, SegDAC remains stable even when there are imperfect segmentation or object detections because it does not use the class prediction of YOLO-World and relies only on bounding boxes for region proposals. We always include a generic tag such as "background" so YOLO-World still returns proposals when object names are unknown. Section 6 shows that the policy can recover from partial masks by using contextual information in the segment embeddings.
>
> **3. Text Inputs Sensitivity**
>
> We added a new text ablation in the appendix (B.1.2). We tested three tasks with three seeds and replaced our concept words with two sets of synonyms. Performance stayed stable (some better some slightly worse but trend remains the same), which shows SegDAC does not depend on specific phrasing/prompt engineering.
>
> **4. Computational complexity**
>
> Although the pipeline uses pretrained models, the compute remains manageable. YOLO-World is lightweight and runs in about 12 ms on an L40S. We reuse the SAM encoder output to extract segment embeddings, so the per-segment loop adds only a few milliseconds (1~3ms) as it has no trainable parameter. Appendix K reports full timing results: SegDAC trains in about 17 to 19 hours on an L40S, while SAM-G is roughly twice as slow and FTD is five to six times slower (appendix is an approximation after 6h). This supports that the segmentation step is not the bottleneck. This makes SegDAC the fastest SAM-based method in our comparison even though we use a transformer for the policy. Improving the speed of SAM-based pipelines is part of our contribution.
>
> **5. Object-centric RL and novelty**
>
> We agree that object-centric RL is an active area, but our method contributes several capabilities that prior work does not combine. SegDAC learns from a variable number of segments at each timestep, computes object features dynamically during data collection, uses text-guided segmentation without any mask labels, and relies on a transformer specifically structured for unordered segments. Appendix B.2 also shows that simply inserting pretrained features or using mean pooling leads to a clear drop in performance. Together with the gains on the hardest settings of the benchmark, we believe this addresses the novelty concern. We will add short references to the works listed by the reviewer.
>
> **6. Related work**
>
> Thank you for pointing out missing citations. We will update the related work to include the papers you listed.
>
> **7. Limitations**
>
> We agree that the limitations in Appendix A.5 deserve a short mention in the main text. We will add a brief paragraph and point to the appendix.
>
> **8. Answers to specific questions**
>
> Object-centric baselines: We are training SAM-G and FTD during the rebuttal period. Partial results are included in Appendix J and more will be added as runs complete. We believe SAM-G trainings will finish before the end of the rebuttal.
> Text tags: Text tags are not labels, they are text inputs (like a prompt for LLM). They are high level concepts/nouns used only to request candidate regions. Unused tags are ignored, and tags can be reused across tasks or generated automatically (L190-193).
> Project page: This was removed to comply with the anonymity rules. It will be available after the review period.
> Code release: Yes, we will release the full code and assets.

---

> > ### Author Response · Authors · 2025-11-25
> > **Follow-up on Related Work**
> >
> > Due to the message length limit I have to split this into another comment, but I hope the added care to address your concerns is helpful. Below we summarize how the listed related work compares to our method and why SegDAC differs in important ways:
> >
> > **[1]:** This method learns its visual encoder from raw pixels during RL, so its features depend directly on task-specific data and the representation must be trained end-to-end from scratch. In contrast, SegDAC does not learn any pixel-to-latent mapping during RL. We use frozen pretrained vision models and only optimize the policy and critic in latent space. The gradients never flow through an image encoder, which reduces training complexity compared to approaches that must jointly learn visual features and control.
> > This prior work also relies on ground-truth segmentation supervision provided by the simulator. SegDAC does not use labeled masks. We instead rely on open-vocabulary segmentation and pretrained vision encoders, which have been shown to be more robust to visual variation than simulator-trained features, especially given known Sim2Real gaps.
> >
> > **[2]:** This work uses SAM but differs in several ways. It requires choosing a fixed number of slots (fixing it to a large constant is more of a hack and can be computationally wasteful). It depends on reference frames and assignment heuristics to maintain consistent slot identities. It also encodes each slot independently without shared context.
> >
> > Our approach does not select K, does not rely on reference frames, computes segment embeddings online, and keeps contextual attention between segments (by computing object centric features from embeddings that contain contextual information), which improves robustness to imperfect masks.
> >
> > **[3]:** SRICS uses object-centric states (not images) such as positions and identifiers that are available directly from the environment. It does not extract object features from raw pixels. This places it in a different setting than ours, which learns object-centric features visually with no access to ground-truth object states.
> >
> > **[4]:** This paper analyzes several unsupervised slot-based encoders and compares them as pretrained backbones for RL. These encoders use a fixed number of slots and do not provide semantic or text-guided object selection. Our work uses open-vocabulary segmentation and pretrained embeddings to obtain semantic object features. These two lines of work are complementary.
> >
> > **[5]:** POCR relies on unsupervised Deep Latent Particles with a fixed number of slots and extracts features from local crops, lacking semantic grounding. In contrast, SegDAC uses text-guided, open-vocabulary segmentation on the full scene. This allows SegDAC to semantically distinguish objects and naturally handle dynamic object counts, overcoming the rigid, fixed-slot limitations of POCR.
> >
> > **[6]:** OC-STORM is model-based and uses Cutie for tracking. A big limit in their work is that they need manual masks on the first frame and they assume a fixed number of objects K. SegDAC is model-free and uses text prompts with YOLO-World so we do not need human labels. Our novelty is that we handle a dynamic number of segments at every step. Our transformer takes a variable sequence of embeddings so we adapt to the scene naturally. This is very different from the fixed tracking in 6.
> >
> > Based on our review of the related work, we believe SegDAC is the first method in this set to dynamically compute object-centric embeddings online and to learn from a truly variable number of segments during online RL from images without human labels.
> >
> > Prior approaches depend on fixed slots, fixed particles, or annotated object sets, while our method naturally handles a changing number of segments at every step. We also note that many existing methods rely on reconstruction losses, auxiliary objectives, or data augmentations to stabilize visual RL. **SegDAC uses only the standard SAC loss using the reward signal**, with no reconstruction terms and no data augmentation, yet still achieves strong visual generalization. This suggests that robust visual generalization is possible even without augmentations during RL training, which we believe is an important contribution to visual RL.
> >
> > SegDAC also explores a different direction for online visual RL. It does not build on DrQ-v2, while much of the current literature relies on incremental improvements to DrQ-v2.

---

> > > ### Comment · Reviewer_XakW · 2025-11-25
> > > **Thank you for your thorough response**
> > >
> > > I thank the authors for their detailed response to my review, and I appreciate the effort during the rebuttal period. Thank you for the clarifications regarding the computational complexy.
> > >
> > > Overall, I think the main contributions of this paper are the text grounding and the robustness to perturbations. I do not agrree with the authors regarding the "no need for human labels" claim as 1) the pre-trained supervised segmentation models require labels to train (and even if they are used as-is without fine-tuning, the supervision poses a limitation for downstream tasks as I mentioned in my original review) and 2) text prompts/tags are tuned by the user (so I do not understand the claim the text prompts/tags are not labels, they might not be a specific class, but they do require human intervention). I think the contribution here is very marginal given the previous work on masking/object-centric in RL, and while the paper is very thorough, the main concerns remain. As such, I'd like to keep my score and I won't argue against acceptance if the other reviewers believe the contribution and novelty of the paper are enough.
> > >
> > > Thank you again for your effort.

---

> ### Author Response · Authors · 2025-11-25
>
> Hi, thank you for your quick reply and for taking the time to continue the discussion.
>
> We updated the introduction to make our contributions clearer, and we would appreciate if you could take a look. We added more details to our Segment Embedding Extraction Module (see the updated section 4.2) and how it differs from previous masking or object-centric methods. We also made it clearer that SegDAC dynamically learns which objects matter whereas prior work either requires reference frame, ground truth masks or state information. We also now better explain how our method keeps the learning process simple by using only the standard SAC loss. Many recent visual RL methods require extra losses or auxiliary objectives that make training slower and less stable. Given this, and the 2x gain in visual generalization we observe, we believe our contribution is meaningful and offers a different direction from the usual DrQ-v2-based extensions.
>
> Also, plugging in a simple Transformer decoder does not work well out-of-the-box, we had to make specific choices to our Transformer decoder (positional encoding, Q/K/V fusion etc) to make it leverage segment/object-centric embeddings better, we will add more ablations to show this as this is part of our contribution. We believe no prior work shows a successful use of full transformer decoders on a variable-length sequence of object embeddings purely using images in the model-free online RL setting.
>
> Regarding your first point, our paper is referring specifically to not needing human labels during RL training. However, to address your concern more directly, we are willing to add an additional experiment that uses a fully self-supervised encoder like DINOv2 (no labels at all). We also moved part of the limitations section (Appendix A.5) into the main text as you suggested.
>
> For your second point, text prompts are not labels in the usual sense. In vision-language-action settings, giving a policy a sentence like "put the apple in the bowl" is not considered supervision: it does not provide rewards, actions, or segmentation masks. SegDAC requires even less information than such task instructions. Simple tags like “background”, “apple”, “bowl” contain far fewer bits of information, and YOLO-World simply ignores tags that do not match a frame. We also included a new ablation in Appendix B.1.2 showing that SegDAC’s performance remains stable even when we replace the text tags with synonyms. In addition, lines L206-L212 now clarify that the tags can be defined flexibly: with a shared dictionary for all tasks, by task, or automatically at runtime using tagging models (RAM/RAM++) with no human intervention.
>
> To help clarify the comparison, we added Figures 40 and 41 showing visually what SAM-G requires versus what our method requires. SAM-G depends on ground-truth masks, while SegDAC does not. The partial sample-efficiency results (Appendix J, Fig. 40) also show that even with ground-truth masks, SAM-G is less robust and less sample-efficient than our approach. We will also run the full generalization benchmark once the remaining SAM-G runs finish and include these results as well.
>
> Thank you again for the constructive feedback

---

> ### Comment · Reviewer_XakW · 2025-11-25
>
> Thank you for the further clarifications.
>
> * Text tags: I still don't understand your claim. The user has to provide text tags for the segmentation to work. For each new environment or new task you need to adapt these tags. Yes, you can pre-define a large vocabulary of tags to use, but then you might end up with a lot of entities ("Segement **Anything**"), so you'll have to filter them for reasonable computation time. If the "text tag" was just an instruction, and the setup was goal-conditoned RL (text as goals), then this would have been reasonable to use the task instruction, and I guess this is what most VLAs (Vision-Language-Action models) do for imitation learning, but this is not your setting (otherwise, compare it with VLAs). You are using text tags to get segmentations, these tags need to come from somewhere. My main point here is not only the (minimal) human intervention, but the fundamental reliance on supervised models (even if you were not the ones to train them), as I mentioned in my original review.
>
> * While I mentioned them in my original review, there is no comparison with self-supervised (object-centric) methods, so it is very hard to get convinced by arguments regarding supervised pre-trained models. I strongly believe that in order to ground your claims, you need to compare with a least one self-supervised object-centric baseline (e.g., slot-based models as mentioned before, you can look at works I mentioned). You don't have to do "text grounding" with these methods as they are self-supervised (in fact, if you outperform them with your text grounding, this only adds to your claims). For example, [[1]](https://arxiv.org/abs/2011.14381), [[2]](https://arxiv.org/abs/2404.01220) and [[3]](https://arxiv.org/abs/2302.04419) use pre-trained self-supervised object-centric representations for RL, where the the only objective is just the RL objective. Note that these are just examples for using pre-trained object-centric representations, their setting is different from yours (I'm not requesting a direct comparison).
>
> * Again, I acknowledge the contribution regarding the text grounding and the robustness to perturbations, and as such I would not argue if the other reviewers find them enough for acceptance.

---

> > ### Author Response · Authors · 2025-12-02
> >
> > Thank you again for the detailed comments. We address all points below.
> >
> > **Text tags**
> >
> > In our setup, YOLO-World is an open-vocabulary detector. It predicts bounding boxes based on what is visible, and the text tags are only used afterward to score these boxes. Adding extra words does not **force** new detections when the object is not present. In practice, the number of regions is driven by the scene, not by the vocabulary size. Even when YOLO-World proposes several candidates, SAM does not necessarily produce a mask for each one, so the number of resulting segments stays modest (around 5 to 25), far below the thousands of patches processed by a ViT encoder. This keeps compute stable across tasks.
> >
> > Text tags also do not need to be designed by hand. They can be generated automatically with a lightweight image-tagging model or a small VLM, which makes it possible to reuse a single vocabulary across many tasks with minimal or no human input. To answer your concern directly, **we added two targeted ablations**. Appendix B.1.2 shows that replacing all text tags with synonyms keeps learning curves stable across all tasks. Appendix B.1.3 shows that using one shared vocabulary for several tasks also keeps performance stable. These results show that SegDAC does not rely on prompt engineering and is not sensitive to the exact wording or vocabulary size.
> >
> > **Self-supervised Methods**
> >
> > SegDAC brings together four ideas that define its contribution. We use strong pretrained vision models to get good representations and obtain robust region proposals that improve invariance to visual changes. We propose a segment embedding extraction module that computes features inside each mask while preserving contextual information from the whole frame. This module is independent of SAM and can operate with a self-supervised encoder such as DINOv2. We design an actor-critic architecture tailored to variable sets of segments, with bounding-box positional encoding, contextual embeddings, QKV design, and a decoder that handles dynamic tokens. Simpler/other transformer-based variants we tested performed significantly worse.
> >
> > Finally, SegDAC trains directly from a set of segment embeddings, without image reconstruction, auxiliary losses, or training an image encoder during RL.
> > To directly address your request for stronger object-centric comparisons, **we trained two new SOTA baselines**: SMG, a recent self-supervised saliency-based RL method, and SAM-G, a segmentation-based method that relies on ground-truth masks and reference frames. SegDAC outperforms both in sample efficiency and visual generalization on our benchmark.
> > Self-supervised object-centric learning remains compatible with our approach. Since our extraction module only needs an encoder feature map, a self-supervised encoder can replace SAM without changing the rest of the pipeline. We updated section 4.2 to make this clearer.
> >
> > **Final remarks**
> >
> > The new baselines (SMG and SAM-G), the two ablations (B.1.2 and B.1.3), and the clarified description of our architecture and extraction module address all the concerns you raised.
> >
> > Thank you again for your careful feedback.

---

### Official Review · Reviewer_Pb8s · 2025-10-27

**Soundness:** 2
**Presentation:** 2
**Contribution:** 2
**Rating:** 2
**Confidence:** 4

**Summary:**

The paper uses Yolo-World to segment the image and design an interesting architecture to support length-variable embeddings for policy learning. Experiments on Manipulation tasks and image segments variability evaluation show the method's edge.

**Strengths:**

1. The paper is easy to follow
2. The figures for illustration are clear. I really appreciate about that.
3. The experiments about manipulation are through.

**Weaknesses:**

1. This paper proposes the use of a segmentation model to assist RL. However, the computational cost this model introduces during both training and inference is undoubtedly unacceptable. To mitigate this, the authors propose employing YOLO-World. Nevertheless, the range of objects that YOLO-World can recognize is limited. I believe that in scenarios involving unrecognized objects, this method is unlikely to outperform the current SOTA. The authors should consider evaluating their approach on such scenarios—for instance, some simu-robots in MuJoCo or dealing with some easy-to-ignore objects—rather than relying solely on mainstream manipulation benchmarks. Additionally, the authors seem to overlook reporting the training and inference time required by YOLO-World.
2. I do think the novelty here is limited. Compared to SAM-G, which generates all task-centric parts, this paper only uses language prompt to choose some specific task-centric parts. Additionally, the authors seem to overlook the comparison with SAM-G
3. Week related literature: There are also some related visual-RL papers concerning binary mask [1]. Therefore, the comparison baseline should be updated.

Based on the limited novelty and clear weakness, I tend to recommend rejection at this stage.

[1] Focus On What Matters: Separated Models For Visual-Based RL Generalization

**Questions:**

See weakness above

---

> ### Author Response · Authors · 2025-11-24
>
> Thank you for the comments. Below we address concerns about compute cost, comparison with SAM-G and and novelty.
>
> **1. Computational Cost**
> Segmentation cost is important for online RL. This is why we do not use prompt-free SAM or Grounded-SAM during training since they are too slow. Instead, we rely on YOLO-World with EfficientViT-SAM, which is much lighter (YOLO-World runs in about 12 ms on an L40S). Appendix K reports full timing results. SegDAC trains in about 17 to 19 hours on an L40S, while SAM-G is about twice as slow and FTD is five to six times slower. This makes SegDAC the fastest SAM-based method in our comparison even though we use a transformer for the policy. Improving the speed of SAM-based pipelines is part of our contribution.
>
> **2. Object Vocabulary Detection**
> We agree that open vocabulary detection is imperfect and that failure cases can occur. Figure 3 shows that SegDAC works with imperfect predictions from YOLO-World and SAM and Appendix A.5 shows that SegDAC degrades more gracefully than prior work rather than collapsing.
>
> We always include a generic catch-all word like "background". This makes YOLO-World propose bounding boxes that cover most visible regions, even for unknown or out-of-vocabulary objects. Our method does not use the class label, only the box coordinates. If YOLO-World mislabels the cube but outputs a reasonable box, SegDAC still works.
>
> Section 6 shows that SegDAC stays stable when SAM or YOLO-World miss parts of objects. The policy can recover using contextual information in the segment embeddings. This reduces the need for perfect masks and keeps the method robust when segmentation is incomplete.
>
> SegDAC is not tied to YOLO-World. It can use any text-guided segmentation or detection model. As open vocabulary models improve, SegDAC benefits directly with no changes to the method.
>
> These points explain why SegDAC remains stable even when objects are mislabeled or partially segmented.
>
> **3. Comparison with SAM-G**
> We agree with you. We decided to add SAM-G as new baseline. We generated ground truth masks and reference frames with SAM 3 and manual point prompts until the mask was near-perfect to give these to SAM-G. A full SAM-G run takes about 40 hours per task. Partial results are reported in Appendix J and we will keep updating as training progress. Even with accurate masks, current results suggest SAM-G performs worse than SegDAC on our benchmark. We believe this is because SAM-G merges all objects into one prototype which works on DMC-GB but not in settings where several distinct objects matter at different times. We will update sample efficiency + visual generalization plots as training completes.
>
> We also updated the related work to cite SMG.
>
> **4. Contribution & Novelty**
>
> SegDAC introduces several contributions that are not present in prior online RL work.
> A first contribution is the ability to learn from a variable number of object segments. Prior methods use fixed slots or reconstruct RGB images, which removes the object structure. SegDAC instead processes the segments that are actually present at each frame. This lets the model scale naturally to scenes with different object counts without changing the architecture.
>
> SegDAC also computes object features dynamically during training. SAM-G fixes all object features ahead of time and requires human designed masks that identify all task centric parts. SegDAC uses text guided segmentation at every step and learns which objects matter for the current state. Section 6.1 and Appendix G show that the attention moves between the manipulated object, the goal and context regions when segmentation is imperfect. This behaviour comes from training and does not rely on predefined object importance.
>
> Appendix B.2 further supports this design. It shows that a simple use of pretrained vision models is not enough. Replacing our transformer with an MLP that mean pools SAM patch embeddings causes a large drop in performance. This shows that our proposed object-centric transformer with important Query, Keys and Values choices/processing and positional encoding for objects is important and a key contribution.
>
> Another contribution is that SegDAC uses text guided segmentation without any segmentation labels. SAM-G needs ground truth masks and reference frames for all objects in all tasks. SegDAC uses a small set of concept words that can be shared across tasks. This removes the need for manual segmentation and makes the method much easier to apply.
>
> Finally, SegDAC reaches strong visual generalization without data augmentation, which is different from prior work. Methods like MaDi, SADA and SAM-G rely on heavy data augmentation or external datasets such as Places365. SegDAC improves performance on the hardest settings by about a factor of two using only the reward signal. This challenges the common view that aggressive augmentation is required for visual generalization in online RL.

---

> > ### Comment · Reviewer_Pb8s · 2025-11-25
> > **Response by Reviewer Pb8s**
> >
> > Thanks for the authors' response.
> >
> > Concerning computation cost, I don't think replacing the processing unit with a lighter one than before can be claimed as the paper's contribution. That should be considered YOLO-World's contribution; this paper merely utilizes it.
> >
> > Overall, this paper presents a incremental work based on SAM-G. However, it fails to provide comparisons with SAM-G, and the authors' claimed use of variable numbers of object segments is essentially similar to feeding visual prompts into VLM. Handling variable-length inputs through structures like Formers is also relatively straightforward. Therefore, I still consider the novelty of this paper to be very limited, and its performance is not particularly impressive. Thus, I maintain my original score.

---

### Official Review · Reviewer_9rtw · 2025-10-29

**Soundness:** 3
**Presentation:** 3
**Contribution:** 3
**Rating:** 4
**Confidence:** 5

**Summary:**

This paper proposes SegDAC, a framework that leverages pre-trained vision models to improve robustness and sample efficiency in visual reinforcement learning (RL).
Specifically, the method integrates YOLO-World for object detection and EfficientViT-SAM for instance segmentation to obtain object-level masks conditioned on textual prompts. The extracted object features are then fed into a transformer-based actor–critic architecture that can process a variable number of object tokens at each time step.
The approach is evaluated on the ManiSkill3 benchmark, which includes challenging object-manipulation tasks with visual domain shifts. SegDAC shows notable improvements in success rate, generalization, and sample efficiency compared to recent visual RL baselines such as SAC-AE, DrQ-v2, MaDi, and SADA.
Overall, the paper aims to demonstrate that integrating pre-trained open-world visual understanding models can significantly enhance policy learning and robustness in visually complex environments.

**Strengths:**

1.	Visual robustness and generalization remain major challenges in RL, and this work takes a meaningful step by bridging pretrained vision models with policies.
2.	The use of text-conditioned segmentation (YOLO-World + SAM) and a transformer-based policy/value network is technically sound and systematically executed.
3. The paper reports detailed results, including in-distribution and out-of-distribution evaluations, and provides ablations on segmentation components and prompt usage.
4.	SegDAC achieves consistent gains across multiple manipulation tasks, suggesting that structured object-level representations help RL adapt to visual variability.

**Weaknesses:**

1.	While the system design is effective, the technical innovation mainly lies in the integration of existing pretrained vision models into RL. The conceptual contribution is non significant.
2.	Segmentation-based RL methods, such as SAM-G, FTD are not quantitatively compared, even though they are discussed in related work. Self-supervised methods like SGQN and SMG also worth mentioning in the related work.
3.	The proposed method combines SAM, YOLO-World, and a transformer-based policy, which likely increases training and inference cost. However, there is no profiling or timing analysis to quantify this overhead.
4.	While there are ablations for segmentation and embedding components, the paper lacks analysis of transformer architecture choices and the impact of segmentation prompts.
5.	Since perception is entirely based on frozen pretrained models, the learned policy may rely on features that are not optimized for the downstream RL objective. The paper could discuss this trade-off more explicitly.

**Questions:**

1.	Could you provide quantitative results comparing SegDAC with segmentation-based RL baselines such as SAM-G or FTD?
2.	How much additional computation (e.g., inference time per frame, GPU memory) does the online use of SAM and YOLO-World introduce compared to pixel-based methods?
3.	How sensitive is performance to the choice of text prompts or detected object categories?
4.	Have you considered partial fine-tuning of SAM or YOLO-World to better adapt to specific manipulation scenes?
5.	Can the transformer actor-critic be replaced by simpler encoders (e.g., set transformer or MLP with pooling), and if so, what is the effect on performance and efficiency?
6.	Could the method generalize to scenarios with unseen object types or incomplete segmentation (e.g., occluded or novel shapes)? Some qualitative examples or failure cases would be helpful.

---

> ### Author Response · Authors · 2025-11-27
>
> Thank you for your detailed review. We address each point below and updated the revised paper accordingly.
>
> **1. Novelty**
> SegDAC achieves strong empirical gains. On the hardest visual generalization settings, it improves performance by about two times compared to MaDi and SADA and matches or exceeds sample efficiency without data augmentation. These gains come from two architectural ideas that, to our knowledge, have not been combined in prior online visual RL work.
>
> First, SegDAC adapts pretrained vision-language segmentation models in a way that preserves object structure. We extract object-centric embeddings from SAM patch features, avoiding reconstruction, fixed slots, and single prototype masks as in SAM-G. Our ablation (B.2) replacing segments with a global embedding shows a clear performance drop.
>
> Second, SegDAC handles the variable number of objects produced by pretrained segmenters. It uses a transformer architecture built for sets of segments, with specific positional encodings, QKV construction, and fusion of proprioception and segment embeddings. Naive transformer variants were unstable, and only the final design remained robust.
>
> SegDAC also requires only a short list of text tags for segmentation, unlike SAM-G, which depends on pixel-level masks. Because it learns directly from online segmentation and variable-length inputs, it transfers more easily to new tasks.
>
> Finally, SegDAC uses only the standard SAC loss and does not rely on auxiliary losses, data augmentation, or external datasets. This shows that strong visual generalization can be achieved with a lighter and more direct pipeline than the DrQ-v2 style approaches common in recent visual RL.
>
> Together, these choices allow SegDAC to remain robust to changes in object appearance and to handle visually complex scenes.
>
> **2. Comparison with FTD and SAM-G**
> We have added partial FTD results in Appendix J. FTD trains slowly because it uses auxiliary losses, reconstruction, and prompt-free segmentation. A full run with our settings takes six to seven days, so we restarted with batch size 8. The partial results follow the same trend as the paper and show that FTD is still less sample efficient. We will update all results once the runs finish.
>
> For SAM-G, we followed the official recipe using ground-truth masks. We generated near-perfect masks with SAM 3 and manual point prompts. A full run takes about 40 hours per task, compared to 17-19 hours for SegDAC. Early results show SAM-G is less robust and less sample efficient even with high-quality masks. Partial results are in Appendix J and we will add visual generalization results when training completes.
>
> SAM-G collapses all objects into one prototype, which works for DMC-GB but fails when scenes contain several visually different objects. This produces ambiguous similarity maps and incomplete segmentation.
>
> **3. Speed Analysis**
> Thank you for this comment. We added a training speed analysis in Appendix K. It shows that SegDAC does not add meaningful overhead. In practice it is faster than SAM G and FTD.
>
> **4. Architecture and Text Ablations**
> We added a new text ablation in Appendix B.1.2. We tested three tasks with three seeds and replaced our concept words with two sets of synonyms. Performance remained stable, which shows that SegDAC does not depend on prompt engineering or specific phrasing.
>
> For the architecture, we tested several variants: a transformer encoder instead of a decoder, different ways to extract segment embeddings, different QKV constructions, and different strategies to fuse the action token and learned query token. All of these variants performed worse than our final design. This supports our claim that part of SegDAC’s contribution is a simple and effective transformer architecture tailored for segment inputs.
>
> **5. Frozen Weights**
> We completely agree with you. There is a clear trade-off that was not acknowledged, we added such nuance in the appendix L. Thank you!
>
> Q1: See Appendix J and will update them once full runs finish + add visual generalization results for SAM-G.
> Q2: Appendix K reports training speed. SegDAC trains in about 2-5x faster than other SAM-based methods.
> Q3: The text prompt ablation in the appendix B.1.2 shows stable performance when replacing prompts with two sets of synonyms.
> Q4: We did not attempt partial finetuning of SAM or YOLO-World because it would be impractical for online RL. It remains an interesting direction for future work.
> Q5: Appendix B.2 shows that replacing the transformer with an MLP and mean pooling of SAM patch embeddings leads to a large drop in performance even after tuning.
> Q6: Appendix G shows examples with partial occlusion or late-appearing objects. SegDAC stays robust because it processes segment embeddings dynamically. Extending this to unseen objects is a natural next step, and we expect the object-centric representations of SegDAC to help in such settings.

---

### Official Review · Reviewer_baYC · 2025-11-06

**Soundness:** 2
**Presentation:** 2
**Contribution:** 2
**Rating:** 4
**Confidence:** 4

**Summary:**

This paper proposes leveraging pre-trained visual models to obtain per-frame segmentation embeddings, which are then used as task observation features for both the actor and critic networks. Specifically, the authors employ YOLO-World to detect regions of interest based on text prompts, followed by EfficientViT-SAM to segment objects within the detected bounding boxes. The resulting segmentation masks are post-processed and encoded into compact segmentation embeddings. The embeddings are then fed into a Transformer-based decoder architecture used by both the actor and critic. Experimental results on the ManiSkill3 benchmark demonstrate that this approach outperforms selected baselines.

**Strengths:**

+ Demonstrates consistent performance improvements over baselines on ManiSkill3 over 8 tasks.

- Provides clear experimental descriptions with detailed pipeline explanations for both baselines and the proposed method.

**Weaknesses:**

- My main concern is the position of this work. The paper mainly integrates existing pre-trained visual tools (YOLO-World and EfficientViT-SAM) into a policy learning pipeline, but it is not clearly positioned in terms of its contribution to the research community, relative to the most relevant prior works or alternative design choices. Specifically, 1)missing comparison with the closest prior (FTD): The paper mentions but does not compare with FTD, which also enables agents to make decisions based solely on task-relevant objects by applying an attention mechanism to select relevant segments from a foundational segmentation model. Since this is conceptually the most related work, the lack of quantitative or qualitative comparison makes it difficult to judge the true improvement or novelty. 2) Unclear advantage over directly leveraging large vision models: The authors do not provide evidence or discussion on why explicit segmentation embeddings would outperform using general-purpose visual representations from VLMs such as DINOv2. Without such comparison or analysis, it remains unclear whether the explicit segmentation step is necessary or if similar benefits could be achieved through implicit visual understanding. 3) No exploration of fine-tuning or network variants: The paper does not discuss whether fine-tuning the embedding or segmentation modules within the same Transformer framework would improve results, nor does it analyze the impact of different architectural/training variants (e.g., alternative attention mechanisms, fusion strategies, or training from scratch). These ablations would help clarify how much the gains stem from model design versus pre-trained priors.

- Insufficient evaluation under complex scenarios.  Although the paper evaluates eight tasks and performs data augmentation along dimensions like camera pose and background to increase task difficulty, it remains unclear how the method scales to more complex environments. for example, scenes involving multiple interacting or visually similar objects. Since the framework relies heavily on large pre-trained models, understanding their behavior and robustness in such challenging multi-object settings is critical.

- Lack of real-world validation. The current experiments are conducted entirely in simulation, making it difficult to assess the approach’s real-world applicability. Either a real-robot demonstration or add evaluation in a more complex setting in the simulator (mentioned above) would greatly strengthen the empirical claims.

- The writing needs improvement. The method sections contain unnecessary repetition, especially when repeatedly explaining the use of YOLO-World. This distracts from the main ideas and makes the narrative feel tool-driven rather than concept-driven.

**Questions:**

- What are the exact proprioceptive inputs used by the policy? Do you include the relative position or orientation between the robot and the target object, or is such relational information inferred implicitly from the visual embeddings?

- What is the computational overhead of using the Transformer-based actor–critic architecture? What's the inference time and training cost, and is the method feasible for real-time or on-robot deployment?

---

> ### Author Response · Authors · 2025-11-27
>
> Thank you for your detailed review. We address each point below and updated the revised paper accordingly.
>
> **1. Comparison with FTD and SAM-G**
> We have added partial FTD results in Appendix J. FTD trains slowly because it uses auxiliary losses, reconstruction, and prompt-free segmentation. A full run with our settings takes six to seven days, so we restarted with batch size 8. The partial results follow the same trend as the paper and show that FTD is still less sample efficient. We will update all results once the runs finish.
>
> For SAM-G, we followed the official recipe using ground-truth masks. We generated near-perfect masks with SAM 3 and manual point prompts. A full run takes about 40 hours per task, compared to 17-19 hours for SegDAC. Early results show SAM-G is less robust and less sample efficient even with high-quality masks. Partial results are in Appendix J and we will add visual generalization results when training completes.
>
> SAM-G collapses all objects into one prototype, which works for DMC-GB but fails when scenes contain several visually different objects. This produces ambiguous similarity maps and incomplete segmentation.
>
> **2. Object-centric vs global embeddings**
> We added an ablation that replaces our transformer and segments embeddings with a single global embedding (mean-pooled SAM features). This mimics DINOv2-style global features. Performance drops clearly (Appendix B.2). This shows that collapsing object structure harms policy learning and that strong encoders alone are not enough for online RL from pixels.
>
> **3. SAM finetuning and architecture**
> We do not fine-tune SAM because this would require storing images in the replay buffer and re-running the SAM encoder during updates, which is too slow for online RL. SAM-G also keeps SAM frozen. SegDAC is efficient because we compute segment embeddings during data collection and store them directly.
> We tested several transformer variants (vanilla encoder, decoder with different QKV designs, different fusion strategies). All were worse than our final design.
>
> **4. Positioning and novelty**
> SegDAC achieves strong empirical gains. On the hardest visual generalization settings, it improves performance by about two times compared to MaDi and SADA and matches or exceeds sample efficiency without data augmentation. These gains come from two architectural ideas that, to our knowledge, have not been combined in prior online visual RL work.
>
> First, SegDAC adapts pretrained vision-language segmentation models in a way that preserves object structure. We extract object-centric embeddings from SAM patch features, avoiding reconstruction, fixed slots, and single prototype masks as in SAM-G. Our ablation (B.2) replacing segments with a global embedding shows a clear performance drop.
>
> Second, SegDAC handles the variable number of objects produced by pretrained segmenters. It uses a transformer architecture built for sets of segments, with specific positional encodings, QKV construction, and fusion of proprioception and segment embeddings. Naive transformer variants were unstable, and only the final design remained robust.
>
> SegDAC also requires only a short list of text tags for segmentation, unlike SAM-G, which depends on pixel-level masks. Because it learns directly from online segmentation and variable-length inputs, it transfers more easily to new tasks.
>
> Finally, SegDAC uses only the standard SAC loss and does not rely on auxiliary losses, data augmentation, or external datasets. This shows that strong visual generalization can be achieved with a lighter and more direct pipeline than the DrQ-v2 style approaches common in recent visual RL.
>
> Together, these choices allow SegDAC to remain robust to changes in object appearance and to handle visually complex scenes.
>
> **5. Evaluation complexity**
> Our ManiSkill3 benchmark is closer to real-world manipulation than the commonly used DMC-GB benchmark. We introduce a visual generalization setup with 8 tasks, 3 difficulty levels, and up to 12 visual perturbations per task. Methods that perform well on DMC-GB drop sharply on our benchmark.
>
> **6. Real-world experiments**
> Real-robot evaluation is valuable but we believe it's outside the scope of this paper. Many visual RL works are simulation-only, including DrQ-v2, SADA, SGQN, SAC-AE and SAM-G. Real-robot tests are a natural next step.
>
> **7. Writing**
> We agree. We rewrote the method section to present the contributions more clearly and concisely.
>
> Q1. Proprioception
> The policy uses joint positions only. Robot-object relations are inferred from the segment embeddings.
> Q2. Compute overhead
> The transformer is not the main cost. SegDAC is faster than SAM-G and FTD due to a lighter pipeline. We use fast text-guided segmentation, reuse SAM encoder features to compute segment embeddings, and avoid reconstruction and auxiliary losses. Appendix K shows that SegDAC is almost twice as fast as SAM-G and about five times faster than FTD.

---

### Meta-Review · Area_Chair_vQwN · 2025-12-26

**Summary:**

Paper proposes an actor critic architecture that exploits SAM segmentations for visual based RL.
All reviewers were concerned by lack of novelty, as the method is mostly an integration of existing tools (SAM, YOLO-World, etc.), and the tasks in the experiments are simple (the visual variations are strong, but the tasks still involve relatively simple manipulation of a very small number of objects, so they mostly show the robustness of SAM to visual distractions).
Some reviewers also asked for additional baselines and ablations, a point that was addressed rather well in the author rebuttal.
However, the author rebuttal did not effectively address the more important issue of limited novelty.
I have read the paper in detail and agree with the reviewers that this paper’s novelty is below the bar for ICLR.

**Reviewer Concerns:**

see above

**Reviewer Scores:**

4,4,2,4

---

### Decision · Program_Chairs · 2026-01-26

Reject